# On Mesa-Optimization in Autoregressively Trained Transformers: Emergence and Capability

**Chenyu Zheng**[1,2], **Wei Huang**[3], **Rongzhen Wang**[1,2], **Guoqiang Wu**[4],
**Jun Zhu**[5], **Chongxuan Li**[1,2*]

[1] Gaoling School of Artificial Intelligence, Renmin University of China
[2] Beijing Key Laboratory of Big Data Management and Analysis Methods
[3] RIKEN AIP  [4] School of Software, Shandong University
[5] Dept. of Comp. Sci. & Tech., BNRist Center, THU-Bosch ML Center, Tsinghua University
{cyzheng,wangrz,chongxuanli}@ruc.edu.cn; wei.huang.vr@riken.jp;
guoqiangwu@sdu.edu.cn; dcszj@mail.tsinghua.edu.cn

## Abstract

Autoregressively trained transformers have brought a profound revolution to the world, especially with their in-context learning (ICL) ability to address downstream tasks. Recently, several studies suggest that transformers learn a mesa-optimizer during autoregressive (AR) pretraining to implement ICL. Namely, the forward pass of the trained transformer is equivalent to optimizing an inner objective function in-context. However, whether the practical non-convex training dynamics will converge to the ideal mesa-optimizer is still unclear. Towards filling this gap, we investigate the non-convex dynamics of a one-layer linear causal self-attention model autoregressively trained by gradient flow, where the sequences are generated by an AR process $x_{t+1} = W x_t$. First, under a certain condition of data distribution, we prove that *an autoregressively trained transformer learns $W$ by implementing one step of gradient descent to minimize an ordinary least squares (OLS) problem in-context. It then applies the learned $\widehat{W}$ for next-token prediction*, thereby verifying the mesa-optimization hypothesis. Next, under the same data conditions, we explore the capability limitations of the obtained mesa-optimizer. We show that a stronger assumption related to the moments of data is the sufficient and necessary condition that the learned mesa-optimizer recovers the distribution. Besides, we conduct exploratory analyses beyond the first data condition and prove that generally, the trained transformer will not perform vanilla gradient descent for the OLS problem. Finally, our simulation results verify the theoretical results, and the code is available at *https://github.com/ML-GSAI/MesaOpt-AR-Transformer*.

## 1   Introduction

Foundation models based on transformers [1] have revolutionized the AI community in lots of fields, such as language modeling [2; 3; 4; 5; 6], computer vision [7; 8; 9; 10] and multi-modal learning [11; 12; 13; 14; 15]. The crux behind these large models is a very simple yet profound strategy named *autoregressive (AR) pretraining*, which encourages transformers to predict the next token when a context is given. In terms of the trained transformers, one of their most intriguing properties is the *in-context learning (ICL) ability* [2], which allows them to adapt their computation and perform downstream tasks based on the information (e.g. examples) provided in the context without any updates to their parameters. However, the reason underlying the emergence of ICL ability is still poorly understood.

---

*Correspondence to Chongxuan Li.

38th Conference on Neural Information Processing Systems (NeurIPS 2024).

Recently, we are aware that some preliminary studies [16; 17] have attempted to understand the ICL ability from the AR training and connected its mechanisms to a popular hypothesis named *mesa-optimization* [18], which suggests that transformers learn some algorithms during the AR pretraining. In other words, the inference process of the trained transformers is equivalent to optimizing some inner objective functions on the in-context data.

Concretely, the seminal work [16] constructs a theoretical example where a single linear causally-masked self-attention layer with manually set parameters can predict the next token using one-step gradient-descent learning for an ordinary least squares (OLS) problem over the historical context. Moreover, they conduct numerous empirical studies to establish a close connection between autoregressively trained transformers and gradient-based mesa-optimization algorithms. Built upon the setting of [16], recent work [17] precisely characterizes the pretraining loss landscape of the one-layer linear transformer trained on a simple first-order AR process with a fixed full-one initial token. As a result, they find that the optimally-trained transformer recovers the theoretical construction in [16]. However, their results rely on imposing the diagonal structure on the parameter matrices of the transformer and do not discuss whether the practical non-convex dynamics can converge to the ideal global minima. Besides, it is still unclear about the impact of data distribution on the trained transformer, which has been proven to be important in practice [19; 20; 21; 22] and theory [23].

In this paper, we take a further step toward understanding the mesa-optimization in autoregressively trained transformers. Specially, without an explicit diagonal structure assumption, we analyze the non-convex dynamics of a one-layer linear transformer trained by gradient flow on a controllable first-order AR process, and try to answer the following questions rigorously:

1. *When do mesa-optimization algorithms emerge in autoregressively trained transformers?*
2. *What is the capability limitation of the mesa-optimizer if it does emerge?*

Our first main contribution is to characterize a sufficient condition (Assumption 4.1) on the data distribution for a mesa-optimizer to emerge in the autoregressively trained transformer, in Section 4.1. We note that any initial token $x_1$ whose coordinates $x_{1i}$ are i.i.d. random variables with zero mean and finite moments satisfy this condition, including normal distribution $\mathcal{N}(\mathbf{0}_d, \sigma^2 \mathbf{I}_d)$. Under this assumption, the non-convex dynamics will exactly converge to the theoretical construction in [16] without any explicit structural assumption in [17], resulting in the trained transformer implementing one step of gradient descent for the minimization of an OLS problem in-context.

Our second main contribution is to characterize the capability limitation of the obtained mesa-optimizer under Assumption 4.1 in Section 4.2. We characterize a stronger assumption (Assumption 4.2) related to the moment of data distribution as the necessary and sufficient condition that the mesa-optimizer recovers the true distribution (a.k.a. predict the next token correctly). Unfortunately, we find that the mesa-optimizer can not recover the data distribution when the initial token is sampled from the standard normal distribution, which suggests that ICL by AR pretraining [16; 17] is different from ICL by few-shot learning pretraining [24; 25; 26; 27; 28; 29; 30; 23] (see details in Section 2), a setup attracting attention from many theorists recently. We think ICL in the setting of AR pretraining needs more attention from the theoretical community.

In Section 4.3, we further study the convergence of the training dynamics when Assumption 4.1 does not hold anymore by adopting the setting in [17]. In this case, as a complement of [17], we prove that under a similar but weaker structural assumption, training dynamics will converge to the theoretical construction in [16] and the trained transformer implements exact gradient-based mesa-optimization. However, we prove that without any structural assumption, the trained transformer will not perform vanilla gradient descent for the OLS problem in general. Finally, we conduct simulations to validate our theoretical findings in Section 6.

## 2 Additional related work

### 2.1 Mesa-optimization in ICL for few-shot linear regression

In addition to AR pretraining, much more empirical [31; 25; 32; 33] and theoretical studies [24; 26; 27; 28; 29; 34; 30] have given evidence to the mesa-optimization hypothesis when transformers are trained to solve few-shot linear regression problem with the labeled training instances in the context. On the experimental side, for example, the seminal empirical works by [31; 35] considers

ICL for linear regression, where they find the ICL performance of trained transformers is close to the OLS. On the theoretical side, considering a one-layer linear transformer, [26; 24] prove that the global minima of the population training loss is equivalent to one step of preconditioned gradient descent. Notably, [24] further proves that the training dynamics do converge to the global minima, and the obtained mesa-optimizer solves the linear problem. For multi-layer attention models, recent works suggest that they can perform efficient high-order optimization algorithms such as Newton's method [36; 37; 34]. Unfortunately, the pretraining goal of these studies is different from the AR training. Therefore, it is still unclear whether these findings can be transferred to transformers autoregressively trained on sequential data.

## 2.2 Other explanations for ICL

In addition to the mesa-optimization hypothesis, there are other explanations for the emergence of ICL. [38; 39; 40; 41] explain ICL as inference over an implicitly learned topic model. [42] connects ICL to multi-task learning and establishes generalization bounds using the algorithmic stability technique. [43] and [44] study the implicit bias of the next(last)-token classification loss when each token is sampled from a finite vocabulary. Specially, [44] proves that self-attention with gradient descent learns an automaton that generates the next token by hard retrieval and soft composition. [45] explains ICL as kernel regression. These results are not directly comparable to ours because we study the ICL of a one-layer linear transformer with AR pretraining on the first-order AR process.

## 3 Problem setup

**Elementary notations.** We define $[n] = \{1, 2, \ldots, n\}$. We use lowercase, lowercase boldface, and uppercase boldface letters to denote scalars, vectors, and matrices, respectively. For a vector $\boldsymbol{a}$, we denote its $i$-th element as $a_i$. For a matrix $\boldsymbol{A}$, we use $\boldsymbol{A}_{k:}$, $\boldsymbol{A}_{:k}$ and $A_{ij}$ to denote its $k$-th row, $k$-th column and $(i, j)$-th element, respectively. For a vector $\boldsymbol{a}$ (matrix $\boldsymbol{A}$), we use $\boldsymbol{a}^*$ ($\boldsymbol{A}^*$) to denote its conjugate transpose. Similarly, we use $\overline{\boldsymbol{a}}$ and $\overline{\boldsymbol{A}}$ to denote their element-wise conjugate. We denote the $n$-dimensional identity matrix by $\boldsymbol{I}_n$. We denote the one vector of size $n$ by $\mathbf{1}_n$. In addition, we denote the zero vector of size $n$ and the zero matrix of size $m \times n$ by $\mathbf{0}_n$ and $\mathbf{0}_{m \times n}$, respectively. We use $\otimes$ and $\odot$ to denote the Kronecker product and the Hadamard product, respectively. Besides, we denote $\mathrm{Vec}(\cdot)$ the vectorization operator in column-wise order.

### 3.1 Data distribution

We consider a first-order AR process as the underlying data distribution, similar to recent works on AR pretraining [16; 17]. Concretely, to generate a sequence $(\boldsymbol{x}_1, \ldots, \boldsymbol{x}_T) \in \mathbb{C}^{d \times T}$, we first randomly sample a unitary matrix $\boldsymbol{W} \in \mathbb{C}^{d \times d}$ uniformly from a candidate set $\mathcal{P}_{\boldsymbol{W}} = \{\mathrm{diag}(\lambda_1, \ldots, \lambda_d) \mid |\lambda_i| = 1, \forall i \in [d]\}$ and the initial data point $\boldsymbol{x}_1$ from a controllable distribution $\mathcal{D}_{\boldsymbol{x}_1}$ to be specified later, then the subsequent elements are generated according to the rule $\boldsymbol{x}_{t+1} = \boldsymbol{W} \boldsymbol{x}_t$ for $t \in [T-1]$. For convenience, we denote the vector $(\lambda_1, \ldots, \lambda_d)^\top$ by $\boldsymbol{\lambda}$. We note that the structural assumption on $\boldsymbol{W}$ is standard in the literature on learning problems involving matrices [46; 17]. In addition, it is a natural extension of the recent studies on ICL for linear problems [24; 26; 28; 29; 30], where they focus on $y = \boldsymbol{w}^\top \boldsymbol{x} = \mathbf{1}_d^\top \mathrm{diag}(\boldsymbol{w}) \boldsymbol{x}$. Furthermore, adopting this new data model is a necessary approach to investigate AR pretraining because the regression dataset in the previous ICL theory is not suitable since each token $\boldsymbol{x}_i$ does not have a relation with the context. In this paper, we mainly investigate the impact of the initial distribution $\mathcal{D}_{\boldsymbol{x}_1}$ on the convergence result of the AR pretraining.

### 3.2 Model details

**Linear causal self-attention layer.** Before introducing the specified transformer module we will analyze in this paper, we first recall the definition of the standard causally-masked self-attention layer [1], whose parameters $\boldsymbol{\theta}$ includes a query matrix $\boldsymbol{W}^Q \in \mathbb{R}^{d_k \times d_e}$, a key matrix $\boldsymbol{W}^K \in \mathbb{R}^{d_k \times d_e}$, a value matrix $\boldsymbol{W}^V \in \mathbb{R}^{d_v \times d_e}$, a projection matrix $\boldsymbol{W}^P \in \mathbb{R}^{d_e \times d_v}$ and a normalization factor $\rho_t > 0$. At time step $t$, let $\boldsymbol{E}_t = (\boldsymbol{e}_1, \ldots, \boldsymbol{e}_t) \in \mathbb{R}^{d_e \times t}$ be the tokens embedded from the prompt sequence

$P_t = (\boldsymbol{x}_1, \ldots, \boldsymbol{x}_t) \in \mathbb{C}^{d \times t}$, the causal self-attention layer will output

$$\boldsymbol{f}_t(\boldsymbol{E}_t; \boldsymbol{\theta}) = \boldsymbol{e}_t + \boldsymbol{W}^P \boldsymbol{W}^V \boldsymbol{E}_t \cdot \text{softmax}\left(\frac{(\boldsymbol{W}^K \boldsymbol{E}_t)^* \boldsymbol{W}^Q \boldsymbol{e}_t}{\rho_t}\right) \in \mathbb{C}^{d_e},$$

where the $\text{softmax}$ operator is applied to each column of the input matrix. Similar to recent theoretical works on ICL with few-shot pretraining [24; 25; 16; 17; 26; 27; 28; 29], we consider the linear attention module in this work, which modifies the standard causal self-attention by dropping the $\text{softmax}$ operator. Reparameterizing $\boldsymbol{W}^{KQ} = \boldsymbol{W}^{K*} \boldsymbol{W}^Q$ and $\boldsymbol{W}^{PV} = \boldsymbol{W}^P \boldsymbol{W}^V$, we have $\boldsymbol{\theta} = (\boldsymbol{W}^{KQ}, \boldsymbol{W}^{PV})$, and the output can be rewritten as:

$$\boldsymbol{f}_t(\boldsymbol{E}_t; \boldsymbol{\theta}) = \boldsymbol{e}_t + \boldsymbol{W}^{PV} \boldsymbol{E}_t \cdot \frac{\boldsymbol{E}_t^* \boldsymbol{W}^{KQ} \boldsymbol{e}_t}{\rho_t}.$$

Though it is called linear attention, we note that the output $\boldsymbol{f}_t(\boldsymbol{E}_t; \boldsymbol{\theta})$ is non-linear w.r.t. the input tokens $\boldsymbol{E}_t$ due to the existence of the $\boldsymbol{E}_t \boldsymbol{E}_t^*$. In terms of the normalization factor $\rho_t$, like existing works [24; 26], we take it to be $t - 1$ because each element in $\boldsymbol{E}_t \boldsymbol{E}_t^*$ is a Hermitian inner product of two vectors of size $t$.

**Embeddings.** We adopt the natural embedding strategy used in recent studies on AR learning [16; 17]. Given a sequence $\boldsymbol{P}_t = (\boldsymbol{x}_1, \ldots, \boldsymbol{x}_t)$, the $i$-th token is defined as $\boldsymbol{e}_i = (\boldsymbol{0}_d^\top, \boldsymbol{x}_i^\top, \boldsymbol{x}_{i-1}^\top)^\top \in \mathbb{C}^{3d}$, thus the corresponding embedding matrix $\boldsymbol{E}_t$ can be formally written as:

$$\boldsymbol{E}_t = (\boldsymbol{e}_1, \ldots, \boldsymbol{e}_t) = \begin{pmatrix} \boldsymbol{0}_d & \boldsymbol{0}_d & \cdots & \boldsymbol{0}_d \\ \boldsymbol{x}_1 & \boldsymbol{x}_2 & \cdots & \boldsymbol{x}_t \\ \boldsymbol{x}_0 & \boldsymbol{x}_1 & \cdots & \boldsymbol{x}_{t-1} \end{pmatrix} \in \mathbb{C}^{3d \times t},$$

where $\boldsymbol{x}_0 = \boldsymbol{0}_d$ as a complement. This embedding strategy is a natural extension of the existing theoretical works about ICL for linear regression [30; 24; 25; 26; 28; 29]. The main difference is that the latter only focus on predicting the last query token while we need to predict each historical token. We note that practical transformers do learn similar token construction in the first softmax attention layer (e.g., see Fig. 4B in [16]).

**Next-token prediction.** Receiving the prompt $\boldsymbol{P}_t = (\boldsymbol{x}_1, \ldots, \boldsymbol{x}_t)$, the network's prediction for the next element $\boldsymbol{x}_{t+1}$ will be the first $d$ coordinates of the output $\boldsymbol{f}_t(\boldsymbol{E}_t; \boldsymbol{\theta})$, aligning with the setup adopted in [16; 17]. Namely, we have

$$\widehat{\boldsymbol{y}}_t(\boldsymbol{E}_t; \boldsymbol{\theta}) = [\boldsymbol{f}_t(\boldsymbol{E}_t; \boldsymbol{\theta})]_{1:d} \in \mathbb{C}^d.$$

Henceforth, we will omit the dependence on $\boldsymbol{E}_t$ and $\boldsymbol{\theta}$, and use $\widehat{\boldsymbol{y}}_t$ if it is not ambiguous. Since only the first $d$ rows are extracted from the output by the attention layer, the prediction $\widehat{\boldsymbol{y}}_t$ just depends on some parts of $\boldsymbol{W}^{PV}$ and $\boldsymbol{W}^{KQ}$. Concretely, we denote that

$$\boldsymbol{W}^{PV} = \begin{pmatrix} \boldsymbol{W}_{11}^{PV} & \boldsymbol{W}_{12}^{PV} & \boldsymbol{W}_{13}^{PV} \\ \boldsymbol{W}_{21}^{PV} & \boldsymbol{W}_{22}^{PV} & \boldsymbol{W}_{23}^{PV} \\ \boldsymbol{W}_{31}^{PV} & \boldsymbol{W}_{32}^{PV} & \boldsymbol{W}_{33}^{PV} \end{pmatrix}, \boldsymbol{W}^{KQ} = \begin{pmatrix} \boldsymbol{W}_{11}^{KQ} & \boldsymbol{W}_{12}^{KQ} & \boldsymbol{W}_{13}^{KQ} \\ \boldsymbol{W}_{21}^{KQ} & \boldsymbol{W}_{22}^{KQ} & \boldsymbol{W}_{23}^{KQ} \\ \boldsymbol{W}_{31}^{KQ} & \boldsymbol{W}_{32}^{KQ} & \boldsymbol{W}_{33}^{KQ} \end{pmatrix},$$

where $\boldsymbol{W}_{ij}^{PV}, \boldsymbol{W}_{ij}^{KQ} \in \mathbb{R}^{d \times d}$ for all $i, j \in [3]$. Then the $\widehat{\boldsymbol{y}}_t$ can be written as

$$\widehat{\boldsymbol{y}}_t = \begin{pmatrix} \boldsymbol{W}_{12}^{PV} & \boldsymbol{W}_{13}^{PV} \end{pmatrix} \frac{\boldsymbol{E}_t^{\boldsymbol{x}} \boldsymbol{E}_t^{\boldsymbol{x}*}}{\rho_t} \begin{pmatrix} \boldsymbol{W}_{22}^{KQ} & \boldsymbol{W}_{23}^{KQ} \\ \boldsymbol{W}_{32}^{KQ} & \boldsymbol{W}_{33}^{KQ} \end{pmatrix} \boldsymbol{e}_t^{\boldsymbol{x}}. \tag{1}$$

Here $\boldsymbol{E}_t^{\boldsymbol{x}} = (\boldsymbol{e}_1^{\boldsymbol{x}}, \ldots, \boldsymbol{e}_t^{\boldsymbol{x}}) \in \mathbb{C}^{2d \times t}$ denotes the last $2d$ rows of the $\boldsymbol{E}_t$, where $\boldsymbol{e}_i^{\boldsymbol{x}} = (\boldsymbol{x}_i^\top, \boldsymbol{x}_{i-1}^\top)^\top$. Therefore, we only need to analyze the selected parameters in Eq. 1 during the training dynamics. The derivation of Eq. 1 can be found in Appendix A.1.

### 3.3 Training procedure

**Loss function.** To train the transformer model over the next-token prediction task, we focus on minimizing the following population loss:

$$L(\boldsymbol{\theta}) = \sum_{t=2}^{T-1} L_t(\boldsymbol{\theta}) = \sum_{t=2}^{T-1} \mathbb{E}_{\boldsymbol{x}_1, \boldsymbol{W}} \left[\frac{1}{2} \|\widehat{\boldsymbol{y}}_t - \boldsymbol{x}_{t+1}\|_2^2\right], \tag{2}$$

where the expectation is taken with respect to the start point $\boldsymbol{x}_1$ and the transition matrix $\boldsymbol{W}$. Henceforth, we will suppress the subscripts of the expectation for simplicity. The population loss is a standard objective in the optimization studies [24; 47], and this objective has been used in recent works on AR modeling [16; 17]. The summation starts from $t = 2$ because we do not have any information to predict $\boldsymbol{x}_2$ given only $\boldsymbol{x}_1$.

**Initialization strategy.** We adopt the following diagonal initialization strategy, and similar settings have been used in recent works on ICL for linear problem [24; 30; 17].

**Assumption 3.1** (Diagonal initialization). *At the initial time $\tau = 0$, we assume that*

$$\boldsymbol{W}^{KQ}(0) = \begin{pmatrix} \mathbf{0}_{d\times d} & \mathbf{0}_{d\times d} & \mathbf{0}_{d\times d} \\ \mathbf{0}_{d\times d} & \mathbf{0}_{d\times d} & \mathbf{0}_{d\times d} \\ \mathbf{0}_{d\times d} & a_0\boldsymbol{I}_d & \mathbf{0}_{d\times d} \end{pmatrix}, \boldsymbol{W}^{PV}(0) = \begin{pmatrix} \mathbf{0}_{d\times d} & b_0\boldsymbol{I}_d & \mathbf{0}_{d\times d} \\ \mathbf{0}_{d\times d} & \mathbf{0}_{d\times d} & \mathbf{0}_{d\times d} \\ \mathbf{0}_{d\times d} & \mathbf{0}_{d\times d} & \mathbf{0}_{d\times d} \end{pmatrix},$$

*where the red submatrices are related to the $\widehat{y}_t$ and changed during the training process.*

The most related paper [17] considers a stronger diagonal structure than ours, and it only investigates the loss landscape. Our results deepened the understanding of AR transformers by considering practical training dynamics. We think this assumption might be inevitable for a tractable analysis and leave theory for standard (Gaussian) initialization to future work. In Section 6, we also conduct experiments under standard initialization, which further supports the rationality of Assumption 3.1.

**Optimization algorithm.** We utilize the gradient flow to minimize the learning objective in Eq. 2, which is equivalent to the gradient descent with infinitesimal step size and governed by the ordinary differential equation (ODE) $\frac{\mathrm{d}}{\mathrm{d}\tau}\boldsymbol{\theta} = -\nabla L(\boldsymbol{\theta})$.

# 4 Main results

In this section, we present the main theoretical results of this paper. First, in Section 4.1, we prove that when $\mathcal{D}_{\boldsymbol{x}_1}$ satisfies some certain condition (Assumption 4.1), the trained transformer implements one step of gradient descent for the minimization of an OLS problem, which validates the rationality of the mesa-optimization hypothesis [16]. Next, in Section 4.2, we further explore the capability limitation of the obtained mesa-optimizer under Assumption 4.1, where we characterize a stronger assumption (Assumption 4.2) as the necessary and sufficient condition that the mesa-optimizer recovers the true distribution. Finally, we go beyond Assumption 4.1, where the exploratory analysis proves that the trained transformer will generally not perform vanilla gradient descent for the OLS problem.

## 4.1 Trained transformer is a mesa-optimizer

In this subsection, we show that under a certain assumption of $\mathcal{D}_{\boldsymbol{x}_1}$, the trained one-layer linear transformer will converge to the mesa-optimizer [16; 17]. Namely, it will perform one step of gradient descent for the minimization of an OLS problem about the received prompt. The sufficient condition of the distribution $\mathcal{D}_{\boldsymbol{x}_1}$ can be summarized as follows.

**Assumption 4.1** (Sufficient condition for the emergence of mesa-optimizer). *We assume that the distribution $\mathcal{D}_{\boldsymbol{x}_1}$ of the initial token $\boldsymbol{x}_1 \in \mathbb{R}^d$ satisfies $\mathbb{E}_{\boldsymbol{x}_1 \sim \mathcal{D}_{\boldsymbol{x}_1}}[x_{1i_1}x_{1i_2}^{r_2}\cdots x_{1i_n}^{r_n}] = 0$ for any subset $\{i_1, \ldots, i_n \mid n \leq 4\}$ of $[d]$, and $r_2, \ldots r_n \in \mathbb{N}$. In addition, we assume that $\kappa_1 = \mathbb{E}[x_{1j}^4]$, $\kappa_2 = \mathbb{E}[x_{1j}^6]$ and $\kappa_3 = \sum_{r \neq j} \mathbb{E}[x_{1j}^2 x_{1r}^4]$ are finite constant for any $j \in [d]$.*

Finding Assumption 4.1 is non-trivial since we need to derive the training dynamics first. The key intuition of this assumption is to keep the gradient of the non-diagonal elements of $\boldsymbol{W}_{32}^{KQ}$ and $\boldsymbol{W}_{12}^{PV}$ as zero, thus they can keep diagonal structure during the training. We note that any random vectors $\boldsymbol{x}_1$ whose coordinates $x_{1i}$ are i.i.d. random variables with zero mean and finite moments of order 2, 4, and 6 satisfy this assumption. For example, it includes the normal distribution $\mathcal{N}(\mathbf{0}_d, \sigma^2\boldsymbol{I}_d)$, which is a common setting in the learning theory field [47; 48; 49; 50; 51]. Under this assumption, the final fixed point found by the gradient flow can be characterized as the following theorem.

**Theorem 4.1** (Convergence of the gradient flow, proof in Section 5). *Consider the gradient flow of the one-layer linear transformer (see Eq. 1) over the population AR pretraining loss (see Eq. 2).*

*Suppose the initialization satisfies Assumption 3.1, and the initial token's distribution $\mathcal{D}_{\boldsymbol{x}_1}$ satisfies Assumption 4.1, then the gradient flow converges to*

$$\begin{pmatrix} \widetilde{\boldsymbol{W}_{22}^{KQ}} & \widetilde{\boldsymbol{W}_{23}^{KQ}} \\ \boldsymbol{W}_{32}^{KQ} & \boldsymbol{W}_{33}^{KQ} \end{pmatrix} = \begin{pmatrix} \mathbf{0}_{d\times d} & \mathbf{0}_{d\times d} \\ \widetilde{a}\boldsymbol{I}_d & \mathbf{0}_{d\times d} \end{pmatrix}, \begin{pmatrix} \widetilde{\boldsymbol{W}_{12}^{PV}} & \widetilde{\boldsymbol{W}_{13}^{PV}} \end{pmatrix} = \begin{pmatrix} \widetilde{b}\boldsymbol{I}_d & \mathbf{0}_{d\times d} \end{pmatrix}.$$

*Though different initialization $(a_0, b_0)$ lead to different $(\widetilde{a}, \widetilde{b})$, the solutions' product $\widetilde{ab}$ satisfies*

$$\widetilde{ab} = \frac{\kappa_1}{\kappa_2 + \frac{\kappa_3}{T-2}\sum_{t=2}^{T-1} \frac{1}{t-1}}.$$

As far as we know, Theorem 4.1 is the first theoretical result for the training dynamics and the mesa-optimization hypothesis of autoregressive transformers. The technical challenge compared to existing ICL theory for regression [24] mainly has two parts. First, our data model breaks the independence between data at different times, which causes difficulty in decomposing and estimating the gradient terms. Second, we modify the embedding strategy (more dimensions), scale the attention model (much more parameters), and change the loss function (more terms) to perform the full AR pertaining. All these parts are not well studied in the literature and make the gradients more complicated.

Theorem 4.1 is also a non-trivial extension of recent work [17], which characterizes the global minima of the AR modeling loss when imposing the diagonal structure on all parameter matrices during the training and fixing $\boldsymbol{x}_1$ as $\mathbf{1}_d$. In comparison, Theorem 4.1 does not depend on the special structure, and further investigates when the mesa-optimizer emerges in practical non-convex optimization.

We highlight that the limiting solution found by the gradient flow shares the same structure with the careful construction in [16], though the pretraining loss is non-convex. Therefore, our result theoretically validates the rationality of the mesa-optimization hypothesis [16] in the AR pretraining setting, which can be formally presented as the following corollary.

**Corollary 4.1** (Trained transformer as a mesa-optimizer, proof in Appendix A.3)**.** *We suppose that the same precondition of Theorem 4.1 holds. When predicting the $(t+1)$-th token, the trained transformer obtains $\widehat{\boldsymbol{W}}$ by implementing one step of gradient descent for the OLS problem $L_{\text{OLS},t}(\boldsymbol{W}) = \frac{1}{2}\sum_{i=1}^{t-1} \|\boldsymbol{x}_{i+1} - \boldsymbol{W}\boldsymbol{x}_i\|^2$, starting from the initialization $\boldsymbol{W} = \mathbf{0}_{d\times d}$ with a step size $\frac{\widetilde{ab}}{t-1}$.*

## 4.2 Capability limitation of the mesa-optimizer

Built upon the findings in Theorem 4.1, a simple calculation (details in Appendix A.3) shows that the prediction of the obtained mesa-optimizer given a new test prompt of length $T_{te}$ is

$$\widehat{\boldsymbol{y}}_{T_{te}} = \boldsymbol{W}\left(\widetilde{ab}\frac{\sum_{i=1}^{T_{te}-1} \boldsymbol{x}_i\boldsymbol{x}_i^*}{T_{te}-1}\right)\boldsymbol{x}_{T_{te}}. \tag{3}$$

It is natural to ask the question: where is the capability limitation of the obtained mesa-optimizer, and what data distribution can the trained transformer learn? Therefore, in this subsection, we study under what assumption of the initial token's distribution $\mathcal{D}_{\boldsymbol{x}_1}$, the one step of gradient descent performed by the trained transformer can exactly recover the underlying data distribution. First, leveraging the result from Eq. 3, we present a negative result, which proves that not all $\mathcal{D}_{\boldsymbol{x}_1}$ satisfies Assumption 4.1 can be recovered by the trained linear transformer.

**Proposition 4.1** (AR process with normal distributed initial token can not be learned, proof in Appendix A.4)**.** *Let $\mathcal{D}_{\boldsymbol{x}_1}$ be the multivariate normal distribution $\mathcal{N}(\mathbf{0}_d, \sigma^2\boldsymbol{I}_d)$ with any $\sigma^2 > 0$, then the "simple" AR process can not be recovered by the trained transformer even in the ideal case with long training context. Formally, when the training sequence length $T_{tr}$ is large enough, for any test context length $T_{te}$ and dimension $j \in [d]$, the prediction from the trained transformer satisfies*

$$E_{x_1,W}\left[\frac{(\widehat{y}_{T_{te}})_j}{(Wx_{T_{te}})_j}\right] \to \frac{1}{5}.$$

*Therefore, the prediction $\widehat{\boldsymbol{y}}_{T_{te}}$ will not converges to the true next token $\boldsymbol{W}\boldsymbol{x}_{T_{te}}$.*

Proposition 4.1 suggests that ICL by AR pretraining [16; 17] is different from ICL by few-shot pretraining [24; 25; 26; 27; 28; 29; 30; 23], which attracts much more attention from the theoretical community. In the latter setting, recent works [24; 26] proves that one step of gradient descent implemented by the trained transformer can in-context learn the linear regression problem with input sampled from $\mathcal{N}(\mathbf{0}_d, \sigma^2 \mathbf{I}_d)$. However, in the AR learning setting, the trained linear transformer fails.

This negative result shows that one-step GD learned by the AR transformer can not recover the distribution, but this can be solved by more complex models. Even for more complex data ($\boldsymbol{W}$ is not diagonal), [16] has empirically verified that multi-layer linear attention can perform multi-step gradient descent to learn the data distribution. Therefore, to address the issue, future works are suggested to study more complex architecture such as softmax attention [30], multi-head attention [52; 53; 54], deeper attention layers [55; 16], transformer block [56; 57; 58], and so on. Future theory considering more complex AR transformers can adopt the same data model and token embeddings in this paper, and try to use a similar proof technique to derive the training dynamics.

Proposition 4.1 implies that if we want the trained transformer to recover the data distribution by performing one step of gradient descent, a stronger condition of $\mathcal{D}_{\boldsymbol{x}_1}$ is needed. Under Assumption 4.1, the following sufficient and necessary condition related to the moment of $\mathcal{D}_{\boldsymbol{x}_1}$ is derived from Eq. 3 by letting $\widehat{\boldsymbol{y}}_{T_{te}}$ converges to $\boldsymbol{W} \boldsymbol{x}_{T_{te}}$ when context length $T_{tr}$ and $T_{te}$ are large enough.

**Assumption 4.2** (Condition for success of mesa-optimizer). *Based on Assumption 4.1, we further suppose that $\frac{\kappa_1}{\kappa_2} \frac{\sum_{i=1}^{T_{te}-1} \boldsymbol{x}_i \boldsymbol{x}_i^*}{T_{te}-1} \boldsymbol{x}_{T_{te}} \to \boldsymbol{x}_{T_{te}}$ for any $\boldsymbol{x}_1$ and $\boldsymbol{W}$, when $T_{te}$ is large enough.*

Assumption 4.2 is strong and shows the poor capability of the trained one-layer linear transformer because common distribution (e.g. Gaussian distribution, Gamma distribution, Poisson distribution, etc) always fails to satisfy this condition. Besides, it is a sufficient and necessary condition for the mesa-optimizer to succeed when the distribution $\mathcal{D}_{\boldsymbol{x}_1}$ has satisfied Assumption 4.1, thus can not be improved in this case. We construct the following example that satisfies Assumption 4.2.

*Example* 4.1 (sparse vector). If the random vector $\boldsymbol{x}_1 \in R^d$ is uniformly sampled from the candidate set of size $2d$ $\{\pm(c, 0, \ldots, 0)^\top, \pm(0, c, \ldots, 0)^\top, \pm(0, \ldots, 0, c)^\top\}$ for any fixed $c \in \mathbb{R}$, then the distribution $\mathcal{D}_{\boldsymbol{x}_1}$ satisfies Assumption 4.2. The derivation can be found in Appendix A.5.

For completeness, we formally summarize the following distribution learning guarantee for the trained transformer under Assumption 3.1 and 4.1.

**Theorem 4.2** (Trained transformer succeed to learn the distribution satisfies Assumption 4.2, proof in Appendix A.6). *Suppose that Assumption 3.1 and 4.1 holds, then Assumption 4.2 is the sufficient and necessary condition for the trained transformer to learn the AR process. Formally, when the training sequence length $T_{tr}$ and test context length $T_{te}$ are large enough, the prediction from the trained transformer satisfies*

$$\widehat{\boldsymbol{y}}_{T_{te}} \to \boldsymbol{W} \boldsymbol{x}_{T_{te}}, \quad T_{tr}, T_{te} \to +\infty.$$

### 4.3 Go beyond the Assumption 4.1

The behavior of the gradient flow under Assumption 4.1 has been clearly understood in Theorem 4.1. The follow-up natural question is what solution will be found by the gradient flow when Assumption 4.1 does not hold. In this subsection, we conduct exploratory analyses by adopting the setting in [17], where the initial token $\boldsymbol{x}_1$ is fixed as $\mathbf{1}_d$.

First, sharing the similar but weaker assumption of [17], we impose $\boldsymbol{W}_{32}^{KQ}$ and $\boldsymbol{W}_{12}^{PV}$ to stay diagonal during training by masking the non-diagonal gradients, then the trained transformer will perform one step of gradient descent, as suggested by [17]. Formally, it can be written as follows.

**Theorem 4.3** (Trained transformer as mesa-optimizer with non-diagonal gradient masking, proof in Appendix A.7). *Suppose the initialization satisfies Assumption 3.1, the initial token is fixed as $\mathbf{1}_d$, and we clip non-diagonal gradients of $\boldsymbol{W}_{32}^{KQ}$ and $\boldsymbol{W}_{12}^{PV}$ during the training, then the gradient flow of the one-layer linear transformer over the population AR loss converges to the same structure as the result in Theorem 4.1, with*

$$\widetilde{ab} = \frac{1}{1 + \frac{d-1}{T-2} \sum_{t=2}^{T-1} \frac{1}{t-1}}.$$

*Therefore, the obtained transformer performs one step of gradient descent in this case.*

Theorem 4.3 can be seen as a complement and an extension of Proposition 2 in [17] from the perspective of optimization. We note that [17] assumes all the parameter matrices to be diagonal and only analyzes the global minima without considering the practical non-convex optimization process.

Next, we adopt some exploratory analyses for the gradient flow without additional non-diagonal gradient masking. The convergence result of the gradient flow can be asserted as the following proposition. The key intuition of its proof is that when the parameters matrices share the same structure as the result in Theorem 4.1, the non-zero gradients of the non-diagonal elements of $\boldsymbol{W}_{32}^{KQ}$ and $\boldsymbol{W}_{12}^{PV}$ will occur. In addition, we note the result does not depend on Assumption 3.1.

**Proposition 4.2** (Trained transformer does not perform on step of gradient descent, proof in Appendix A.8). *The limiting point found by the gradient does not share the same structure as that in Theorem 4.1, thus the trained transformer will not implement one step of vanilla gradient descent for minimizing the OLS problem $\frac{1}{2}\sum_{i=1}^{t-1}\|\boldsymbol{x}_{i+1} - W\boldsymbol{x}_i\|^2$.*

To fully solve the problem and find the limiting point of the gradient flow in this case (or more generally, any case beyond Assumption 3.1 and 4.1), one can not enjoy the diagonal structure of $\boldsymbol{W}_{32}^{KQ}$ and $\boldsymbol{W}_{12}^{PV}$ anymore. When $\boldsymbol{W}_{32}^{KQ}$ and $\boldsymbol{W}_{12}^{PV}$ are general dense matrices, computation of the gradient will be much more difficult than that in Proposition 4.2. Therefore, we leave the general rigorous result of convergence without Assumption 3.1 and 4.1 for future work.

We are aware that recent theoretical studies on ICL for linear regression have faced a similar problem. [24; 26; 23] find that when the input's distribution does not satisfy Assumption 4.1 (e.g., $\mathcal{N}(\boldsymbol{0}_d, \Sigma)$), the trained transformer will implement one step of preconditioned gradient descent on for some inner objective function. We conjecture similar results will hold in the case of in-context AR learning. We will empirically verify this conjecture when $\boldsymbol{x}_1$ is a full one vector, in Section 6.

# 5 Proof skeleton

In this section, we outline the proof ideas of Theorem 4.1, which is one of the core findings of this paper, and also a theoretical base of the more complex proofs of Theorem 4.3 and Proposition 4.2. The full proof of this Theorem is placed in Appendix A.2.

The first key step is to observe that each coordinate of prediction $\widehat{y}_t$ (Eq. 1) can be written as the output of a quadratic function, which will greatly simplify the follow-up gradient operation.

**Lemma 5.1** (Simplification of $\widehat{y}_{t,j}$, proof in Appendix A.2.1). *Each element of the network's prediction $\widehat{y}_{t,j}$ ($j \in [d]$) can be expressed as the following.*

$$\widehat{y}_{t,j} = \boldsymbol{B}_j^\top \cdot \boldsymbol{e}_t^{\boldsymbol{x}\top} \otimes \frac{\boldsymbol{E}_t^{\boldsymbol{x}}\boldsymbol{E}_t^{\boldsymbol{x}*}}{\rho_t} \cdot \mathrm{Vec}(\boldsymbol{A}) = \mathrm{Vec}^\top(\boldsymbol{A}) \cdot \boldsymbol{e}_t^{\boldsymbol{x}} \otimes \frac{\overline{\boldsymbol{E}_t^{\boldsymbol{x}}}\boldsymbol{E}_t^{\boldsymbol{x}\top}}{\rho_t} \cdot \boldsymbol{B}_j,$$

*where the $\boldsymbol{A}$ and $\boldsymbol{B}_j$ are defined as*

$$\boldsymbol{A} = \begin{pmatrix} \boldsymbol{a}_1 & \cdots & \boldsymbol{a}_{2d} \end{pmatrix} = \begin{pmatrix} \boldsymbol{W}_{22}^{KQ} & \boldsymbol{W}_{23}^{KQ} \\ \boldsymbol{W}_{32}^{KQ} & \boldsymbol{W}_{33}^{KQ} \end{pmatrix}, \quad \boldsymbol{B}_j = \begin{pmatrix} \boldsymbol{b}_{j1} \\ \boldsymbol{b}_{j2} \end{pmatrix} = \begin{pmatrix} \boldsymbol{W}_{12,j:}^{PV\top} \\ \boldsymbol{W}_{13,j:}^{PV\top} \end{pmatrix},$$

*with $\boldsymbol{a}_i \in \mathbb{R}^{2d}$ and $\boldsymbol{b}_{j1}, \boldsymbol{b}_{j2} \in \mathbb{R}^d$.*

Next, We calculate the gradient for the parameter matrices of the linear transformer and present the dynamical system result, which is the most complex part in the proof of Theorem 4.1.

**Lemma 5.2** (dynamical system of gradient flow under Assumption 4.1, proof in Appendix A.2.2). *Suppose that Assumption 4.1 holds, then the dynamical process of the parameters in the diagonal of $\boldsymbol{W}_{32}^{KQ}$ and $\boldsymbol{W}_{12}^{PV}$ satisfies*

$$\frac{\mathrm{d}}{\mathrm{d}\tau}a = -ab^2\left[(T-2)\kappa_2 + \sum_{t=2}^{T-1}\frac{1}{t-1}\kappa_3\right] + b(T-2)\kappa_1,$$

$$\frac{\mathrm{d}}{\mathrm{d}\tau}b = -a^2b\left[(T-2)\kappa_2 + \sum_{t=2}^{T-1}\frac{1}{t-1}\kappa_3\right] + a(T-2)\kappa_1,$$

*while the gradients for all other parameters were kept at zero during the training process.*

Similar ODEs have occurred in existing studies, such as the deep linear networks [59] and recent ICL for linear regression [24]. Notably, these dynamics are the same as those of gradient flow on a non-convex objective function with clear global minima, which is summarized as the following.

**Lemma 5.3** (Surrogate objective function, proof in Appendix A.2.3). *Suppose that Assumption 4.1 holds and denote $(T-2)\kappa_2 + \sum_{t=2}^{T-1} \frac{1}{t-1}\kappa_3$ and $(T-2)\kappa_1$ by $c_1$ and $c_2$, respectively. Then, the dynamics in Lemma 5.2 are the same as those of gradient flow on the following objective function:*

$$\widetilde{\ell}(a,b) = \frac{1}{2c_1}(c_2 - c_1 ab)^2,$$

*whose global minimums satisfy $ab = c_2/c_1$.*

Furthermore, We show that although the objective $\widetilde{\ell}(a,b)$ is non-convex, the Polyak-Łojasiewicz (PL) inequality [60; 61] holds, which implies that gradient flow converges to the global minimum.

**Lemma 5.4** (Global convergence of gradient flow, proof in Appendix A.2.4). *Suppose that Assumption 4.1 holds, then $\widetilde{\ell}(a,b)$ is a non-convex function and satisfies the PL inequality as follows.*

$$\left|\frac{\partial}{\partial a}\widetilde{\ell}(a,b)\right|^2 + \left|\frac{\partial}{\partial b}\widetilde{\ell}(a,b)\right|^2 \geq 2c_1(a^2 + b^2)\left(\widetilde{\ell}(a,b) - \min_{a,b}\widetilde{\ell}(a,b)\right).$$

*Therefore, the gradient flow in Lemma 5.2 converges to the global minimum of $\widetilde{\ell}(a,b)$.*

Finally, Theorem 4.1 can be proved by directly applying the above lemmas.

# 6 Simulation results

In this section, we conduct simulations to verify and generalize our theoretical results. In terms of the train set, we generate $10k$ sequences with $T_{tr} = 100$ and $d = 5$. In addition, we generate another test set with $10k$ sequences of the same shape. We train for 200 epochs with vanilla gradient descent, with different diagonal initialization of $(a_0, b_0)$ by $(0.1, 0.1)$, $(0.5, 1.5)$, $(2, 2)$. The detailed configurations (e.g., step size) and results of different experiments can be found in Appendix B.

**Initial token sampled from $\mathcal{N}(\mathbf{0}_d, \sigma^2 \mathbf{I}_d)$.** We conduct simulations with $\sigma = 0.5, 1, 2$ respectively. With any initialization of $(a_0, b_0)$, simulations show that $ab$ converges to $\kappa_1/\kappa_2 = 1/5\sigma^2$, and $\widehat{\mathbf{y}}_{T_{te}-1}$ converges to $\mathbf{x}_{T_{te}}/5$ in expectation, which verifies Theorem 4.1 and Proposition 4.1, respectively. In the main paper, we present the convergence results with $\sigma = 0.5$ in Fig. 1a and 1b. We also verify our theory in the small-context scenarios ($T_{tr} = 5$), which is placed in Fig. 4 in Appendix B.3.

**Initial token sampled from Example 4.1.** We conduct simulations with scale $c = 0.5, 1, 2$ respectively. With any initialization of $(a_0, b_0)$, simulations show that $ab$ converges to $\kappa_1/\kappa_2 = 1/c^2$ (see details in Appendix A.5), and $\widehat{\mathbf{y}}_{T_{te}-1}$ converges to the truth $\mathbf{x}_{T_{te}}$, which verifies Theorem 4.1 and Theorem 4.2, respectively. In the main paper, we present the results with $c = 0.5$ in Fig. 1c and 1d.

**Initial token fixed as $\mathbf{1}_d$.** We conduct experiment with $\mathbf{x}_1 = \mathbf{1}_d$. The results Fig. 7 in Appendix B.5 show that $\mathbf{W}_{32}^{KQ}$ and $\mathbf{W}_{12}^{PV}$ converge to dense matrices with strong diagonals, and other matrices converge to $\mathbf{0}_{d \times d}$, which means that the trained transformer performs somewhat preconditioned gradient descent. The detailed derivation is placed in Appendix B.5.

**Go beyond the diagonal initialization.** Finally, in order to extend our theory, we repeat experiments under Gaussian initialization with different variance ($\sigma_w = 0.001, 0.01, 0.1$). The results of Gaussian start points and sparse start points (Example 4.1) can be found in Fig. 5 of Appendix B.3 and Fig. 6 of Appendix B.4, respectively. As a result, though the convergence results of parameters are not the same as those under diagonal initialization, they keep the same diagonal structure, which can be understood as GD with adaptive learning rate in different dimensions. In addition, the test results (ratio or MSE loss) under the standard Gaussian initialization are the same as those under diagonal initialization, which further verifies the capability limitation of the trained transformers. To sum up, these experimental results demonstrate that our theoretical results have a certain representativeness, which further supports the rationality of the diagonal initialization.

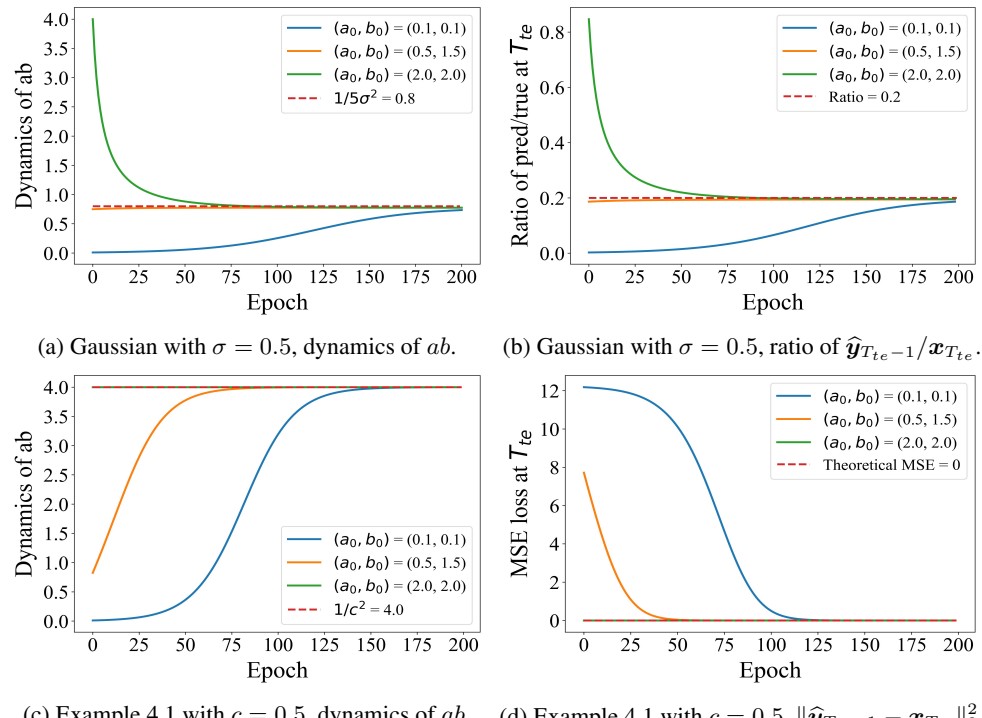

(a) Gaussian with $\sigma = 0.5$, dynamics of $ab$.  (b) Gaussian with $\sigma = 0.5$, ratio of $\widehat{\boldsymbol{y}}_{T_{te}-1}/\boldsymbol{x}_{T_{te}}$.

(c) Example 4.1 with $c = 0.5$, dynamics of $ab$.  (d) Example 4.1 with $c = 0.5$, $\|\widehat{\boldsymbol{y}}_{T_{te}-1} - \boldsymbol{x}_{T_{te}}\|_2^2$.

Figure 1: Simulations results on Gaussian and Example 4.1 show that the convergence of $ab$ satisfies Theorem 4.1. In addition, the trained transformer can recover the sequence with the initial token from Example 4.1, but fails to recover the Gaussian initial token, which verifies Theorem 4.2 and Proposition 4.1, respectively.

# 7 Conclusion and Discussion

In this paper, we towards understanding the the mechanisms underlying the ICL by analyzing the mesa-optimization hypothesis. To achieve this goal, we investigate the non-convex dynamics of a one-layer linear transformer autoregressively trained by gradient flow on a controllable AR process. First, we find a sufficient condition (Assumption 4.1) for the emergence of mesa-optimizer. Second, we explore the capability of the mesa-optimizer, where we find a sufficient and necessary condition (Assumption 4.2) that the trained transformer recovers the true distribution. Third, we analyze the case where Assumption 4.1 does not hold, and find that the trained transformer will not perform vanilla gradient descent in general. Finally, our simulation results verify the theoretical results.

**Limitations and social impact.** First, our theory only focuses on the one-layer linear transformer, thus whether the results hold when more complex models are adopted is still unclear. We believe that our analysis can give insight to those cases. Second, the general case where Assumption 3.1 and 4.1 does not hold is not fully addressed in this paper due to technical difficulties. Future work can consider that setting based on our theoretical and empirical findings. Finally, this is mainly theoretical work and we do not see a direct social impact of our theory.

# 8 Acknowledgement

This work was supported by Beijing Natural Science Foundation (L247030); NSF of China (Nos. 62076145, 62206159); Beijing Nova Program (No. 20230484416); Major Innovation & Planning Interdisciplinary Platform for the "Double-First Class" Initiative, Renmin University of China; the Fundamental Research Funds for the Central Universities, and the Research Funds of Renmin University of China (22XNKJ13); the Natural Science Foundation of Shandong Province (Nos. ZR2022QF117), the Fundamental Research Funds of Shandong University. The work was partially done at the Engineering Research Center of Next-Generation Intelligent Search and Recommendation, Ministry of Education. G. Wu was also sponsored by the TaiShan Scholars Program.

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

# Contents of Appendix

# Appendix A   Proofs

## A.1   Proof of eq. (1)

*Proof.* We first calculate the output by causal linear attention layer as the following:

$$\boldsymbol{f}_t(\boldsymbol{E}_t; \boldsymbol{\theta})$$

$$= \boldsymbol{e}_t + \frac{1}{\rho_t} \boldsymbol{W}^{PV} \boldsymbol{E}_t \boldsymbol{E}_t^* \boldsymbol{W}^{KQ} \boldsymbol{e}_t$$

$$= \boldsymbol{e}_t + \frac{1}{\rho_t} \boldsymbol{W}^{PV} \left( \sum_{i=1}^{t} \boldsymbol{e}_i \boldsymbol{e}_i^* \right) \boldsymbol{W}^{KQ} \boldsymbol{e}_t$$

$$= \boldsymbol{e}_t + \frac{1}{\rho_t} \sum_{i=1}^{t} \left( \boldsymbol{W}^{PV} \boldsymbol{e}_i \cdot \left( \boldsymbol{W}^{KQ\top} \boldsymbol{e}_i \right)^* \right) \boldsymbol{e}_t$$

$$= \boldsymbol{e}_t + \frac{1}{\rho_t} \sum_{i=1}^{t} \left( \begin{pmatrix} \boldsymbol{W}_{11}^{PV} & \boldsymbol{W}_{12}^{PV} & \boldsymbol{W}_{13}^{PV} \\ \boldsymbol{W}_{21}^{PV} & \boldsymbol{W}_{22}^{PV} & \boldsymbol{W}_{23}^{PV} \\ \boldsymbol{W}_{31}^{PV} & \boldsymbol{W}_{32}^{PV} & \boldsymbol{W}_{33}^{PV} \end{pmatrix} \boldsymbol{e}_i \cdot \left( \begin{pmatrix} \boldsymbol{W}_{11}^{KQ} & \boldsymbol{W}_{12}^{KQ} & \boldsymbol{W}_{13}^{KQ} \\ \boldsymbol{W}_{21}^{KQ} & \boldsymbol{W}_{22}^{KQ} & \boldsymbol{W}_{23}^{KQ} \\ \boldsymbol{W}_{31}^{KQ} & \boldsymbol{W}_{32}^{KQ} & \boldsymbol{W}_{33}^{KQ} \end{pmatrix}^\top \boldsymbol{e}_i \right)^* \right) \boldsymbol{e}_t$$

$$= \begin{pmatrix} \boldsymbol{0}_d \\ \boldsymbol{x}_t \\ \boldsymbol{x}_{t-1} \end{pmatrix} + \frac{1}{\rho_t} \sum_{i=1}^{t} \left( \begin{pmatrix} \boldsymbol{W}_{12}^{PV} \boldsymbol{x}_i + \boldsymbol{W}_{13}^{PV} \boldsymbol{x}_{i-1} \\ \times \\ \times \end{pmatrix} \cdot \begin{pmatrix} \times \\ \boldsymbol{W}_{22}^{KQ\top} \boldsymbol{x}_i + \boldsymbol{W}_{32}^{KQ\top} \boldsymbol{x}_{i-1} \\ \boldsymbol{W}_{23}^{KQ\top} \boldsymbol{x}_i + \boldsymbol{W}_{33}^{KQ\top} \boldsymbol{x}_{i-1} \end{pmatrix}^* \right) \begin{pmatrix} \boldsymbol{0}_d \\ \boldsymbol{x}_t \\ \boldsymbol{x}_{t-1} \end{pmatrix},$$

where $\times$s are the elements that will not contribute to the final $\widehat{\boldsymbol{y}}_t$. A further simple computation shows that

$$\widehat{\boldsymbol{y}}_t = \boldsymbol{0}_d + \frac{1}{\rho_t} \sum_{i=1}^{t} (\boldsymbol{W}_{12}^{PV} \boldsymbol{x}_i + \boldsymbol{W}_{13}^{PV} \boldsymbol{x}_{i-1})(\boldsymbol{W}_{22}^{KQ\top} \boldsymbol{x}_i + \boldsymbol{W}_{32}^{KQ\top} \boldsymbol{x}_{i-1})^* \boldsymbol{x}_t$$

$$+ \frac{1}{\rho_t} \sum_{i=1}^{t} (\boldsymbol{W}_{12}^{PV} \boldsymbol{x}_i + \boldsymbol{W}_{13}^{PV} \boldsymbol{x}_{i-1})(\boldsymbol{W}_{23}^{KQ\top} \boldsymbol{x}_i + \boldsymbol{W}_{33}^{KQ\top} \boldsymbol{x}_{i-1})^* \boldsymbol{x}_{t-1}$$

$$= \frac{1}{\rho_t} \sum_{i=1}^{t} \left( (\boldsymbol{W}_{12}^{PV} \boldsymbol{x}_i + \boldsymbol{W}_{13}^{PV} \boldsymbol{x}_{i-1}) \begin{pmatrix} \boldsymbol{W}_{22}^{KQ\top} \boldsymbol{x}_i + \boldsymbol{W}_{32}^{KQ\top} \boldsymbol{x}_{i-1} \\ \boldsymbol{W}_{23}^{KQ\top} \boldsymbol{x}_i + \boldsymbol{W}_{33}^{KQ\top} \boldsymbol{x}_{i-1} \end{pmatrix}^* \right) \begin{pmatrix} \boldsymbol{x}_t \\ \boldsymbol{x}_{t-1} \end{pmatrix}$$

$$= \frac{1}{\rho_t} \sum_{i=1}^{t} \left( \begin{pmatrix} \boldsymbol{W}_{12}^{PV} & \boldsymbol{W}_{13}^{PV} \end{pmatrix} \begin{pmatrix} \boldsymbol{x}_i \\ \boldsymbol{x}_{i-1} \end{pmatrix} \cdot \left( \begin{pmatrix} \boldsymbol{W}_{22}^{KQ} & \boldsymbol{W}_{23}^{KQ} \\ \boldsymbol{W}_{32}^{KQ} & \boldsymbol{W}_{33}^{KQ} \end{pmatrix}^\top \begin{pmatrix} \boldsymbol{x}_i \\ \boldsymbol{x}_{i-1} \end{pmatrix} \right)^* \right) \begin{pmatrix} \boldsymbol{x}_t \\ \boldsymbol{x}_{t-1} \end{pmatrix}$$

$$= \frac{1}{\rho_t} \sum_{i=1}^{t} \left( \begin{pmatrix} \boldsymbol{W}_{12}^{PV} & \boldsymbol{W}_{13}^{PV} \end{pmatrix} \boldsymbol{e}_i^{\boldsymbol{x}} \cdot \left( \begin{pmatrix} \boldsymbol{W}_{22}^{KQ} & \boldsymbol{W}_{23}^{KQ} \\ \boldsymbol{W}_{32}^{KQ} & \boldsymbol{W}_{33}^{KQ} \end{pmatrix}^\top \boldsymbol{e}_i^{\boldsymbol{x}} \right)^* \right) \boldsymbol{e}_t^{\boldsymbol{x}}$$

$$= \frac{1}{\rho_t} \begin{pmatrix} \boldsymbol{W}_{12}^{PV} & \boldsymbol{W}_{13}^{PV} \end{pmatrix} \left( \sum_{i=1}^{t} \boldsymbol{e}_i^{\boldsymbol{x}} \boldsymbol{e}_i^{\boldsymbol{x}*} \right) \begin{pmatrix} \boldsymbol{W}_{22}^{KQ} & \boldsymbol{W}_{23}^{KQ} \\ \boldsymbol{W}_{32}^{KQ} & \boldsymbol{W}_{33}^{KQ} \end{pmatrix} \boldsymbol{e}_t^{\boldsymbol{x}}$$

$$= \begin{pmatrix} \boldsymbol{W}_{12}^{PV} & \boldsymbol{W}_{13}^{PV} \end{pmatrix} \frac{\boldsymbol{E}_t^{\boldsymbol{x}} \boldsymbol{E}_t^{\boldsymbol{x}*}}{\rho_t} \begin{pmatrix} \boldsymbol{W}_{22}^{KQ} & \boldsymbol{W}_{23}^{KQ} \\ \boldsymbol{W}_{32}^{KQ} & \boldsymbol{W}_{33}^{KQ} \end{pmatrix} \boldsymbol{e}_t^{\boldsymbol{x}} \in \mathbb{C}^d,$$

which completes the proof. □

## A.2   Proof of Theorem 4.1

### A.2.1   Proof of Lemma 5.1

For the reader's convenience, we restate the lemma as the following.

**Lemma A.1.** *Each element of the network's prediction $\widehat{y}_{t,j}$ ($j \in [d]$) can be expressed as the following.*

$$\widehat{y}_{t,j} = \boldsymbol{B}_j^\top \cdot \boldsymbol{e}_t^{\boldsymbol{x}\top} \otimes \frac{\boldsymbol{E}_t^{\boldsymbol{x}} \boldsymbol{E}_t^{\boldsymbol{x}*}}{\rho_t} \cdot \mathrm{Vec}(\boldsymbol{A}) = \mathrm{Vec}^\top(\boldsymbol{A}) \cdot \boldsymbol{e}_t^{\boldsymbol{x}} \otimes \frac{\overline{\boldsymbol{E}_t^{\boldsymbol{x}} \boldsymbol{E}_t^{\boldsymbol{x}\top}}}{\rho_t} \cdot \boldsymbol{B}_j,$$

*where the $\boldsymbol{A}$ and $\boldsymbol{B}_j$ are defined as*

$$\boldsymbol{A} = \begin{pmatrix} \boldsymbol{a}_1 & \cdots & \boldsymbol{a}_{2d} \end{pmatrix} = \begin{pmatrix} \boldsymbol{W}_{22}^{KQ} & \boldsymbol{W}_{23}^{KQ} \\ \boldsymbol{W}_{32}^{KQ} & \boldsymbol{W}_{33}^{KQ} \end{pmatrix}, \quad \boldsymbol{B}_j = \begin{pmatrix} \boldsymbol{b}_{j1} \\ \boldsymbol{b}_{j2} \end{pmatrix} = \begin{pmatrix} \boldsymbol{W}_{12,j:}^{PV\top} \\ \boldsymbol{W}_{13,j:}^{PV\top} \end{pmatrix},$$

*with $\boldsymbol{a}_i \in \mathbb{R}^{2d}$ and $\boldsymbol{b}_{j1}, \boldsymbol{b}_{j2} \in \mathbb{R}^d$.*

*Proof.* Based on the result in Eq. 1, we can write

$$\widehat{y}_{t,j} = \begin{pmatrix} \boldsymbol{W}_{12}^{PV} & \boldsymbol{W}_{13}^{PV} \end{pmatrix}_{j:} \frac{\boldsymbol{E}_t^{\boldsymbol{x}} \boldsymbol{E}_t^{\boldsymbol{x}*}}{\rho_t} \begin{pmatrix} \boldsymbol{W}_{22}^{KQ} & \boldsymbol{W}_{23}^{KQ} \\ \boldsymbol{W}_{32}^{KQ} & \boldsymbol{W}_{33}^{KQ} \end{pmatrix} \boldsymbol{e}_t^{\boldsymbol{x}}$$

$$= \boldsymbol{B}_j^\top \frac{\boldsymbol{E}_t^{\boldsymbol{x}} \boldsymbol{E}_t^{\boldsymbol{x}*}}{\rho_t} \begin{pmatrix} \boldsymbol{a}_1 & \cdots & \boldsymbol{a}_{2d} \end{pmatrix} \boldsymbol{e}_t^{\boldsymbol{x}}$$

$$= \sum_{i=1}^{2d} e_{t,i}^{\boldsymbol{x}} \boldsymbol{B}_j^\top \frac{\boldsymbol{E}_t^{\boldsymbol{x}} \boldsymbol{E}_t^{\boldsymbol{x}*}}{\rho_t} \boldsymbol{a}_i$$

$$= \sum_{i=1}^{2d} e_{t,i}^{\boldsymbol{x}} \mathrm{tr}\left( \boldsymbol{B}_j^\top \frac{\boldsymbol{E}_t^{\boldsymbol{x}} \boldsymbol{E}_t^{\boldsymbol{x}*}}{\rho_t} \boldsymbol{a}_i \right)$$

$$= \sum_{i=1}^{2d} e_{t,i}^{\boldsymbol{x}} \mathrm{tr}\left( \boldsymbol{a}_i \boldsymbol{B}_j^\top \frac{\boldsymbol{E}_t^{\boldsymbol{x}} \boldsymbol{E}_t^{\boldsymbol{x}*}}{\rho_t} \right)$$

$$= \sum_{i=1}^{2d} \mathrm{tr}\left( \boldsymbol{a}_i \boldsymbol{B}_j^\top \cdot e_{t,i}^{\boldsymbol{x}} \frac{\boldsymbol{E}_t^{\boldsymbol{x}} \boldsymbol{E}_t^{\boldsymbol{x}*}}{\rho_t} \right)$$

$$= \mathrm{tr}\left[ \begin{pmatrix} \boldsymbol{a}_1 \boldsymbol{B}_j^\top \\ \vdots \\ \boldsymbol{a}_{2d} \boldsymbol{B}_j^\top \end{pmatrix} \cdot \begin{pmatrix} e_{t,1}^{\boldsymbol{x}} \frac{\boldsymbol{E}_t^{\boldsymbol{x}} \boldsymbol{E}_t^{\boldsymbol{x}*}}{\rho_t} & \cdots & e_{t,2d}^{\boldsymbol{x}} \frac{\boldsymbol{E}_t^{\boldsymbol{x}} \boldsymbol{E}_t^{\boldsymbol{x}*}}{\rho_t} \end{pmatrix} \right]$$

$$= \mathrm{tr}\left( \mathrm{Vec}(\boldsymbol{A}) \boldsymbol{B}_j^\top \cdot \boldsymbol{e}_t^{\boldsymbol{x}\top} \otimes \frac{\boldsymbol{E}_t^{\boldsymbol{x}} \boldsymbol{E}_t^{\boldsymbol{x}*}}{\rho_t} \right)$$

$$= \mathrm{tr}\left( \boldsymbol{B}_j^\top \cdot \boldsymbol{e}_t^{\boldsymbol{x}\top} \otimes \frac{\boldsymbol{E}_t^{\boldsymbol{x}} \boldsymbol{E}_t^{\boldsymbol{x}*}}{\rho_t} \cdot \mathrm{Vec}(\boldsymbol{A}) \right) = \boldsymbol{B}_j^\top \cdot \boldsymbol{e}_t^{\boldsymbol{x}\top} \otimes \frac{\boldsymbol{E}_t^{\boldsymbol{x}} \boldsymbol{E}_t^{\boldsymbol{x}*}}{\rho_t} \cdot \mathrm{Vec}(\boldsymbol{A})$$

$$= \mathrm{tr}\left( \mathrm{Vec}^\top(\boldsymbol{A}) \cdot \boldsymbol{e}_t^{\boldsymbol{x}} \otimes \frac{\overline{\boldsymbol{E}_t^{\boldsymbol{x}} \boldsymbol{E}_t^{\boldsymbol{x}\top}}}{\rho_t} \cdot \boldsymbol{B}_j \right) = \mathrm{Vec}^\top(\boldsymbol{A}) \cdot \boldsymbol{e}_t^{\boldsymbol{x}} \otimes \frac{\overline{\boldsymbol{E}_t^{\boldsymbol{x}} \boldsymbol{E}_t^{\boldsymbol{x}\top}}}{\rho_t} \cdot \boldsymbol{B}_j,$$

which finishes the proof. $\qquad\qquad\square$

### A.2.2 Proof of Lemma 5.2

For the reader's convenience, we restate the lemma as the following.

**Lemma A.2.** *Suppose that Assumption 4.1 holds, then the dynamical process of the parameters in the diagonal of $\boldsymbol{W}_{32}^{KQ}$ and $\boldsymbol{W}_{12}^{PV}$ satisfies*

$$\frac{\mathrm{d}}{\mathrm{d}\tau} a = -ab^2 \left[ (T-2)\kappa_2 + \sum_{t=2}^{T-1} \frac{1}{t-1} \kappa_3 \right] + b(T-2)\kappa_1,$$

$$\frac{\mathrm{d}}{\mathrm{d}\tau} b = -a^2 b \left[ (T-2)\kappa_2 + \sum_{t=2}^{T-1} \frac{1}{t-1} \kappa_3 \right] + a(T-2)\kappa_1,$$

*while the gradients for all other parameters were kept at zero during the training process.*

*Proof.* The population loss $L(\boldsymbol{\theta})$ in Eq. 2 can be rewritten as

$$L(\boldsymbol{\theta}) = \sum_{t=2}^{T-1} L_t(\boldsymbol{\theta}) = \sum_{t=2}^{T-1} \mathbb{E}\left[\frac{1}{2}\sum_{j=1}^{d}|\widehat{y}_{t,j} - x_{t+1,j}|^2\right] = \sum_{t=2}^{T-1}\sum_{j=1}^{d} \mathbb{E}\left[\frac{1}{2}|\widehat{y}_{t,j} - x_{t+1,j}|^2\right]$$

$$= \sum_{t=2}^{T-1}\sum_{j=1}^{d} \mathbb{E}\left[\frac{1}{2}\left(\widehat{y}_{t,j}^*\widehat{y}_{t,j} - 2\mathrm{Re}\left(x_{t+1,j}^*\widehat{y}_{t,j}\right) + x_{t+1,j}^* x_{t+1,j}\right)\right]$$

$$= \sum_{t=2}^{T-1}\sum_{j=1}^{d} \mathbb{E}\left[\frac{1}{2}\widehat{y}_{t,j}^*\widehat{y}_{t,j} - \mathrm{Re}\left(x_{t+1,j}^*\widehat{y}_{t,j}\right) + \frac{1}{2}x_{t+1,j}^* x_{t+1,j}\right].$$

Then, we can calculate the derivatives of $L_t(\theta)$ with respect to $\boldsymbol{B}_j$ and $\mathrm{Vec}(\boldsymbol{A})$ as

$$\nabla_{\boldsymbol{B}_j} L_t(\boldsymbol{\theta}) = \sum_{j=1}^{d} \mathbb{E}\left[\frac{1}{2}\nabla_{\boldsymbol{B}_j}\widehat{y}_{t,j}^*\widehat{y}_{t,j} - \nabla_{\boldsymbol{B}_j}\mathrm{Re}\left(x_{t+1,j}^*\widehat{y}_{t,j}\right)\right]$$

$$= \mathbb{E}\left[\frac{1}{2}\nabla_{\boldsymbol{B}_j}\widehat{y}_{t,j}^*\widehat{y}_{t,j} - \nabla_{\boldsymbol{B}_j}\mathrm{Re}\left(x_{t+1,j}^*\widehat{y}_{t,j}\right)\right]$$

$$= \frac{1}{2}\mathbb{E}\left[\nabla_{\boldsymbol{B}_j}\widehat{y}_{t,j}^*\widehat{y}_{t,j}\right] - \mathbb{E}\left[\nabla_{\boldsymbol{B}_j}\mathrm{Re}\left(x_{t+1,j}^*\widehat{y}_{t,j}\right)\right],$$

and

$$\nabla_{\mathrm{Vec}(\boldsymbol{A})} L_t(\boldsymbol{\theta}) = \sum_{j=1}^{d} \mathbb{E}\left[\frac{1}{2}\nabla_{\mathrm{Vec}(\boldsymbol{A})}\widehat{y}_{t,j}^*\widehat{y}_{t,j} - \nabla_{\mathrm{Vec}(\boldsymbol{A})}\mathrm{Re}\left(x_{t+1,j}^*\widehat{y}_{t,j}\right)\right]$$

$$= \frac{1}{2}\sum_{j=1}^{d} \mathbb{E}\left[\nabla_{\mathrm{Vec}(\boldsymbol{A})}\widehat{y}_{t,j}^*\widehat{y}_{t,j}\right] - \sum_{j=1}^{d} \mathbb{E}\left[\nabla_{\mathrm{Vec}(\boldsymbol{A})}\mathrm{Re}\left(x_{t+1,j}^*\widehat{y}_{t,j}\right)\right].$$

**Step one: calculate** $\mathbb{E}\left[\nabla_{\boldsymbol{B}_j}\mathrm{Re}\left(x_{t+1,j}^*\widehat{y}_{t,j}\right)\right]$. Based on Lemma 5.1, we have

$$\widehat{y}_{t,j} = \boldsymbol{B}_j^\top \cdot \boldsymbol{e}_t^{\boldsymbol{x}\top} \otimes \frac{\boldsymbol{E}_t^{\boldsymbol{x}}\boldsymbol{E}_t^{\boldsymbol{x}*}}{\rho_t} \cdot \mathrm{Vec}(\boldsymbol{A}).$$

Then, the $\mathbb{E}\left[\nabla_{\boldsymbol{B}_j}\mathrm{Re}\left(x_{t+1,j}^*\widehat{y}_{t,j}\right)\right]$ can be derived as the following.

$$\mathbb{E}\left[\nabla_{\boldsymbol{B}_j}\mathrm{Re}\left(x_{t+1,j}^*\widehat{y}_{t,j}\right)\right] = \mathbb{E}\left[\nabla_{\boldsymbol{B}_j}\mathrm{Re}\left(x_{t+1,j}^*\boldsymbol{B}_j^\top \cdot \boldsymbol{e}_t^{\boldsymbol{x}\top} \otimes \frac{\boldsymbol{E}_t^{\boldsymbol{x}}\boldsymbol{E}_t^{\boldsymbol{x}*}}{\rho_t} \cdot \mathrm{Vec}(\boldsymbol{A})\right)\right]$$

$$= \mathbb{E}\left[\nabla_{\boldsymbol{B}_j}\boldsymbol{B}_j^\top \cdot \mathrm{Re}\left(x_{t+1,j}^* \cdot \boldsymbol{e}_t^{\boldsymbol{x}\top} \otimes \frac{\boldsymbol{E}_t^{\boldsymbol{x}}\boldsymbol{E}_t^{\boldsymbol{x}*}}{\rho_t} \cdot \mathrm{Vec}(\boldsymbol{A})\right)\right]$$

$$= \mathbb{E}\left[\mathrm{Re}\left(x_{t+1,j}^* \cdot \boldsymbol{e}_t^{\boldsymbol{x}\top} \otimes \frac{\boldsymbol{E}_t^{\boldsymbol{x}}\boldsymbol{E}_t^{\boldsymbol{x}*}}{\rho_t} \cdot \mathrm{Vec}(\boldsymbol{A})\right)\right]$$

$$= \mathrm{Re}\left(\mathbb{E}\left[x_{t+1,j}^* \cdot \boldsymbol{e}_t^{\boldsymbol{x}\top} \otimes \frac{\boldsymbol{E}_t^{\boldsymbol{x}}\boldsymbol{E}_t^{\boldsymbol{x}*}}{\rho_t} \cdot \mathrm{Vec}(\boldsymbol{A})\right]\right)$$

$$= \mathrm{Re}\left(\mathbb{E}\left[\lambda_j^{-t}x_{1j} \cdot \mathrm{Vec}(\frac{\boldsymbol{E}_t^{\boldsymbol{x}}\boldsymbol{E}_t^{\boldsymbol{x}*}}{\rho_t}\boldsymbol{A}\boldsymbol{e}_t^{\boldsymbol{x}})\right]\right) \qquad \text{(use generating process)}$$

$$= \mathrm{Re}\left(\mathbb{E}\left[\lambda_j^{-t}x_{1j} \cdot (\frac{\boldsymbol{E}_t^{\boldsymbol{x}}\boldsymbol{E}_t^{\boldsymbol{x}*}}{\rho_t}\boldsymbol{A}\boldsymbol{e}_t^{\boldsymbol{x}})\right]\right),$$

where the penultimate equality uses the property of Kronecker and Vec operator $\text{Vec}(\boldsymbol{AXB}) = (\boldsymbol{B}^\top \otimes \boldsymbol{A})\text{Vec}(\boldsymbol{X})$, we refer Section 10.2 in [62] for details.

For $\frac{\boldsymbol{E}_t^{\boldsymbol{x}} \boldsymbol{E}_t^{\boldsymbol{x}*}}{\rho_t}$, we can simplify it as

$$\frac{\boldsymbol{E}_t^{\boldsymbol{x}} \boldsymbol{E}_t^{\boldsymbol{x}*}}{\rho_t} = \frac{1}{\rho_t} \sum_{i=1}^{t} \boldsymbol{e}_i^{\boldsymbol{x}} \boldsymbol{e}_i^{\boldsymbol{x}*}$$

$$= \frac{1}{\rho_t} \sum_{i=1}^{t} \begin{pmatrix} \boldsymbol{x}_i \boldsymbol{x}_i^* & \boldsymbol{x}_i \boldsymbol{x}_{i-1}^* \\ \boldsymbol{x}_{i-1} \boldsymbol{x}_i^* & \boldsymbol{x}_{i-1} \boldsymbol{x}_{i-1}^* \end{pmatrix}$$

$$= \frac{1}{\rho_t} \begin{pmatrix} \sum_{i=1}^{t} \boldsymbol{x}_i \boldsymbol{x}_i^* & \boldsymbol{W} \sum_{i=1}^{t-1} \boldsymbol{x}_i \boldsymbol{x}_i^* \\ \sum_{i=1}^{t-1} \boldsymbol{x}_i \boldsymbol{x}_i^* \boldsymbol{W}^* & \sum_{i=1}^{t-1} \boldsymbol{x}_i \boldsymbol{x}_i^* \end{pmatrix}.$$

Based on the diagonal property of $\boldsymbol{W}$, we can simplify the $\boldsymbol{x}_i \boldsymbol{x}_i^*$ as the following.

$$\boldsymbol{x}_i \boldsymbol{x}_i^* = (\boldsymbol{W}^{i-1} \boldsymbol{x}_1)(\boldsymbol{W}^{i-1} \boldsymbol{x}_1)^* = \boldsymbol{M}_i \odot \widehat{\Sigma},$$

where we define $\widehat{\Sigma} = \boldsymbol{x}_1 \boldsymbol{x}_1^*$ and $\boldsymbol{M}_i = \boldsymbol{\lambda}^{i-1} \boldsymbol{\lambda}^{i-1*}$. Therefore, we have

$$\frac{\boldsymbol{E}_t^{\boldsymbol{x}} \boldsymbol{E}_t^{\boldsymbol{x}*}}{\rho_t} = \frac{1}{\rho_t} \begin{pmatrix} \sum_{i=1}^{t} \boldsymbol{M}_i \odot \widehat{\Sigma} & \boldsymbol{W} \sum_{i=1}^{t-1} \boldsymbol{M}_i \odot \widehat{\Sigma} \\ \sum_{i=1}^{t-1} \boldsymbol{M}_i \odot \widehat{\Sigma} \boldsymbol{W}^* & \sum_{i=1}^{t-1} \boldsymbol{M}_i \odot \widehat{\Sigma} \end{pmatrix}. \tag{4}$$

Then, leveraging the sparse property of $\boldsymbol{A}$, we can derive $\frac{\boldsymbol{E}_t^{\boldsymbol{x}} \boldsymbol{E}_t^{\boldsymbol{x}*}}{\rho_t} \boldsymbol{A} \boldsymbol{e}_t^{\boldsymbol{x}}$ as follows.

$$\frac{\boldsymbol{E}_t^{\boldsymbol{x}} \boldsymbol{E}_t^{\boldsymbol{x}*}}{\rho_t} \boldsymbol{A} \boldsymbol{e}_t^{\boldsymbol{x}} = \frac{\boldsymbol{E}_t^{\boldsymbol{x}} \boldsymbol{E}_t^{\boldsymbol{x}*}}{\rho_t} \begin{pmatrix} \boldsymbol{0}_d \\ a \boldsymbol{x}_t \end{pmatrix}$$

$$= \frac{a}{\rho_t} \begin{pmatrix} \boldsymbol{W}(\sum_{i=1}^{t-1} \boldsymbol{M}_i \odot \widehat{\Sigma}) \boldsymbol{x}_t \\ (\sum_{i=1}^{t-1} \boldsymbol{M}_i \odot \widehat{\Sigma}) \boldsymbol{x}_t \end{pmatrix}.$$

Therefore, for any $l \in [d]$, we have

$$\left( \frac{\boldsymbol{E}_t^{\boldsymbol{x}} \boldsymbol{E}_t^{\boldsymbol{x}*}}{\rho_t} \boldsymbol{A} \boldsymbol{e}_t^{\boldsymbol{x}} \right)_l = \frac{a}{\rho_t} \lambda_l \sum_{i=1}^{t-1} (\boldsymbol{M}_i \odot \widehat{\Sigma})_{l:} \boldsymbol{x}_t$$

$$= \frac{a}{\rho_t} \lambda_l \sum_{i=1}^{t-1} \begin{pmatrix} \lambda_l^{i-1} \lambda_1^{1-i} x_{1l} x_{11} & \cdots & \lambda_l^{i-1} \lambda_d^{1-i} x_{1l} x_{1d} \end{pmatrix} \begin{pmatrix} \lambda_1^{t-1} x_{11} \\ \cdots \\ \lambda_d^{t-1} x_{1d} \end{pmatrix}$$

$$= \frac{a}{\rho_t} \lambda_l \sum_{i=1}^{t-1} \sum_{r=1}^{d} \lambda_l^{i-1} \lambda_r^{t-i} x_{1l} x_{1r}^2$$

$$= \frac{a}{\rho_t} \sum_{i=1}^{t-1} \sum_{r=1}^{d} \lambda_l^{i} \lambda_r^{t-i} x_{1l} x_{1r}^2.$$

Similarly, for any $l \in [2d] - [d]$, we have

$$\left( \frac{\boldsymbol{E}_t^{\boldsymbol{x}} \boldsymbol{E}_t^{\boldsymbol{x}*}}{\rho_t} \boldsymbol{A} \boldsymbol{e}_t^{\boldsymbol{x}} \right)_l = \frac{a}{\rho_t} \sum_{i=1}^{t-1} \sum_{r=1}^{d} \lambda_l^{i-1} \lambda_r^{t-i} x_{1l} x_{1r}^2.$$

To sum up, for any $l \in [2d]$, we have

$$\left( \frac{\boldsymbol{E}_t^{\boldsymbol{x}} \boldsymbol{E}_t^{\boldsymbol{x}*}}{\rho_t} \boldsymbol{A} \boldsymbol{e}_t^{\boldsymbol{x}} \right)_l = \begin{cases} \frac{a}{\rho_t} \sum_{i=1}^{t-1} \sum_{r=1}^{d} \lambda_l^{i} \lambda_r^{t-i} x_{1l} x_{1r}^2, & l \in [d], \\ \frac{a}{\rho_t} \sum_{i=1}^{t-1} \sum_{r=1}^{d} \lambda_{l-d}^{i-1} \lambda_r^{t-i} x_{1,l-d} x_{1r}^2, & l \in [2d] - [d]. \end{cases}$$

Next, we calculate the $\mathbb{E}\left[\lambda_j^{-t} x_{1j} \cdot \left(\frac{\boldsymbol{E}_t^{\boldsymbol{x}} \boldsymbol{E}_t^{\boldsymbol{x}*}}{\rho_t} \boldsymbol{A} \boldsymbol{e}_t^{\boldsymbol{x}}\right)\right]$. For any $l \in [d]$, we have

$$\mathbb{E}\left[\lambda_j^{-t} x_{1j} \cdot \left(\frac{\boldsymbol{E}_t^{\boldsymbol{x}} \boldsymbol{E}_t^{\boldsymbol{x}*}}{\rho_t} \boldsymbol{A} \boldsymbol{e}_t^{\boldsymbol{x}}\right)_l\right] = \mathbb{E}\left[\lambda_j^{-t} x_{1j} \cdot \frac{a}{\rho_t} \sum_{i=1}^{t-1} \sum_{r=1}^{d} \lambda_l^i \lambda_r^{t-i} x_{1l} x_{1r}^2\right]$$

$$= \frac{a}{\rho_t} \sum_{i=1}^{t-1} \sum_{r=1}^{d} \mathbb{E}[\lambda_j^{-t} \lambda_l^i \lambda_r^{t-i}] \mathbb{E}[x_{1j} x_{1l} x_{1r}^2].$$

We discuss them in the following categories,

1. $l \neq j$. In this case, $\mathbb{E}[x_{1j} x_{1l} x_{1r}^2] = 0$ by Assumption 4.1, thus

$$\mathbb{E}\left[\lambda_j^{-t} x_{1j} \cdot \left(\frac{\boldsymbol{E}_t^{\boldsymbol{x}} \boldsymbol{E}_t^{\boldsymbol{x}*}}{\rho_t} \boldsymbol{A} \boldsymbol{e}_t^{\boldsymbol{x}}\right)_l\right] = \frac{a}{\rho_t} \sum_{i=1}^{t-1} \sum_{r=1}^{d} \mathbb{E}[\lambda_j^{-t} \lambda_l^i \lambda_r^{t-i}] 0 = 0.$$

2. $l = j, r = j$. Because $\lambda_i$s are i.i.d. and $\mathbb{E}[\lambda_i]^k = \delta(k = 0)$, we have

$$\mathbb{E}\left[\lambda_j^{-t} x_{1j} \cdot \left(\frac{\boldsymbol{E}_t^{\boldsymbol{x}} \boldsymbol{E}_t^{\boldsymbol{x}*}}{\rho_t} \boldsymbol{A} \boldsymbol{e}_t^{\boldsymbol{x}}\right)_l\right] = \frac{a}{\rho_t} \sum_{r=1}^{d} \mathbb{E}[\lambda_j^{-t} \lambda_j^i \lambda_j^{t-i}] \mathbb{E}[x_{1j} x_{1j} x_{1j}^2]$$

$$= \frac{a}{\rho_t} (t-1) \mathbb{E}[x_{1j}^4].$$

Similarly, for any $l \in [2d] - [d]$, we have

$$\mathbb{E}\left[\lambda_j^{-t} x_{1j} \cdot \left(\frac{\boldsymbol{E}_t^{\boldsymbol{x}} \boldsymbol{E}_t^{\boldsymbol{x}*}}{\rho_t} \boldsymbol{A} \boldsymbol{e}_t^{\boldsymbol{x}}\right)_l\right] = 0.$$

Therefore, for any $l \in [2d]$, we have

$$\mathbb{E}\left[\lambda_j^{-t} x_{1j} \cdot \left(\frac{\boldsymbol{E}_t^{\boldsymbol{x}} \boldsymbol{E}_t^{\boldsymbol{x}*}}{\rho_t} \boldsymbol{A} \boldsymbol{e}_t^{\boldsymbol{x}}\right)\right] = \begin{cases} \frac{a}{\rho_t}(t-1)\mathbb{E}[x_{1j}^4], & l = j, \\ 0, & l \neq j, \end{cases}$$

and the $l$-th element of $\mathbb{E}\left[\nabla_{\boldsymbol{B}_j} \mathrm{Re}\left(x_{t+1,j}^* \widehat{y}_{t,j}\right)\right]$ is

$$\mathbb{E}\left[\nabla_{\boldsymbol{B}_j} \mathrm{Re}\left(x_{t+1,j}^* \widehat{y}_{t,j}\right)\right]_l = \mathrm{Re}\left(\mathbb{E}\left[\lambda_j^{-t} x_{1j} \cdot \left(\frac{\boldsymbol{E}_t^{\boldsymbol{x}} \boldsymbol{E}_t^{\boldsymbol{x}*}}{\rho_t} \boldsymbol{A} \boldsymbol{e}_t^{\boldsymbol{x}}\right)\right]\right)_l$$

$$= \begin{cases} \frac{a}{\rho_t}(t-1)\mathbb{E}[x_{1j}^4], & l = j, \\ 0, & l \neq j. \end{cases}$$

**Step two: calculate $\mathbb{E}\left[\nabla_{\boldsymbol{B}_j} \widehat{y}_{t,j}^* \widehat{y}_{t,j}\right]$.** Based on Lemma 5.1, we have

$$\widehat{y}_{t,j} = \mathrm{Vec}^\top(\boldsymbol{A}) \cdot \boldsymbol{e}_t^{\boldsymbol{x}} \otimes \frac{\overline{\boldsymbol{E}_t^{\boldsymbol{x}} \boldsymbol{E}_t^{\boldsymbol{x}\top}}}{\rho_t} \cdot \boldsymbol{B}_j,$$

then we can simplify the $\mathbb{E}\left[\nabla_{\boldsymbol{B}_j} \widehat{y}_{t,j}^* \widehat{y}_{t,j}\right]$ as follows.

$$\mathbb{E}\left[\nabla_{\boldsymbol{B}_j} \widehat{y}_{t,j}^* \widehat{y}_{t,j}\right]$$

$$= \mathbb{E}\left[\nabla_{\boldsymbol{B}_j} \left(\mathrm{Vec}^\top(\boldsymbol{A}) \cdot \boldsymbol{e}_t^{\boldsymbol{x}} \otimes \frac{\overline{\boldsymbol{E}_t^{\boldsymbol{x}} \boldsymbol{E}_t^{\boldsymbol{x}\top}}}{\rho_t} \cdot \boldsymbol{B}_j\right)^* \mathrm{Vec}^\top(\boldsymbol{A}) \cdot \boldsymbol{e}_t^{\boldsymbol{x}} \otimes \frac{\overline{\boldsymbol{E}_t^{\boldsymbol{x}} \boldsymbol{E}_t^{\boldsymbol{x}\top}}}{\rho_t} \cdot \boldsymbol{B}_j\right]$$

$$= \mathbb{E}\left[\nabla_{\boldsymbol{B}_j} \boldsymbol{B}_j^\top \cdot \boldsymbol{e}_t^{\boldsymbol{x}*} \otimes \frac{\overline{\boldsymbol{E}_t^{\boldsymbol{x}} \boldsymbol{E}_t^{\boldsymbol{x}\top}}}{\rho_t} \text{Vec}(\boldsymbol{A}) \cdot \text{Vec}^\top(\boldsymbol{A}) \boldsymbol{e}_t^{\boldsymbol{x}} \otimes \frac{\overline{\boldsymbol{E}_t^{\boldsymbol{x}} \boldsymbol{E}_t^{\boldsymbol{x}\top}}}{\rho_t} \cdot \boldsymbol{B}_j\right]$$

$$= \mathbb{E}\left[\boldsymbol{e}_t^{\boldsymbol{x}*} \otimes \frac{\overline{\boldsymbol{E}_t^{\boldsymbol{x}} \boldsymbol{E}_t^{\boldsymbol{x}\top}}}{\rho_t} \text{Vec}(\boldsymbol{A}) \cdot \text{Vec}^\top(\boldsymbol{A}) \boldsymbol{e}_t^{\boldsymbol{x}} \otimes \frac{\overline{\boldsymbol{E}_t^{\boldsymbol{x}} \boldsymbol{E}_t^{\boldsymbol{x}\top}}}{\rho_t} \cdot \boldsymbol{B}_j\right]$$

$$+ \mathbb{E}\left[\boldsymbol{e}_t^{\boldsymbol{x}\top} \otimes \frac{\boldsymbol{E}_t^{\boldsymbol{x}} \boldsymbol{E}_t^{\boldsymbol{x}*}}{\rho_t} \text{Vec}(\boldsymbol{A}) \cdot \text{Vec}^\top(\boldsymbol{A})\overline{\boldsymbol{e}_t^{\boldsymbol{x}}} \otimes \frac{\boldsymbol{E}_t^{\boldsymbol{x}} \boldsymbol{E}_t^{\boldsymbol{x}*}}{\rho_t} \cdot \boldsymbol{B}_j\right]$$

$$= \mathbb{E}\left[2\text{Re}\left(\boldsymbol{e}_t^{\boldsymbol{x}\top} \otimes \frac{\boldsymbol{E}_t^{\boldsymbol{x}} \boldsymbol{E}_t^{\boldsymbol{x}*}}{\rho_t} \text{Vec}(\boldsymbol{A}) \cdot \text{Vec}^\top(\boldsymbol{A})\overline{\boldsymbol{e}_t^{\boldsymbol{x}}} \otimes \frac{\boldsymbol{E}_t^{\boldsymbol{x}} \boldsymbol{E}_t^{\boldsymbol{x}*}}{\rho_t} \cdot \boldsymbol{B}_j\right)\right]$$

$$= 2\text{Re}\left(\mathbb{E}\left[\boldsymbol{e}_t^{\boldsymbol{x}\top} \otimes \frac{\boldsymbol{E}_t^{\boldsymbol{x}} \boldsymbol{E}_t^{\boldsymbol{x}*}}{\rho_t} \text{Vec}(\boldsymbol{A}) \cdot \text{Vec}^\top(\boldsymbol{A})\overline{\boldsymbol{e}_t^{\boldsymbol{x}}} \otimes \frac{\boldsymbol{E}_t^{\boldsymbol{x}} \boldsymbol{E}_t^{\boldsymbol{x}*}}{\rho_t} \cdot \boldsymbol{B}_j\right]\right).$$

We further derive that

$$\mathbb{E}\left[\boldsymbol{e}_t^{\boldsymbol{x}\top} \otimes \frac{\boldsymbol{E}_t^{\boldsymbol{x}} \boldsymbol{E}_t^{\boldsymbol{x}*}}{\rho_t} \text{Vec}(\boldsymbol{A}) \cdot \underbrace{\text{Vec}^\top(\boldsymbol{A})\overline{\boldsymbol{e}_t^{\boldsymbol{x}}} \otimes \frac{\boldsymbol{E}_t^{\boldsymbol{x}} \boldsymbol{E}_t^{\boldsymbol{x}*}}{\rho_t} \cdot \boldsymbol{B}_j}_{\in \mathbb{C}}\right]$$

$$= \mathbb{E}\left[\text{Vec}^\top(\boldsymbol{A})\overline{\boldsymbol{e}_t^{\boldsymbol{x}}} \otimes \frac{\boldsymbol{E}_t^{\boldsymbol{x}} \boldsymbol{E}_t^{\boldsymbol{x}*}}{\rho_t} \cdot \boldsymbol{B}_j \cdot \boldsymbol{e}_t^{\boldsymbol{x}\top} \otimes \frac{\boldsymbol{E}_t^{\boldsymbol{x}} \boldsymbol{E}_t^{\boldsymbol{x}*}}{\rho_t} \text{Vec}(\boldsymbol{A})\right]$$

$$= \mathbb{E}\left[\boldsymbol{B}_j^\top \cdot \boldsymbol{e}_t^{\boldsymbol{x}*} \otimes \frac{\overline{\boldsymbol{E}_t^{\boldsymbol{x}} \boldsymbol{E}_t^{\boldsymbol{x}\top}}}{\rho_t} \text{Vec}(\boldsymbol{A}) \cdot \boldsymbol{e}_t^{\boldsymbol{x}\top} \otimes \frac{\boldsymbol{E}_t^{\boldsymbol{x}} \boldsymbol{E}_t^{\boldsymbol{x}*}}{\rho_t} \text{Vec}(\boldsymbol{A})\right]$$

$$= \mathbb{E}\left[\boldsymbol{B}_j^\top \cdot \text{Vec}\left(\frac{\overline{\boldsymbol{E}_t^{\boldsymbol{x}} \boldsymbol{E}_t^{\boldsymbol{x}\top}}}{\rho_t} \boldsymbol{A}\overline{\boldsymbol{e}_t^{\boldsymbol{x}}}\right) \cdot \text{Vec}\left(\frac{\boldsymbol{E}_t^{\boldsymbol{x}} \boldsymbol{E}_t^{\boldsymbol{x}*}}{\rho_t} \boldsymbol{A}\boldsymbol{e}_t^{\boldsymbol{x}}\right)\right]$$

$$= \mathbb{E}\left[\boldsymbol{B}_j^\top \cdot \frac{\overline{\boldsymbol{E}_t^{\boldsymbol{x}} \boldsymbol{E}_t^{\boldsymbol{x}\top}}}{\rho_t} \boldsymbol{A}\overline{\boldsymbol{e}_t^{\boldsymbol{x}}} \cdot \frac{\boldsymbol{E}_t^{\boldsymbol{x}} \boldsymbol{E}_t^{\boldsymbol{x}*}}{\rho_t} \boldsymbol{A}\boldsymbol{e}_t^{\boldsymbol{x}}\right]$$

$$= \mathbb{E}\left[\sum_{k=1}^{2d} B_{jk}\left(\frac{\overline{\boldsymbol{E}_t^{\boldsymbol{x}} \boldsymbol{E}_t^{\boldsymbol{x}\top}}}{\rho_t} \boldsymbol{A}\overline{\boldsymbol{e}_t^{\boldsymbol{x}}}\right)_k \cdot \frac{\boldsymbol{E}_t^{\boldsymbol{x}} \boldsymbol{E}_t^{\boldsymbol{x}*}}{\rho_t} \boldsymbol{A}\boldsymbol{e}_t^{\boldsymbol{x}}\right]$$

$$= \mathbb{E}\left[b\left(\frac{\overline{\boldsymbol{E}_t^{\boldsymbol{x}} \boldsymbol{E}_t^{\boldsymbol{x}\top}}}{\rho_t} \boldsymbol{A}\overline{\boldsymbol{e}_t^{\boldsymbol{x}}}\right)_j \cdot \frac{\boldsymbol{E}_t^{\boldsymbol{x}} \boldsymbol{E}_t^{\boldsymbol{x}*}}{\rho_t} \boldsymbol{A}\boldsymbol{e}_t^{\boldsymbol{x}}\right]. \qquad\qquad \text{(sparsity of } \boldsymbol{B}\text{)}$$

For any and $l \in [2d]$, recall that

$$\left(\frac{\boldsymbol{E}_t^{\boldsymbol{x}} \boldsymbol{E}_t^{\boldsymbol{x}*}}{\rho_t} \boldsymbol{A}\boldsymbol{e}_t^{\boldsymbol{x}}\right)_l = \begin{cases} \frac{a}{\rho_t} \sum_{i=1}^{t-1} \sum_{r=1}^{d} \lambda_l^i \lambda_r^{t-i} x_{1l} x_{1r}^2, & l \in [d], \\ \frac{a}{\rho_t} \sum_{i=1}^{t-1} \sum_{r=1}^{d} \lambda_{l-d}^{i-1} \lambda_r^{t-i} x_{1,l-d} x_{1r}^2, & l \in [2d] - [d], \end{cases}$$

and for any $j \in [d]$, we have

$$\left(\frac{\overline{\boldsymbol{E}_t^{\boldsymbol{x}} \boldsymbol{E}_t^{\boldsymbol{x}\top}}}{\rho_t} \boldsymbol{A}\overline{\boldsymbol{e}_t^{\boldsymbol{x}}}\right)_j = \left(\frac{\boldsymbol{E}_t^{\boldsymbol{x}} \boldsymbol{E}_t^{\boldsymbol{x}*}}{\rho_t} \boldsymbol{A}\boldsymbol{e}_t^{\boldsymbol{x}}\right)_j = \frac{a}{\rho_t} \sum_{i=1}^{t-1} \sum_{r=1}^{d} \lambda_j^{-i} \lambda_r^{i-t} x_{1j} x_{1r}^2.$$

For any $l \in [d]$, with careful computing, we have

$$\mathbb{E}\left[b\left(\frac{\overline{\boldsymbol{E}_t^{\boldsymbol{x}} \boldsymbol{E}_t^{\boldsymbol{x}\top}}}{\rho_t} \boldsymbol{A}\overline{\boldsymbol{e}_t^{\boldsymbol{x}}}\right)_j \cdot \left(\frac{\boldsymbol{E}_t^{\boldsymbol{x}} \boldsymbol{E}_t^{\boldsymbol{x}*}}{\rho_t} \boldsymbol{A}\boldsymbol{e}_t^{\boldsymbol{x}}\right)_l\right]$$

$$= \frac{a^2 b}{\rho_t^2} \sum_{i_1=1}^{t-1} \sum_{i_2=1}^{t-1} \sum_{r_1=1}^{d} \sum_{r_2=1}^{d} \mathbb{E}[\lambda_j^{-i_1} \lambda_l^{i_2} \lambda_{r_1}^{i_1-t} \lambda_{r_2}^{t-i_2}] \mathbb{E}[x_{1j} x_{il} x_{1r_1}^2 x_{1r_2}^2].$$

We discuss it in the following categories,

1. $l \neq j$. In this case, $\mathbb{E}[x_{1j}x_{il}x_{1r_1}^2 x_{1r_2}^2] = 0$ by Assumption 4.1, thus it becomes 0.

2. $l = j, r_1 = r_2 = j$. It becomes

$$\frac{a^2 b}{\rho_t^2} \sum_{i_1=1}^{t-1} \sum_{i_2=1}^{t-1} \mathbb{E}[\lambda_j^{-i_1} \lambda_j^{i_2} \lambda_j^{i_1-t} \lambda_j^{t-i_2}] \mathbb{E}[x_{1j}^6] = \frac{a^2 b}{\rho_t^2} \sum_{i_1=1}^{t-1} \sum_{i_2=1}^{t-1} \mathbb{E}[x_{1j}^6] = \frac{a^2 b}{\rho_t^2}(t-1)^2 \mathbb{E}[x_{1j}^6].$$

3. $l = j, r_1 = r_2 = r \neq j$. It becomes

$$\frac{a^2 b}{\rho_t^2} \sum_{i_1=1}^{t-1} \sum_{i_2=1}^{t-1} \sum_{r \neq j} \mathbb{E}[\lambda_j^{i_2-i_1} \lambda_r^{i_1-i_2}] \mathbb{E}[x_{1j}^2 x_{1r}^4] = \frac{a^2 b}{\rho_t^2} \sum_{i_1=i_2} \sum_{r \neq j} \mathbb{E}[x_{1j}^2 x_{1r}^4]$$
$$= \frac{a^2 b}{\rho_t^2}(t-1) \sum_{r \neq j} \mathbb{E}[x_{1j}^2 x_{1r}^4].$$

4. $l = j, r_1 \neq r_2$. In this case, $\mathbb{E}[\lambda_j^{-i_1} \lambda_l^{i_2} \lambda_{r_1}^{i_1-t} \lambda_{r_2}^{t-i_2}] = \mathbb{E}[\lambda_j^{i_2-i_1} \lambda_{r_1}^{i_1-t} \lambda_{r_2}^{t-i_2}] = 0$, thus it becomes 0.

Similarly, for any $l \in [2d] - [d]$, we have

$$\mathbb{E}\left[ b \left( \frac{\overline{E_t^x E_t^{x\top}}}{\rho_t} A \overline{e_t^x} \right)_j \cdot \left( \frac{E_t^x E_t^{x*}}{\rho_t} A e_t^x \right)_l \right] = 0.$$

To sum up, we have

$$\mathbb{E}\left[ b \left( \frac{\overline{E_t^x E_t^{x\top}}}{\rho_t} A \overline{e_t^x} \right)_j \cdot \left( \frac{E_t^x E_t^{x*}}{\rho_t} A e_t^x \right)_l \right]$$
$$= \begin{cases} \frac{a^2 b}{\rho_t^2}\left[ (t-1)^2 \mathbb{E}[x_{1j}^6] + (t-1) \sum_{r \neq j} \mathbb{E}[x_{1j}^2 x_{1r}^4] \right], & l = j, \\ 0, & \text{otherwise,} \end{cases}$$

which implies that the $l$-th element of $\frac{1}{2}\mathbb{E}\left[ \nabla_{\boldsymbol{B}_j} \widehat{y}_{t,j}^* \widehat{y}_{t,j} \right]$ is

$$\frac{1}{2}\mathbb{E}\left[ \nabla_{\boldsymbol{B}_j} \widehat{y}_{t,j}^* \widehat{y}_{t,j} \right]_l = \begin{cases} \frac{a^2 b}{\rho_t^2}\left[ (t-1)^2 \mathbb{E}[x_{1j}^6] + (t-1) \sum_{r \neq j} \mathbb{E}[x_{1j}^2 x_{1r}^4] \right], & l = j, \\ 0, & \text{otherwise.} \end{cases}$$

**Step three: calculate $\nabla_{\boldsymbol{B}_j} L_t(\boldsymbol{\theta})$ and $\nabla_{\boldsymbol{B}_j} L(\boldsymbol{\theta})$.** Based on steps one and two, the $l$-th element of $\nabla_{\boldsymbol{B}_j} L_t(\boldsymbol{\theta})$ can be derived as follows.

$$\nabla_{\boldsymbol{B}_j} L_t(\boldsymbol{\theta})_l = \frac{1}{2}\mathbb{E}\left[ \nabla_{\boldsymbol{B}_j} \widehat{y}_{t,j}^* \widehat{y}_{t,j} \right]_l - \mathbb{E}\left[ \nabla_{\boldsymbol{B}_j} \mathrm{Re}\left( x_{t+1,j}^* \widehat{y}_{t,j} \right) \right]_l$$
$$= \begin{cases} \frac{a^2 b}{\rho_t^2}\left[ (t-1)^2 \mathbb{E}[x_{1j}^6] + (t-1) \sum_{r \neq j} \mathbb{E}[x_{1j}^2 x_{1r}^4] \right] - \frac{a}{\rho_t}(t-1)\mathbb{E}[x_{1j}^4], & l = j, \\ 0, & \text{otherwise,} \end{cases}.$$

Furthermore, the $l$-th element of $\nabla_{\boldsymbol{B}_j} L(\boldsymbol{\theta})$ is

$$\nabla_{\boldsymbol{B}_j} L(\boldsymbol{\theta})_l = \sum_{t=2}^{T-1} \nabla_{\boldsymbol{B}_j} L_t(\boldsymbol{\theta})_l$$
$$= \begin{cases} \sum_{t=2}^{T-1}\left( \frac{a^2 b}{\rho_t^2}\left[ (t-1)^2 \mathbb{E}[x_{1j}^6] + (t-1) \sum_{r \neq j} \mathbb{E}[x_{1j}^2 x_{1r}^4] \right] - \frac{a}{\rho_t}(t-1)\mathbb{E}[x_{1j}^4] \right), & l = j, \\ 0, & \text{otherwise,} \end{cases}$$

$$
= \begin{cases} \sum_{t=2}^{T-1} \left( a^2 b \left[ \mathbb{E}[x_{1j}^6] + \frac{1}{t-1} \sum_{r \neq j} \mathbb{E}[x_{1j}^2 x_{1r}^4] \right] - a\mathbb{E}[x_{1j}^4] \right), & l = j, \\ 0, & \text{otherwise,} \end{cases}
$$

$$
= \begin{cases} a^2 b \left[ (T-2)\mathbb{E}[x_{1j}^6] + \sum_{t=2}^{T-1} \frac{1}{t-1} \sum_{r \neq j} \mathbb{E}[x_{1j}^2 x_{1r}^4] \right] - a(T-2)\mathbb{E}[x_{1j}^4], & l = j, \\ 0, & \text{otherwise,} \end{cases}
$$

$$
= \begin{cases} a^2 b \left[ (T-2)\kappa_2 + \sum_{t=2}^{T-1} \frac{1}{t-1} \kappa_3 \right] - a(T-2)\kappa_1, & l = j, \\ 0, & \text{otherwise,} \end{cases}
$$

where the last equality is from Assumption 4.1.

**Step four:** calculate $\mathbb{E}\left[ \nabla_{\mathrm{Vec}(A)} \mathrm{Re}\left( x_{t+1,j}^* \widehat{y}_{t,j} \right) \right]$ and $\sum_{j=1}^d \mathbb{E}\left[ \nabla_{\mathrm{Vec}(A)} \mathrm{Re}\left( x_{t+1,j}^* \widehat{y}_{t,j} \right) \right]$. Based on Lemma 5.1, we have

$$
\widehat{y}_{t,j} = \mathrm{Vec}^\top(A) \cdot e_t^x \otimes \frac{\overline{E_t^x} E_t^{x\top}}{\rho_t} \cdot B_j.
$$

Then, the $\mathbb{E}\left[ \nabla_{\mathrm{Vec}(A)} \mathrm{Re}\left( x_{t+1,j}^* \widehat{y}_{t,j} \right) \right]$ can be derived as the following.

$$
\begin{aligned}
\mathbb{E}\left[ \nabla_{\mathrm{Vec}(A)} \mathrm{Re}\left( x_{t+1,j}^* \widehat{y}_{t,j} \right) \right] &= \mathbb{E}\left[ \nabla_{\mathrm{Vec}(A)} \mathrm{Re}\left( x_{t+1,j}^* \mathrm{Vec}^\top(A) \cdot e_t^x \otimes \frac{\overline{E_t^x} E_t^{x\top}}{\rho_t} \cdot B_j \right) \right] \\
&= \mathbb{E}\left[ \nabla_{\mathrm{Vec}(A)} \mathrm{Vec}^\top(A) \cdot \mathrm{Re}\left( x_{t+1,j}^* \cdot e_t^x \otimes \frac{\overline{E_t^x} E_t^{x\top}}{\rho_t} \cdot B_j \right) \right] \\
&= \mathbb{E}\left[ \mathrm{Re}\left( x_{t+1,j}^* \cdot e_t^x \otimes \frac{\overline{E_t^x} E_t^{x\top}}{\rho_t} \cdot B_j \right) \right] \\
&= \mathrm{Re}\left( \mathbb{E}\left[ x_{t+1,j}^* \cdot e_t^x \otimes \frac{\overline{E_t^x} E_t^{x\top}}{\rho_t} \cdot B_j \right] \right) \\
&= \mathrm{Re}\left( \mathbb{E}\left[ \lambda_j^{-t} x_{1j} \cdot \mathrm{Vec}(\frac{\overline{E_t^x} E_t^{x\top}}{\rho_t} B_j e_t^{x\top}) \right] \right).
\end{aligned}
$$

In terms of $\frac{\overline{E_t^x} E_t^{x\top}}{\rho_t} B_j e_t^{x\top}$, based on the sparsity of $B$ and Eq. 4, we can derive that

$$
\begin{aligned}
\frac{\overline{E_t^x} E_t^{x\top}}{\rho_t} B_j e_t^{x\top} &= (\frac{\overline{E_t^x} E_t^{x\top}}{\rho_t})_{:j} b e_t^{x\top} = (\frac{E_t^x E_t^{x*}}{\rho_t})_{j:}^\top b e_t^{x\top} \\
&= b \left( \sum_{i=1}^t M_i \odot \widehat{\Sigma} \quad W \sum_{i=1}^{t-1} M_i \odot \widehat{\Sigma} \right)_{j:}^\top e_t^{x\top}.
\end{aligned}
$$

Then, for any $s, r \in [2d]$, we have

$$
\left( \frac{\overline{E_t^x} E_t^{x\top}}{\rho_t} B_j e_t^{x\top} \right)_{sr} = \begin{cases} \frac{b}{\rho_t} \sum_{i=1}^t \lambda_j^{i-1} \lambda_s^{1-i} \lambda_r^{t-1} x_{1j} x_{1s} x_{1r}, & s \in [d], r \in [d], \\ \frac{b}{\rho_t} \sum_{i=1}^{t-1} \lambda_j^i \lambda_{s-d}^{1-i} \lambda_r^{t-1} x_{1j} x_{1,s-d} x_{1r}, & s \in [2d] - [d], r \in [d], \\ \frac{b}{\rho_t} \sum_{i=1}^t \lambda_j^{i-1} \lambda_s^{1-i} \lambda_{r-d}^{t-2} x_{1j} x_{1s} x_{1,r-d}, & s \in [d], r \in [2d] - [d], \\ \frac{b}{\rho_t} \sum_{i=1}^{t-1} \lambda_j^i \lambda_{s-d}^{1-i} \lambda_{r-d}^{t-2} x_{1j} x_{1,s-d} x_{1,r-d}, & s \in [2d] - [d], r \in [2d] - [d]. \end{cases}
$$

Next, we calculate $\mathbb{E}\left[\lambda_j^{-t}x_{1j} \cdot \frac{\overline{E_t^x E_t^{x\top}}}{\rho_t} B_j e_t^{x\top}\right]$. For any $s \in [2d] - [d], r \in [d]$, we have

$$\mathbb{E}\left[\lambda_j^{-t}x_{1j}\left(\frac{\overline{E_t^x E_t^{x\top}}}{\rho_t}B_j e_t^{x\top}\right)_{sr}\right] = \mathbb{E}\left[\frac{b}{\rho_t}\sum_{i=1}^{t-1}\lambda_j^{i-t}\lambda_{s-d}^{1-i}\lambda_r^{t-1}x_{1j}^2 x_{1,s-d}x_{1r}\right]$$

$$= \frac{b}{\rho_t}\sum_{i=1}^{t-1}\mathbb{E}[\lambda_j^{i-t}\lambda_{s-d}^{1-i}\lambda_r^{t-1}]\mathbb{E}[x_{1j}^2 x_{1,s-d}x_{1r}].$$

We discuss it in the following categories,

1. $s - d \neq r$. In this case, $\mathbb{E}[x_{1j}^2 x_{1,s-d}x_{1r}] = 0$ by Assumption 4.1, thus it becomes 0.

2. $s - d = r = j$. It becomes

$$\frac{b}{\rho_t}\sum_{i=1}^{t-1}\mathbb{E}[\lambda_j^{i-t}\lambda_{s-d}^{1-i}\lambda_r^{t-1}]\mathbb{E}[x_{1j}^2 x_{1,s-d}x_{1r}] = \frac{b}{\rho_t}\sum_{i=1}^{t-1}\mathbb{E}[\lambda_j^{i-t}\lambda_j^{1-i}\lambda_j^{t-1}]\mathbb{E}[x_{1j}^4]$$

$$= \frac{b}{\rho_t}\sum_{i=1}^{t-1}\mathbb{E}[x_{1j}^4] = \frac{b}{\rho_t}(t-1)\mathbb{E}[x_{1j}^4].$$

3. $s - d = r \neq j$. In this case, $\mathbb{E}[\lambda_j^{i-t}\lambda_{s-d}^{1-i}\lambda_r^{t-1}] = \mathbb{E}[\lambda_j^{i-t}\lambda_{s-d}^{t-i}] = 0$, thus it becomes 0.

Similarly, for any other $s, r$, we can calculate that

$$\mathbb{E}\left[\lambda_j^{-t}\left(\frac{\overline{E_t^x E_t^{x\top}}}{\rho_t}B_j e_t^{x\top}\right)_{sr}\right] = 0.$$

To sum up, we have

$$\mathbb{E}\left[\lambda_j^{-t}\left(\frac{\overline{E_t^x E_t^{x\top}}}{\rho_t}B_j e_t^{x\top}\right)_{sr}\right] = \begin{cases} \frac{b}{\rho_t}(t-1)\mathbb{E}[x_{1j}^4], & s = d+j, r = j, \\ 0, & \text{otherwise,} \end{cases}$$

thus the $(s, r)$-th element of $\mathbb{E}\left[\nabla_A \mathrm{Re}\left(x_{t+1,j}^* \widehat{y}_{t,j}\right)\right]$ is

$$\mathbb{E}\left[\nabla_A \mathrm{Re}\left(x_{t+1,j}^* \widehat{y}_{t,j}\right)\right]_{sr} = \begin{cases} \frac{b}{\rho_t}(t-1)\mathbb{E}[x_{1j}^4], & s = d+j, r = j, \\ 0, & \text{otherwise,} \end{cases}$$

$$= \begin{cases} \frac{b}{\rho_t}(t-1)\kappa_1, & s = d+j, r = j, \\ 0, & \text{otherwise.} \end{cases}$$

Finally, we can calculate $\sum_{j=1}^{d}\mathbb{E}\left[\nabla_A \mathrm{Re}\left(x_{t+1,j}^* \widehat{y}_{t,j}\right)\right]$ as

$$\sum_{j=1}^{d}\mathbb{E}\left[\nabla_A \mathrm{Re}\left(x_{t+1,j}^* \widehat{y}_{t,j}\right)\right]_{sr} = \begin{cases} \frac{b}{\rho_t}(t-1)\kappa_1, & s - d = r, \\ 0, & \text{otherwise.} \end{cases}$$

**Step five: calculate** $\mathbb{E}\left[\nabla_{\mathrm{Vec}(A)}\widehat{y}_{t,j}^* \widehat{y}_{t,j}\right]$ **and** $\sum_{j=1}^{d}\mathbb{E}\left[\nabla_{\mathrm{Vec}(A)}\widehat{y}_{t,j}^* \widehat{y}_{t,j}\right]$. Based on Lemma 5.1, we have

$$\widehat{y}_{t,j} = B_j^\top \cdot e_t^{x\top} \otimes \frac{E_t^x E_t^{x*}}{\rho_t} \cdot \mathrm{Vec}(A),$$

then we can simplify the $\mathbb{E}\left[\nabla_{\mathrm{Vec}(\boldsymbol{A})}\widehat{y}^*_{t,j}\widehat{y}_{t,j}\right]$ as follows

$$\mathbb{E}\left[\nabla_{\mathrm{Vec}(\boldsymbol{A})}\widehat{y}^*_{t,j}\widehat{y}_{t,j}\right]$$

$$= \mathbb{E}\left[\nabla_{\mathrm{Vec}(\boldsymbol{A})}\left(\boldsymbol{B}_j^\top \cdot \boldsymbol{e}_t^{\boldsymbol{x}\top} \otimes \frac{\boldsymbol{E}_t^{\boldsymbol{x}}\boldsymbol{E}_t^{\boldsymbol{x}*}}{\rho_t}\cdot \mathrm{Vec}(\boldsymbol{A})\right)^* \boldsymbol{B}_j^\top \cdot \boldsymbol{e}_t^{\boldsymbol{x}\top} \otimes \frac{\boldsymbol{E}_t^{\boldsymbol{x}}\boldsymbol{E}_t^{\boldsymbol{x}*}}{\rho_t}\cdot \mathrm{Vec}(\boldsymbol{A})\right]$$

$$= \mathbb{E}\left[\nabla_{\mathrm{Vec}(\boldsymbol{A})}\mathrm{Vec}(\boldsymbol{A})^\top \cdot \overline{\boldsymbol{e}_t^{\boldsymbol{x}}} \otimes \frac{\boldsymbol{E}_t^{\boldsymbol{x}}\boldsymbol{E}_t^{\boldsymbol{x}*}}{\rho_t}\boldsymbol{B}_j \cdot \boldsymbol{B}_j^\top \cdot \boldsymbol{e}_t^{\boldsymbol{x}\top} \otimes \frac{\boldsymbol{E}_t^{\boldsymbol{x}}\boldsymbol{E}_t^{\boldsymbol{x}*}}{\rho_t}\cdot \mathrm{Vec}(\boldsymbol{A})\right]$$

$$= \mathbb{E}\left[\overline{\boldsymbol{e}_t^{\boldsymbol{x}}} \otimes \frac{\boldsymbol{E}_t^{\boldsymbol{x}}\boldsymbol{E}_t^{\boldsymbol{x}*}}{\rho_t}\boldsymbol{B}_j \cdot \boldsymbol{B}_j^\top \cdot \boldsymbol{e}_t^{\boldsymbol{x}\top} \otimes \frac{\boldsymbol{E}_t^{\boldsymbol{x}}\boldsymbol{E}_t^{\boldsymbol{x}*}}{\rho_t}\cdot \mathrm{Vec}(\boldsymbol{A})\right]$$

$$+ \mathbb{E}\left[\boldsymbol{e}_t^{\boldsymbol{x}} \otimes \frac{\overline{\boldsymbol{E}_t^{\boldsymbol{x}}\boldsymbol{E}_t^{\boldsymbol{x}\top}}}{\rho_t}\boldsymbol{B}_j \cdot \boldsymbol{B}_j^\top \cdot \boldsymbol{e}_t^{\boldsymbol{x}*} \otimes \frac{\overline{\boldsymbol{E}_t^{\boldsymbol{x}}\boldsymbol{E}_t^{\boldsymbol{x}\top}}}{\rho_t}\cdot \mathrm{Vec}(\boldsymbol{A})\right]$$

$$= \mathbb{E}\left[2\mathrm{Re}\left(\boldsymbol{e}_t^{\boldsymbol{x}} \otimes \frac{\overline{\boldsymbol{E}_t^{\boldsymbol{x}}\boldsymbol{E}_t^{\boldsymbol{x}\top}}}{\rho_t}\boldsymbol{B}_j \cdot \boldsymbol{B}_j^\top \cdot \boldsymbol{e}_t^{\boldsymbol{x}*} \otimes \frac{\overline{\boldsymbol{E}_t^{\boldsymbol{x}}\boldsymbol{E}_t^{\boldsymbol{x}\top}}}{\rho_t}\cdot \mathrm{Vec}(\boldsymbol{A})\right)\right]$$

$$= 2\mathrm{Re}\left(\mathbb{E}\left[\boldsymbol{e}_t^{\boldsymbol{x}} \otimes \frac{\overline{\boldsymbol{E}_t^{\boldsymbol{x}}\boldsymbol{E}_t^{\boldsymbol{x}\top}}}{\rho_t}\boldsymbol{B}_j \cdot \boldsymbol{B}_j^\top \cdot \boldsymbol{e}_t^{\boldsymbol{x}*} \otimes \frac{\overline{\boldsymbol{E}_t^{\boldsymbol{x}}\boldsymbol{E}_t^{\boldsymbol{x}\top}}}{\rho_t}\cdot \mathrm{Vec}(\boldsymbol{A})\right]\right).$$

We further derive that

$$\mathbb{E}\left[\boldsymbol{e}_t^{\boldsymbol{x}} \otimes \frac{\overline{\boldsymbol{E}_t^{\boldsymbol{x}}\boldsymbol{E}_t^{\boldsymbol{x}\top}}}{\rho_t}\boldsymbol{B}_j \cdot \underbrace{\boldsymbol{B}_j^\top \cdot \boldsymbol{e}_t^{\boldsymbol{x}*} \otimes \frac{\overline{\boldsymbol{E}_t^{\boldsymbol{x}}\boldsymbol{E}_t^{\boldsymbol{x}\top}}}{\rho_t}\cdot \mathrm{Vec}(\boldsymbol{A})}_{\in\mathbb{C}}\right]$$

$$= \mathbb{E}\left[\boldsymbol{B}_j^\top \cdot \boldsymbol{e}_t^{\boldsymbol{x}*} \otimes \frac{\overline{\boldsymbol{E}_t^{\boldsymbol{x}}\boldsymbol{E}_t^{\boldsymbol{x}\top}}}{\rho_t}\cdot \mathrm{Vec}(\boldsymbol{A}) \cdot \boldsymbol{e}_t^{\boldsymbol{x}} \otimes \frac{\overline{\boldsymbol{E}_t^{\boldsymbol{x}}\boldsymbol{E}_t^{\boldsymbol{x}\top}}}{\rho_t}\boldsymbol{B}_j\right]$$

$$= \mathbb{E}\left[\mathrm{Vec}(\boldsymbol{A})^\top \cdot \overline{\boldsymbol{e}_t^{\boldsymbol{x}}} \otimes \frac{\boldsymbol{E}_t^{\boldsymbol{x}}\boldsymbol{E}_t^{\boldsymbol{x}*}}{\rho_t}\boldsymbol{B}_j \cdot \boldsymbol{e}_t^{\boldsymbol{x}} \otimes \frac{\overline{\boldsymbol{E}_t^{\boldsymbol{x}}\boldsymbol{E}_t^{\boldsymbol{x}\top}}}{\rho_t}\boldsymbol{B}_j\right]$$

$$= \mathbb{E}\left[\mathrm{Vec}(\boldsymbol{A})^\top \cdot \mathrm{Vec}(\frac{\boldsymbol{E}_t^{\boldsymbol{x}}\boldsymbol{E}_t^{\boldsymbol{x}*}}{\rho_t}\boldsymbol{B}_j\boldsymbol{e}_t^{\boldsymbol{x}*}) \cdot \mathrm{Vec}(\frac{\overline{\boldsymbol{E}_t^{\boldsymbol{x}}\boldsymbol{E}_t^{\boldsymbol{x}\top}}}{\rho_t}\boldsymbol{B}_j\boldsymbol{e}_t^{\boldsymbol{x}\top})\right]$$

$$= \mathbb{E}\left[\sum_{k,l=1}^{2d} A_{kl}\left(\frac{\boldsymbol{E}_t^{\boldsymbol{x}}\boldsymbol{E}_t^{\boldsymbol{x}*}}{\rho_t}\boldsymbol{B}_j\boldsymbol{e}_t^{\boldsymbol{x}*}\right)_{kl} \cdot \mathrm{Vec}(\frac{\overline{\boldsymbol{E}_t^{\boldsymbol{x}}\boldsymbol{E}_t^{\boldsymbol{x}\top}}}{\rho_t}\boldsymbol{B}_j\boldsymbol{e}_t^{\boldsymbol{x}\top})\right]$$

$$= \mathbb{E}\left[\sum_{k=d+1}^{2d} a\left(\frac{\boldsymbol{E}_t^{\boldsymbol{x}}\boldsymbol{E}_t^{\boldsymbol{x}*}}{\rho_t}\boldsymbol{B}_j\boldsymbol{e}_t^{\boldsymbol{x}*}\right)_{k,k-d} \cdot \mathrm{Vec}(\frac{\overline{\boldsymbol{E}_t^{\boldsymbol{x}}\boldsymbol{E}_t^{\boldsymbol{x}\top}}}{\rho_t}\boldsymbol{B}_j\boldsymbol{e}_t^{\boldsymbol{x}\top})\right]. \qquad \text{(sparsity of } \boldsymbol{A}\text{)}$$

Recall that for any $s, r \in [2d]$, we have

$$\left(\frac{\overline{\boldsymbol{E}_t^{\boldsymbol{x}}\boldsymbol{E}_t^{\boldsymbol{x}\top}}}{\rho_t}\boldsymbol{B}_j\boldsymbol{e}_t^{\boldsymbol{x}\top}\right)_{sr} = \begin{cases} \frac{b}{\rho_t}\sum_{i=1}^t \lambda_j^{i-1}\lambda_s^{1-i}\lambda_r^{t-1}x_{1j}x_{1s}x_{1r}, & s \in [d], r \in [d], \\ \frac{b}{\rho_t}\sum_{i=1}^{t-1} \lambda_j^i\lambda_{s-d}^{1-i}\lambda_r^{t-1}x_{1j}x_{1,s-d}x_{1r}, & s \in [2d]-[d], r \in [d], \\ \frac{b}{\rho_t}\sum_{i=1}^t \lambda_j^{i-1}\lambda_s^{1-i}\lambda_{r-d}^{t-2}x_{1j}x_{1s}x_{1,r-d}, & s \in [d], r \in [2d]-[d], \\ \frac{b}{\rho_t}\sum_{i=1}^{t-1} \lambda_j^i\lambda_{s-d}^{1-i}\lambda_{r-d}^{t-2}x_{1j}x_{1,s-d}x_{1,r-d}, & s \in [2d]-[d], r \in [2d]-[d]. \end{cases}$$

Furthermore, for any $k \in [2d]-[d]$, we can calculate that

$$\left(\frac{\boldsymbol{E}_t^{\boldsymbol{x}}\boldsymbol{E}_t^{\boldsymbol{x}*}}{\rho_t}\boldsymbol{B}_j\boldsymbol{e}_t^{\boldsymbol{x}*}\right)_{k,k-d} = \left(\frac{\overline{\boldsymbol{E}_t^{\boldsymbol{x}}\boldsymbol{E}_t^{\boldsymbol{x}\top}}}{\rho_t}\boldsymbol{B}_j\boldsymbol{e}_t^{\boldsymbol{x}\top}\right)_{k,k-d}$$

$$= \frac{b}{\rho_t} \sum_{i=1}^{t-1} \lambda_j^{-i} \lambda_{k-d}^{i-1} \lambda_{k-d}^{1-t} x_{1j} x_{1,k-d}^2 = \frac{b}{\rho_t} \sum_{i=1}^{t-1} \lambda_j^{-i} \lambda_{k-d}^{i-t} x_{1j} x_{1,k-d}^2,$$

With careful computing, for any $s \in [2d] - [d], r \in [d]$, we have

$$\mathbb{E}\left[ \sum_{k=d+1}^{2d} a\left( \frac{\boldsymbol{E}_t^{\boldsymbol{x}} \boldsymbol{E}_t^{\boldsymbol{x}*}}{\rho_t} \boldsymbol{B}_j \boldsymbol{e}_t^{\boldsymbol{x}*} \right)_{k,k-d} \cdot \left( \frac{\overline{\boldsymbol{E}_t^{\boldsymbol{x}} \boldsymbol{E}_t^{\boldsymbol{x}\top}}}{\rho_t} \boldsymbol{B}_j \boldsymbol{e}_t^{\boldsymbol{x}\top} \right)_{sr} \right]$$

$$= \frac{ab^2}{\rho_t^2} \sum_{k=d+1}^{2d} \sum_{i_1=1}^{t-1} \sum_{i_2=1}^{t-1} \mathbb{E}[\lambda_j^{i_2-i_1} \lambda_{k-d}^{i_1-t} \lambda_{s-d}^{1-i_2} \lambda_r^{t-1}] \mathbb{E}[x_{1j}^2 x_{1,k-d}^2 x_{1,s-d} x_{1r}].$$

We discuss it in the following categories,

1. $s - d \neq r$. In this case, $\mathbb{E}[x_{1j}^2 x_{1,k-d}^2 x_{1,s-d} x_{1r}] = 0$ by Assumption 4.1, thus it becomes $0$.

2. $s - d = r = k - d = j$. It becomes

$$\frac{ab^2}{\rho_t^2} \sum_{i_1=1}^{t-1} \sum_{i_2=1}^{t-1} \mathbb{E}[\lambda_j^0] \mathbb{E}[x_{1j}^6] = \frac{ab^2}{\rho_t^2} (t-1)^2 \mathbb{E}[x_{1j}^6].$$

3. $s - d = r = k - d \neq j$. It becomes

$$\frac{ab^2}{\rho_t^2} \sum_{i_1=1}^{t-1} \sum_{i_2=1}^{t-1} \mathbb{E}[\lambda_j^{i_2-i_1} \lambda_r^{i_1-i_2}] \mathbb{E}[x_{1j}^2 x_{1,r}^4] = \frac{ab^2}{\rho_t^2} \sum_{i_1=1}^{t-1} \mathbb{E}[\lambda_j^{i_1-i_1} \lambda_r^{i_1-i_1}] \mathbb{E}[x_{1j}^2 x_{1,r}^4]$$

$$= \frac{ab^2}{\rho_t^2} (t-1) \mathbb{E}[x_{1j}^2 x_{1,r}^4].$$

4. $s - d = r \neq k - d$. In these case, $\mathbb{E}[\lambda_j^{i_2-i_1} \lambda_{k-d}^{i_1-t} \lambda_{s-d}^{1-i_2} \lambda_r^{t-1}] = \mathbb{E}[\lambda_j^{i_2-i_1} \lambda_{k-d}^{i_1-t} \lambda_r^{t-i_2}] = 0$, thus it becomes $0$.

Similarly, for any other $s, r$, we can calculate that

$$\mathbb{E}\left[ \sum_{k=d+1}^{2d} a\left( \frac{\boldsymbol{E}_t^{\boldsymbol{x}} \boldsymbol{E}_t^{\boldsymbol{x}*}}{\rho_t} \boldsymbol{B}_j \boldsymbol{e}_t^{\boldsymbol{x}*} \right)_{k,k-d} \cdot \left( \frac{\overline{\boldsymbol{E}_t^{\boldsymbol{x}} \boldsymbol{E}_t^{\boldsymbol{x}\top}}}{\rho_t} \boldsymbol{B}_j \boldsymbol{e}_t^{\boldsymbol{x}\top} \right)_{sr} \right] = 0.$$

To sum up, we have

$$\mathbb{E}\left[ \sum_{k=d+1}^{2d} a\left( \frac{\boldsymbol{E}_t^{\boldsymbol{x}} \boldsymbol{E}_t^{\boldsymbol{x}*}}{\rho_t} \boldsymbol{B}_j \boldsymbol{e}_t^{\boldsymbol{x}*} \right)_{k,k-d} \cdot \left( \frac{\overline{\boldsymbol{E}_t^{\boldsymbol{x}} \boldsymbol{E}_t^{\boldsymbol{x}\top}}}{\rho_t} \boldsymbol{B}_j \boldsymbol{e}_t^{\boldsymbol{x}\top} \right)_{sr} \right]$$

$$= \begin{cases} \frac{ab^2}{\rho_t^2} (t-1)^2 \mathbb{E}[x_{1j}^6], & s = d+r, r = j, \\ \frac{ab^2}{\rho_t^2} (t-1) \mathbb{E}[x_{1j}^2 x_{1r}^4], & s = d+r, r \neq j, \\ 0, & \text{otherwise}, \end{cases}$$

which implies that the $(s,r)$-th element of $\frac{1}{2}\mathbb{E}\left[ \nabla_{\boldsymbol{A}} \widehat{y}_{t,j}^* \widehat{y}_{t,j} \right]$ is

$$\frac{1}{2}\mathbb{E}\left[ \nabla_{\boldsymbol{A}} \widehat{y}_{t,j}^* \widehat{y}_{t,j} \right]_{sr} = \begin{cases} \frac{ab^2}{\rho_{t_2}^2} (t-1)^2 \mathbb{E}[x_{1j}^6], & s = d+r, r = j, \\ \frac{ab^2}{\rho_t^2} (t-1) \mathbb{E}[x_{1j}^2 x_{1r}^4], & s = d+r, r \neq j, \\ 0, & \text{otherwise}, \end{cases}$$

$$= \begin{cases} \frac{ab^2}{\rho_{t_2}^2} (t-1)^2 \kappa_2, & s = d+r, r = j, \\ \frac{ab^2}{\rho_t^2} (t-1) \mathbb{E}[x_{1j}^2 x_{1r}^4], & s = d+r, r \neq j, \\ 0, & \text{otherwise}, \end{cases} \cdot$$

Based on these results, we can derive

$$\frac{1}{2}\sum_{j=1}^{d}\mathbb{E}\Big[\nabla_{\boldsymbol{A}}\widehat{y}_{t,j}^{*}\,\widehat{y}_{t,j}\Big]_{sr} = \begin{cases} \frac{ab^2}{\rho_t^2}\big[(t-1)^2\kappa_2 + (t-1)\kappa_3\big], & s-d=r, \\ 0, & \text{otherwise.} \end{cases}$$

**Step six: calculate $\nabla_{\mathrm{Vec}(\boldsymbol{A})}L_t(\boldsymbol{\theta})$ and $\nabla_{\mathrm{Vec}(\boldsymbol{A})}L(\boldsymbol{\theta})$.** Based on steps four and five, the $(s,r)$-th element of $\nabla_{\boldsymbol{A}}L_t(\boldsymbol{\theta})$ can be derived as follows.

$$\begin{aligned}
\nabla_{\boldsymbol{A}}L_t(\boldsymbol{\theta})_{sr} &= \frac{1}{2}\sum_{j=1}^{d}\mathbb{E}\Big[\nabla_{\boldsymbol{A}}\widehat{y}_{t,j}^{*}\,\widehat{y}_{t,j}\Big] - \sum_{j=1}^{d}\mathbb{E}\Big[\nabla_{\boldsymbol{A}}\mathrm{Re}\left(x_{t+1,j}^{*}\widehat{y}_{t,j}\right)\Big] \\
&= \begin{cases} \frac{ab^2}{\rho_t^2}\big[(t-1)^2\kappa_2 + (t-1)\kappa_3\big] - \frac{b}{\rho_t}(t-1)\kappa_1, & s-d=r, \\ 0, & \text{otherwise.} \end{cases}
\end{aligned}$$

Furthermore, the $(s,r)$-th element of $\nabla_{\boldsymbol{A}}L(\boldsymbol{\theta})$ is

$$\begin{aligned}
\nabla_{\boldsymbol{A}}L(\boldsymbol{\theta})_{sr} &= \sum_{t=2}^{T-1}\nabla_{\boldsymbol{A}}L_t(\boldsymbol{\theta})_{sr} \\
&= \begin{cases} \sum_{t=2}^{T-1}\left(\frac{ab^2}{\rho_t^2}\big[(t-1)^2\kappa_2 + (t-1)\kappa_3\big] - \frac{b}{\rho_t}(t-1)\kappa_1\right), & s-d=r, \\ 0, & \text{otherwise,} \end{cases} \\
&= \begin{cases} \sum_{t=2}^{T-1}\left(ab^2\big[\kappa_2 + \frac{1}{t-1}\kappa_3\big] - b\kappa_1\right), & s-d=r, \\ 0, & \text{otherwise,} \end{cases} \\
&= \begin{cases} ab^2\big[(T-2)\kappa_2 + \sum_{t=2}^{T-1}\frac{1}{t-1}\kappa_3\big] - b(T-2)\kappa_1, & s-d=r, \\ 0, & \text{otherwise.} \end{cases}
\end{aligned}$$

**Step seven: summarize the result by induction.** From the gradient of $\nabla_{\boldsymbol{A}}L(\boldsymbol{\theta})$ and $\nabla_{\boldsymbol{B}_j}L(\boldsymbol{\theta})$, we observe that non-zero gradients only emerge in the diagonal of $\boldsymbol{W}_{32}^{KQ}$ and $\boldsymbol{W}_{12}^{PV}$, and they are same. Therefore, the parameter matrices keep the same structure as the initial time. Thus we can summarize the dynamic system as the following.

$$\frac{\mathrm{d}}{\mathrm{d}\tau}a = -ab^2\left[(T-2)\kappa_2 + \sum_{t=2}^{T-1}\frac{1}{t-1}\kappa_3\right] + b(T-2)\kappa_1,$$

$$\frac{\mathrm{d}}{\mathrm{d}\tau}b = -a^2b\left[(T-2)\kappa_2 + \sum_{t=2}^{T-1}\frac{1}{t-1}\kappa_3\right] + a(T-2)\kappa_1.$$

which completes the proof. $\qquad\square$

### A.2.3 Proof of Lemma 5.3

For the reader's convenience, we restate the lemma as the following.

**Lemma A.3.** *Suppose that Assumption 4.1 holds and denote $(T-2)\kappa_2 + \sum_{t=2}^{T-1}\frac{1}{t-1}\kappa_3$ and $(T-2)\kappa_1$ by $c_1$ and $c_2$, respectively. Then, the dynamics in Lemma 5.2 are the same as those of gradient flow on the following objective function:*

$$\widetilde{\ell}(a,b) = \frac{1}{2c_1}(c_2 - c_1 ab)^2,$$

*whose global minimums satisfy $ab = c_2/c_1$.*

*Proof.* Basic calculus shows that

$$\frac{\partial}{\partial a}\widetilde{\ell}(a,b) = \frac{1}{2c_1}2(c_2 - c_1 ab)(-c_1 b) = -b(c_2 - c_1 ab) = -\frac{\mathrm{d}}{\mathrm{d}\tau}a,$$

$$\frac{\partial}{\partial b}\widetilde{\ell}(a,b) = \frac{1}{2c_1}2(c_2 - c_1 ab)(-c_1 a) = -a(c_2 - c_1 ab) = -\frac{\mathrm{d}}{\mathrm{d}\tau}b.$$

Therefore, the dynamics in Lemma 5.2 are the same as those of gradient flow on $\widetilde{\ell}(a,b)$, whose global minimums satisfy $ab = c_2/c_1$. □

### A.2.4 Proof of Lemma 5.4

For the reader's convenience, we restate the lemma as the following.

**Lemma A.4.** *Suppose that Assumption 4.1 holds, then $\widetilde{\ell}(a,b)$ is a non-convex function and satisfies the PL inequality as follows.*

$$\left|\frac{\partial}{\partial a}\widetilde{\ell}(a,b)\right|^2 + \left|\frac{\partial}{\partial b}\widetilde{\ell}(a,b)\right|^2 \geq 2c_1(a^2 + b^2)\left(\widetilde{\ell}(a,b) - \min_{a,b}\widetilde{\ell}(a,b)\right).$$

*Therefore, the gradient flow in Lemma 5.2 converges to the global minimum of $\widetilde{\ell}(a,b)$.*

*Proof.* First, we prove that $\widetilde{\ell}(a,b)$ is non-convex. The Hessian matrix of $\widetilde{\ell}(a,b)$ can be derived as follows.

$$\nabla^2\widetilde{\ell}(a,b) = \begin{pmatrix} c_1 b^2 & 2c_1 ab - c_2 \\ 2c_1 ab - c_2 & c_1 a^2 \end{pmatrix}.$$

Its determinant $c_1 a^2 b^2 - (2c_1 ab - c_2)^2 = (c_2 - c_1 ab)(3c_1 ab - c_2) < 0$ when $ab < \frac{c_2}{3c_1}$ or $ab > \frac{c_2}{c_1}$. Thus, $\widetilde{\ell}(a,b)$ is non-convex.

Besides, the PL inequality holds because

$$
\begin{aligned}
\left|\frac{\partial}{\partial a}\widetilde{\ell}(a,b)\right|^2 + \left|\frac{\partial}{\partial b}\widetilde{\ell}(a,b)\right|^2 &= b^2(c_2 - c_1 ab)^2 + a^2(c_2 - c_1 ab)^2 \\
&= (a^2 + b^2)(c_2 - c_1 ab)^2 \\
&= 2c_1(a^2 + b^2)\cdot\frac{1}{2c_1}(c_2 - c_1 ab)^2 \\
&= 2c_1(a^2 + b^2)\left(\widetilde{\ell}(a,b) - \min_{a,b}\widetilde{\ell}(a,b)\right) \\
&\geq 2c_1(a^2 + b^2)\left(\widetilde{\ell}(a,b) - \min_{a,b}\widetilde{\ell}(a,b)\right).
\end{aligned}
$$

□

### A.3 Proof of Corollary 4.1

For the reader's convenience, we restate the corollary as the following.

**Corollary A.1.** *We suppose that the same precondition of Theorem 4.1 holds. When predicting the $t$-th token, the trained transformer implements one step of gradient descent for the minimization of the OLS problem $L_{\mathrm{OLS}}(W) = \frac{1}{2}\sum_{i=1}^{t-1}\|x_{t+1} - Wx_t\|^2$, starting from the initialization $W = 0_{d\times d}$ with a step size $\frac{\widetilde{a}\widetilde{b}}{t-1}$.*

*Proof.* The proof is stem from the theoretical construction in [16]. First, we simplify the prediction $\widehat{y}_t$ as follows.

$$\widehat{y}_t = \begin{pmatrix} W_{12}^{PV} & W_{13}^{PV} \end{pmatrix}\frac{E_t^x E_t^{x*}}{\rho_t}\begin{pmatrix} W_{22}^{KQ} & W_{23}^{KQ} \\ W_{32}^{KQ} & W_{33}^{KQ} \end{pmatrix}e_t^x$$

$$= \begin{pmatrix} \widetilde{b}\boldsymbol{I}_d & \boldsymbol{0}_{d \times d} \end{pmatrix} \frac{\boldsymbol{E}_t^{\boldsymbol{x}} \boldsymbol{E}_t^{\boldsymbol{x}*}}{\rho_t} \begin{pmatrix} \boldsymbol{0}_{d \times d} & \boldsymbol{0}_{d \times d} \\ \widetilde{a}\boldsymbol{I}_d & \boldsymbol{0}_{d \times d} \end{pmatrix} \boldsymbol{e}_t^{\boldsymbol{x}}$$

$$= \frac{1}{\rho_t} \begin{pmatrix} \widetilde{b}\boldsymbol{I}_d & \boldsymbol{0}_{d \times d} \end{pmatrix} \sum_{i=1}^{t} \boldsymbol{e}_i^{\boldsymbol{x}} \boldsymbol{e}_i^{\boldsymbol{x}*} \begin{pmatrix} \boldsymbol{0}_{d \times d} & \boldsymbol{0}_{d \times d} \\ \widetilde{a}\boldsymbol{I}_d & \boldsymbol{0}_{d \times d} \end{pmatrix} \boldsymbol{e}_t^{\boldsymbol{x}}$$

$$= \frac{1}{\rho_t} \sum_{i=1}^{t} \widetilde{b}\boldsymbol{x}_i \begin{pmatrix} \widetilde{a}\boldsymbol{x}_{i-1}^* & \boldsymbol{0}_{d \times d} \end{pmatrix} \boldsymbol{e}_t^{\boldsymbol{x}}$$

$$= \frac{1}{\rho_t} \sum_{i=1}^{t} \widetilde{b}\boldsymbol{x}_i \widetilde{a}\boldsymbol{x}_{i-1}^* \boldsymbol{x}_t = \left( \frac{\widetilde{a}\widetilde{b}}{t-1} \sum_{i=1}^{t} \boldsymbol{x}_i \boldsymbol{x}_{i-1}^* \right) \boldsymbol{x}_t$$

$$= \left( \frac{\widetilde{a}\widetilde{b}}{t-1} \sum_{i=1}^{t-1} \boldsymbol{x}_{i+1} \boldsymbol{x}_i^* \right) \boldsymbol{x}_t.$$

Then, we connect it to the one step of gradient descent for the OLS problem $L_{\mathrm{OLS}}(\boldsymbol{W}) = \frac{1}{2} \sum_{i=1}^{t-1} \|\boldsymbol{x}_{i+1} - \boldsymbol{W}\boldsymbol{x}_i\|^2$.

$$\boldsymbol{W} - \frac{\widetilde{a}\widetilde{b}}{t-1} \nabla_{\boldsymbol{W}} \frac{1}{2} \sum_{i=1}^{t-1} \|\boldsymbol{x}_{i+1} - \boldsymbol{W}\boldsymbol{x}_i\|^2$$

$$= \boldsymbol{W} - \frac{\widetilde{a}\widetilde{b}}{t-1} \sum_{i=1}^{t-1} (\boldsymbol{x}_{i+1} - \boldsymbol{W}\boldsymbol{x}_i)(-\boldsymbol{x}_i^*)$$

$$= \boldsymbol{0} - \frac{\widetilde{a}\widetilde{b}}{t-1} \sum_{i=1}^{t-1} (\boldsymbol{x}_{i+1} - \boldsymbol{0}\boldsymbol{x}_i)(-\boldsymbol{x}_i^*)$$

$$= \frac{\widetilde{a}\widetilde{b}}{t-1} \sum_{i=1}^{t-1} \boldsymbol{x}_{i+1} \boldsymbol{x}_i^*.$$

Thus, the proof is completed. $\qquad\square$

## A.4 Proof of Proposition 4.1

For the reader's convenience, we restate the proposition as the following.

**Proposition A.1.** *Let $\mathcal{D}_{\boldsymbol{x}_1}$ be the multivariate normal distribution $\mathcal{N}(\boldsymbol{0}_d, \sigma^2 \boldsymbol{I}_d)$ with any $\sigma^2 > 0$, then the "simple" AR process can not be recovered by the trained transformer even in the ideal case with long training context. Formally, when the training sequence length $T_{tr}$ is large enough, for any test context length $T_{te}$ and dimension $j \in [d]$, the prediction from the trained transformer satisfies*

$$E_{x_1, W} \left[ \frac{(\widehat{y}_{T_{te}})_j}{(W x_{T_{te}})_j} \right] \to \frac{1}{5}.$$

*Therefore, the prediction $\widehat{\boldsymbol{y}}_{T_{te}}$ will not converges to the true next token $\boldsymbol{W}\boldsymbol{x}_{T_{te}}$.*

*Proof.* First, built upon the results in Theorem 4.1, when $T_{tr}$ is large enough, we have

$$\widetilde{a}\widetilde{b} = \frac{\kappa_1}{\kappa_2 + \frac{\kappa_3}{T_{tr}-2} \sum_{t=2}^{T_{tr}-1} \frac{1}{t-1}} \to \frac{\kappa_1}{\kappa_2} = \frac{\mathbb{E}[x_{1j}^4]}{\mathbb{E}[x_{1j}^6]} = \frac{3\sigma^4}{15\sigma^6} = \frac{1}{5\sigma^2}.$$

Second, by directly calculating, we have

$$(W x_{T_{te}})_j = (W^{T_{te}} x_1)_j = \lambda_j^{T_{te}} x_{1j},$$

and

$$(\widehat{y}_{T_{te}})_j = \frac{\widetilde{a}\widetilde{b}}{T_{te}-1} \sum_{i=1}^{T_{te}-1} \sum_{k=1}^{d} \lambda_j^i \lambda_k^{T_{te}-i} x_{1j} x_{1k}^2.$$

Therefore, we have

$$E_{x_1, W}\left[\frac{(\widehat{y}_{T_{te}})_j}{(W x_{T_{te}})_j}\right] = E_{x_1, W}\left[\frac{\widetilde{ab}}{T_{te} - 1} \sum_{i=1}^{T_{te}-1} \sum_{k=1}^{d} \lambda_j^{i-T_{te}} \lambda_k^{T_{te}-i} x_{1k}^2\right]$$

$$= E_{x_1}\left[\frac{\widetilde{ab}}{T_{te} - 1} \sum_{i=1}^{T_{te}-1} x_{1j}^2\right] = \widetilde{ab}\sigma^2.$$

Since $\widetilde{ab} < \frac{1}{5\sigma^2}$ and converges to $\frac{1}{5\sigma^2}$ when $T_{tr}$ is large enough, the proof is completed. $\square$

## A.5 Derivation of Example 4.1

*Proof.* We first prove that the example satisfies Assumption 4.1. Because only one element of $x_1$ sampled from Example 4.1 will be non-zero, we have $\mathbb{E}_{x_1 \sim \mathcal{D}_{x_1}}[x_{1i_1} x_{1i_2}^{r_2} \cdots x_{1i_n}^{r_n}] = \mathbb{E}_{x_1 \sim \mathcal{D}_{x_1}}[0] = 0$ for any subset $\{i_1, \ldots, i_n \mid n \leq 4\}$ of $[d]$, and $r_2, \ldots r_n \in \mathbb{N}$. In addition, for any $j \in [d]$, we can derive that

$$\kappa_1 = \mathbb{E}[x_{1j}^4] = \frac{1}{d} \cdot c^4 + \frac{d-1}{d} \cdot 0 = \frac{c^4}{d},$$

$$\kappa_2 = \mathbb{E}[x_{1j}^6] = \frac{1}{d} \cdot c^6 + \frac{d-1}{d} \cdot 0 = \frac{c^6}{d},$$

$$\kappa_3 = \sum_{r \neq j} \mathbb{E}[x_{1j}^2 x_{1r}^4] = 0.$$

Second, we prove that it satisfies Assumption 4.2 as follows. Without loss of general, we assume that the first coordinate of $x_1$ is $c$.

$$\frac{\kappa_1}{\kappa_2} \frac{\sum_{i=1}^{T_{te}-1} x_i x_i^*}{T_{te} - 1} x_{T_{te}} = \frac{1}{c^2} \text{diag}(c^2, 0, \ldots, 0)(\lambda_1^{T_{te}-1} c, 0, \ldots, 0)^\top$$

$$= (\lambda_1^{T_{te}-1} c, 0, \ldots, 0)^\top = x_{T_{te}}.$$

The proof is finished. $\square$

## A.6 Proof of Theorem 4.2

For the reader's convenience, we restate the theorem as the following.

**Theorem A.1.** *Suppose that Assumption 4.1 holds, then Assumption 4.2 is the sufficient and necessary condition for the trained transformer to learn the AR process. Formally, when the training sequence length $T_{tr}$ and test context length $T_{te}$ are large enough, the prediction from the trained transformer satisfies*

$$\widehat{y}_{T_{te}} \to W x_{T_{te}}, \quad T_{tr}, T_{te} \to +\infty.$$

*Proof.* First, built upon the results in Theorem 4.1, when $T_{tr}$ is large enough, we have

$$\widetilde{ab} = \frac{\kappa_1}{\kappa_2 + \frac{\kappa_3}{T_{tr}-2} \sum_{t=2}^{T_{tr}-1} \frac{1}{t-1}} \to \frac{\kappa_1}{\kappa_2}.$$

Second, when $T_{te}$ is large enough, by Assumption 4.2

$$\frac{\kappa_1}{\kappa_2} \frac{\sum_{i=1}^{T_{te}-1} x_i x_i^*}{T_{te} - 1} x_{T_{te}} \to x_{T_{te}}.$$

Therefore, we have

$$\widehat{y}_{T_{te}} = W\left(\widetilde{ab} \frac{\sum_{i=1}^{T_{te}-1} x_i x_i^*}{T_{te} - 1}\right) x_{T_{te}} \to_{T_{tr}} W\left(\frac{\kappa_1}{\kappa_2} \frac{\sum_{i=1}^{T_{te}-1} x_i x_i^*}{T_{te} - 1}\right) x_{T_{te}} \to_{T_{te}} W x_{T_{te}}$$

which finishes the proof. $\square$

## A.7 Proof of Theorem 4.3

For the reader's convenience, we restate the theorem as the following.

**Theorem A.2.** *Suppose the initialization satisfies Assumption 3.1, the initial token is fixed as $\mathbf{1}_d$, and we clip non-diagonal gradients of $\boldsymbol{W}_{32}^{KQ}$ and $\boldsymbol{W}_{12}^{PV}$ during the training, then the gradient flow of the one-layer linear transformer over the population AR loss converges to the same structure as the result in Theorem 4.1, with*

$$\widetilde{ab} = \frac{1}{1 + \frac{d-1}{T-2}\sum_{t=2}^{T-1}\frac{1}{t-1}}.$$

*Therefore, the obtained transformer performs one step of gradient descent in this case.*

The proof is similar to the proof of Theorem 4.1 in Appendix A.2. But the calculating for the gradients is more difficult than that of Theorem 4.1. Similarly, we first present and prove the following lemmas.

**Lemma A.5** (dynamical system of gradient flow). *Under the same assumption as in Theorem 4.3, the dynamical process of the parameters in the diagonal of $\boldsymbol{W}_{32}^{KQ}$ and $\boldsymbol{W}_{12}^{PV}$ satisfies*

$$\frac{\mathrm{d}}{\mathrm{d}\tau}a = -ab^2\left[(T-2) + \sum_{t=2}^{T-1}\frac{d-1}{t-1}\right] + b(T-2),$$

$$\frac{\mathrm{d}}{\mathrm{d}\tau}b = -a^2b\left[(T-2) + \sum_{t=2}^{T-1}\frac{d-1}{t-1}\right] + a(T-2),$$

*while the gradients for all other parameters were kept at zero during the training process.*

*Proof.* Recall that in Appendix A.2.2, we have already known that the population loss $L(\boldsymbol{\theta})$ in Eq. 2 can be rewritten as

$$L(\boldsymbol{\theta}) = \sum_{t=2}^{T-1} L_t(\boldsymbol{\theta}) = \sum_{t=2}^{T-1}\sum_{j=1}^{d}\mathbb{E}\left[\frac{1}{2}\widehat{y}_{t,j}^*\widehat{y}_{t,j} - \mathrm{Re}\left(x_{t+1,j}^*\widehat{y}_{t,j}\right) + \frac{1}{2}x_{t+1,j}^*x_{t+1,j}\right].$$

Besides, the derivatives of $L_t(\boldsymbol{\theta})$ with respect to $\mathrm{Vec}(\boldsymbol{A})$ and $\boldsymbol{B}_j$ are

$$\nabla_{\boldsymbol{B}_j}L_t(\boldsymbol{\theta}) = \frac{1}{2}\mathbb{E}\left[\nabla_{\boldsymbol{B}_j}\widehat{y}_{t,j}^*\widehat{y}_{t,j}\right] - \mathbb{E}\left[\nabla_{\boldsymbol{B}_j}\mathrm{Re}\left(x_{t+1,j}^*\widehat{y}_{t,j}\right)\right],$$

and

$$\nabla_{\mathrm{Vec}(\boldsymbol{A})}L_t(\boldsymbol{\theta}) = \frac{1}{2}\sum_{j=1}^{d}\mathbb{E}\left[\nabla_{\mathrm{Vec}(\boldsymbol{A})}\widehat{y}_{t,j}^*\widehat{y}_{t,j}\right] - \sum_{j=1}^{d}\mathbb{E}\left[\nabla_{\mathrm{Vec}(\boldsymbol{A})}\mathrm{Re}\left(x_{t+1,j}^*\widehat{y}_{t,j}\right)\right].$$

**Step one: calculate** $\mathbb{E}\left[\nabla_{\boldsymbol{B}_j}\mathrm{Re}\left(x_{t+1,j}^*\widehat{y}_{t,j}\right)\right]$. Similarly to the step one in Appendix A.2.2, the $\mathbb{E}\left[\nabla_{\boldsymbol{B}_j}\mathrm{Re}\left(x_{t+1,j}^*\widehat{y}_{t,j}\right)\right]$ can be derived as the following.

$$\mathbb{E}\left[\nabla_{\boldsymbol{B}_j}\mathrm{Re}\left(x_{t+1,j}^*\widehat{y}_{t,j}\right)\right] = \mathrm{Re}\left(\mathbb{E}\left[x_{t+1,j}^*\cdot e_t^{\boldsymbol{x}\top}\otimes\frac{\boldsymbol{E}_t^{\boldsymbol{x}}\boldsymbol{E}_t^{\boldsymbol{x}*}}{\rho_t}\cdot\mathrm{Vec}(\boldsymbol{A})\right]\right)$$

$$= \mathrm{Re}\left(\mathbb{E}\left[\lambda_j^{-t}\cdot\mathrm{Vec}(\frac{\boldsymbol{E}_t^{\boldsymbol{x}}\boldsymbol{E}_t^{\boldsymbol{x}*}}{\rho_t}\boldsymbol{A}e_t^{\boldsymbol{x}})\right]\right) \qquad \text{(use generating process)}$$

$$= \mathrm{Re}\left(\mathbb{E}\left[\lambda_j^{-t}\cdot(\frac{\boldsymbol{E}_t^{\boldsymbol{x}}\boldsymbol{E}_t^{\boldsymbol{x}*}}{\rho_t}\boldsymbol{A}e_t^{\boldsymbol{x}})\right]\right).$$

We note that for any $l \in [2d]$, based on the sparsity of $\boldsymbol{A}$, we have

$$
\left( \frac{\boldsymbol{E}_t^{\boldsymbol{x}} \boldsymbol{E}_t^{\boldsymbol{x}*}}{\rho_t} \boldsymbol{A} e_t^{\boldsymbol{x}} \right)_l = \left\{ \begin{array}{ll} \frac{a}{\rho_t} \sum_{i=1}^{t-1} \sum_{r=1}^{d} \lambda_l^i \lambda_r^{t-i}, & l \in [d], \\ \frac{a}{\rho_t} \sum_{i=1}^{t-1} \sum_{r=1}^{d} \lambda_{l-d}^{i-1} \lambda_r^{t-i}, & l \in [2d] - [d], \end{array} \right.
$$

which implies that

$$
\mathbb{E} \left[ \lambda_j^{-t} \cdot \left( \frac{\boldsymbol{E}_t^{\boldsymbol{x}} \boldsymbol{E}_t^{\boldsymbol{x}*}}{\rho_t} \boldsymbol{A} e_t^{\boldsymbol{x}} \right) \right] = \left\{ \begin{array}{ll} \frac{a}{\rho_t}(t-1), & l = j, \\ 0, & l \neq j, \end{array} \right.
$$

and the $l$-th element of $\mathbb{E} \left[ \nabla_{\boldsymbol{B}_j} \mathrm{Re} \left( x_{t+1,j}^* \widehat{y}_{t,j} \right) \right]$ is

$$
\mathbb{E} \left[ \nabla_{\boldsymbol{B}_j} \mathrm{Re} \left( x_{t+1,j}^* \widehat{y}_{t,j} \right) \right]_l = \left\{ \begin{array}{ll} \frac{a}{\rho_t}(t-1), & l = j, \\ 0, & l \neq j. \end{array} \right.
$$

**Step two: calculate $\mathbb{E} \left[ \nabla_{\boldsymbol{B}_j} \widehat{y}_{t,j}^* \widehat{y}_{t,j} \right]$.** Similarly to the step two in Appendix A.2.2, we can simplify the $\mathbb{E} \left[ \nabla_{\boldsymbol{B}_j} \widehat{y}_{t,j}^* \widehat{y}_{t,j} \right]$ as follows.

$$
\mathbb{E} \left[ \nabla_{\boldsymbol{B}_j} \widehat{y}_{t,j}^* \widehat{y}_{t,j} \right] = 2\mathrm{Re} \left( \mathbb{E} \left[ e_t^{\boldsymbol{x}\top} \otimes \frac{\boldsymbol{E}_t^{\boldsymbol{x}} \boldsymbol{E}_t^{\boldsymbol{x}*}}{\rho_t} \mathrm{Vec}(\boldsymbol{A}) \cdot \mathrm{Vec}^\top (\boldsymbol{A}) \overline{e_t^{\boldsymbol{x}}} \otimes \frac{\boldsymbol{E}_t^{\boldsymbol{x}} \boldsymbol{E}_t^{\boldsymbol{x}*}}{\rho_t} \cdot \boldsymbol{B}_j \right] \right)
$$

$$
= 2\mathrm{Re} \left( \mathbb{E} \left[ b \left( \frac{\overline{\boldsymbol{E}_t^{\boldsymbol{x}}} \boldsymbol{E}_t^{\boldsymbol{x}\top}}{\rho_t} \boldsymbol{A} \overline{e_t^{\boldsymbol{x}}} \right)_j \cdot \frac{\boldsymbol{E}_t^{\boldsymbol{x}} \boldsymbol{E}_t^{\boldsymbol{x}*}}{\rho_t} \boldsymbol{A} e_t^{\boldsymbol{x}} \right] \right). \qquad \text{(sparsity of } \boldsymbol{B})
$$

For any $j \in [d]$ and $l \in [2d]$, we can calculate that

$$
\left( \frac{\overline{\boldsymbol{E}_t^{\boldsymbol{x}}} \boldsymbol{E}_t^{\boldsymbol{x}\top}}{\rho_t} \boldsymbol{A} \overline{e_t^{\boldsymbol{x}}} \right)_j = \frac{a}{\rho_t} \sum_{i=1}^{t-1} \sum_{r=1}^{d} \lambda_j^{-i} \lambda_r^{i-t},
$$

and recall that

$$
\left( \frac{\boldsymbol{E}_t^{\boldsymbol{x}} \boldsymbol{E}_t^{\boldsymbol{x}*}}{\rho_t} \boldsymbol{A} e_t^{\boldsymbol{x}} \right)_l = \left\{ \begin{array}{ll} \frac{a}{\rho_t} \sum_{i=1}^{t-1} \sum_{r=1}^{d} \lambda_l^i \lambda_r^{t-i}, & l \in [d], \\ \frac{a}{\rho_t} \sum_{i=1}^{t-1} \sum_{r=1}^{d} \lambda_{l-d}^{i-1} \lambda_r^{t-i}, & l \in [2d] - [d]. \end{array} \right.
$$

With careful computing, we have

$$
\mathbb{E} \left[ b \left( \frac{\overline{\boldsymbol{E}_t^{\boldsymbol{x}}} \boldsymbol{E}_t^{\boldsymbol{x}\top}}{\rho_t} \boldsymbol{A} \overline{e_t^{\boldsymbol{x}}} \right)_j \cdot \left( \frac{\boldsymbol{E}_t^{\boldsymbol{x}} \boldsymbol{E}_t^{\boldsymbol{x}*}}{\rho_t} \boldsymbol{A} e_t^{\boldsymbol{x}} \right)_l \right] = \left\{ \begin{array}{ll} \frac{a^2 b}{\rho_t^2} \left[ (t-1)^2 + (d-1)(t-1) \right], & l = j, \\ \frac{a^2 b}{\rho_t^2}(t-1), & l \in [d] - j, \\ 0, & \text{otherwise,} \end{array} \right.
$$

which implies that the $l$-th element of $\frac{1}{2} \mathbb{E} \left[ \nabla_{\boldsymbol{B}_j} \widehat{y}_{t,j}^* \widehat{y}_{t,j} \right]$ is

$$
\frac{1}{2} \mathbb{E} \left[ \nabla_{\boldsymbol{B}_j} \widehat{y}_{t,j}^* \widehat{y}_{t,j} \right]_l = \left\{ \begin{array}{ll} \frac{a^2 b}{\rho_t^2} \left[ (t-1)^2 + (d-1)(t-1) \right], & l = j, \\ \frac{a^2 b}{\rho_t^2}(t-1), & l \in [d] - j, \\ 0, & \text{otherwise.} \end{array} \right.
$$

**Step three: calculate $\nabla_{\boldsymbol{B}_j} L_t(\boldsymbol{\theta})$ and $\nabla_{\boldsymbol{B}_j} L(\boldsymbol{\theta})$.** Based on steps one and two, the $l$-th element of $\nabla_{\boldsymbol{B}_j} L_t(\boldsymbol{\theta})$ can be derived as follows.

$$
\nabla_{\boldsymbol{B}_j} L_t(\boldsymbol{\theta})_l = \frac{1}{2} \mathbb{E} \left[ \nabla_{\boldsymbol{B}_j} \widehat{y}_{t,j}^* \widehat{y}_{t,j} \right]_l - \mathbb{E} \left[ \nabla_{\boldsymbol{B}_j} \mathrm{Re} \left( x_{t+1,j}^* \widehat{y}_{t,j} \right) \right]_l
$$

$$= \begin{cases} \frac{a^2 b}{\rho_t^2}\left[(t-1)^2 + (d-1)(t-1)\right] - \frac{a}{\rho_t}(t-1), & l = j, \\ \frac{a^2 b}{\rho_t^2}(t-1), & l \in [d] - j, \\ 0, & \text{otherwise.} \end{cases}$$

Furthermore, the $l$-th element of $\nabla_{\boldsymbol{B}_j} L(\boldsymbol{\theta})$ is

$$\nabla_{\boldsymbol{B}_j} L(\boldsymbol{\theta})_l = \sum_{t=2}^{T-1} \nabla_{\boldsymbol{B}_j} L_t(\boldsymbol{\theta})_l$$

$$= \begin{cases} \sum_{t=2}^{T-1}\left(\frac{a^2 b}{\rho_t^2}\left[(t-1)^2 + (d-1)(t-1)\right] - \frac{a}{\rho_t}(t-1)\right), & l = j, \\ \sum_{t=2}^{T-1} \frac{a^2 b}{\rho_t^2}(t-1), & l \in [d] - j, \\ 0, & \text{otherwise.} \end{cases} \tag{5}$$

If we clip the non-diagonal gradient of $\boldsymbol{W}_{12}^{PV}$, we have

$$\nabla_{\boldsymbol{B}_j} L(\boldsymbol{\theta})_l = \begin{cases} \sum_{t=2}^{T-1}\left(\frac{a^2 b}{\rho_t^2}\left[(t-1)^2 + (d-1)(t-1)\right] - \frac{a}{\rho_t}(t-1)\right), & l = j, \\ 0, & \text{otherwise,} \end{cases}$$

$$= \begin{cases} a^2 b\left[(T-2) + \sum_{t=2}^{T-1}\frac{d-1}{t-1}\right] - a(T-2), & l = j, \\ 0, & \text{otherwise.} \end{cases}$$

**Step four: calculate** $\mathbb{E}\left[\nabla_{\mathrm{Vec}(\boldsymbol{A})}\mathrm{Re}\left(x_{t+1,j}^* \widehat{y}_{t,j}\right)\right]$ **and** $\sum_{j=1}^{d} \mathbb{E}\left[\nabla_{\mathrm{Vec}(\boldsymbol{A})}\mathrm{Re}\left(x_{t+1,j}^* \widehat{y}_{t,j}\right)\right]$. Similarly to the step four in Appendix A.2.2, the $\mathbb{E}\left[\nabla_{\mathrm{Vec}(\boldsymbol{A})}\mathrm{Re}\left(x_{t+1,j}^* \widehat{y}_{t,j}\right)\right]$ can be derived as the following.

$$\mathbb{E}\left[\nabla_{\mathrm{Vec}(\boldsymbol{A})}\mathrm{Re}\left(x_{t+1,j}^* \widehat{y}_{t,j}\right)\right] = \mathrm{Re}\left(\mathbb{E}\left[x_{t+1,j}^* \cdot \boldsymbol{e}_t^{\boldsymbol{x}} \otimes \frac{\overline{\boldsymbol{E}_t^{\boldsymbol{x}}}\boldsymbol{E}_t^{\boldsymbol{x}\top}}{\rho_t} \cdot \boldsymbol{B}_j\right]\right)$$

$$= \mathrm{Re}\left(\mathbb{E}\left[\lambda_j^{-t} \cdot \mathrm{Vec}(\frac{\overline{\boldsymbol{E}_t^{\boldsymbol{x}}}\boldsymbol{E}_t^{\boldsymbol{x}\top}}{\rho_t}\boldsymbol{B}_j \boldsymbol{e}_t^{\boldsymbol{x}\top})\right]\right).$$

For any $s, r \in [2d]$, we have

$$\left(\frac{\overline{\boldsymbol{E}_t^{\boldsymbol{x}}}\boldsymbol{E}_t^{\boldsymbol{x}\top}}{\rho_t}\boldsymbol{B}_j \boldsymbol{e}_t^{\boldsymbol{x}\top}\right)_{sr} = \begin{cases} \frac{b}{\rho_t}(\sum_{i=1}^{t}\lambda_j^{i-1}\lambda_s^{1-i})\lambda_r^{t-1}, & s \in [d], r \in [d], \\ \frac{b}{\rho_t}(\sum_{i=1}^{t-1}\lambda_j^{i}\lambda_{s-d}^{1-i})\lambda_r^{t-1}, & s \in [2d] - [d], r \in [d], \\ \frac{b}{\rho_t}(\sum_{i=1}^{t}\lambda_j^{i-1}\lambda_s^{1-i})\lambda_{r-d}^{t-2}, & s \in [d], r \in [2d] - [d], \\ \frac{b}{\rho_t}(\sum_{i=1}^{t-1}\lambda_j^{i}\lambda_{s-d}^{1-i})\lambda_{r-d}^{t-2}, & s \in [2d] - [d], r \in [2d] - [d], \end{cases}$$

which implies that

$$\mathbb{E}\left[\lambda_j^{-t}\left(\frac{\overline{\boldsymbol{E}_t^{\boldsymbol{x}}}\boldsymbol{E}_t^{\boldsymbol{x}\top}}{\rho_t}\boldsymbol{B}_j \boldsymbol{e}_t^{\boldsymbol{x}\top}\right)_{sr}\right] = \begin{cases} \frac{b}{\rho_t}(t-1), & s = d+j, r = j, \\ \frac{b}{\rho_t}, & s \neq d+j, r = j, \\ 0, & \text{otherwise.} \end{cases}$$

and the $(s, r)$-th element of $\mathbb{E}\left[\nabla_{\boldsymbol{A}}\mathrm{Re}\left(x_{t+1,j}^* \widehat{y}_{t,j}\right)\right]$ is

$$\mathbb{E}\left[\nabla_{\boldsymbol{A}}\mathrm{Re}\left(x_{t+1,j}^* \widehat{y}_{t,j}\right)\right]_{sr} = \begin{cases} \frac{b}{\rho_t}(t-1), & s = d+j, r = j, \\ \frac{b}{\rho_t}, & s \neq d+j, r = j, \\ 0, & \text{otherwise.} \end{cases}$$

Finally, we can calculate $\sum_{j=1}^{d} \mathbb{E}\left[\nabla_{\boldsymbol{A}} \mathrm{Re}\left(x_{t+1,j}^{*} \widehat{y}_{t,j}\right)\right]$ as

$$\sum_{j=1}^{d} \mathbb{E}\left[\nabla_{\boldsymbol{A}} \mathrm{Re}\left(x_{t+1,j}^{*} \widehat{y}_{t,j}\right)\right]_{sr} = \begin{cases} \frac{b}{\rho_t}(t-1), & s-d=r, A_{sr} \in \boldsymbol{W}_{32}^{KQ}, \\ \frac{b}{\rho_t}, & s-d \neq r, A_{sr} \in \boldsymbol{W}_{32}^{KQ}, \\ 0, & \text{otherwise.} \end{cases}$$

**Step five: calculate $\mathbb{E}\left[\nabla_{\mathrm{Vec}(\boldsymbol{A})} \widehat{y}_{t,j}^{*} \widehat{y}_{t,j}\right]$ and $\sum_{j=1}^{d} \mathbb{E}\left[\nabla_{\mathrm{Vec}(\boldsymbol{A})} \widehat{y}_{t,j}^{*} \widehat{y}_{t,j}\right]$.** Similarly to the step five in Appendix A.2.2, the $\mathbb{E}\left[\nabla_{\mathrm{Vec}(\boldsymbol{A})} \widehat{y}_{t,j}^{*} \widehat{y}_{t,j}\right]$ is simplified as follows.

$$\mathbb{E}\left[\nabla_{\mathrm{Vec}(\boldsymbol{A})} \widehat{y}_{t,j}^{*} \widehat{y}_{t,j}\right] = 2\mathrm{Re}\left(\mathbb{E}\left[\boldsymbol{e}_t^{\boldsymbol{x}} \otimes \frac{\overline{\boldsymbol{E}_t^{\boldsymbol{x}} \boldsymbol{E}_t^{\boldsymbol{x}\top}}}{\rho_t} \boldsymbol{B}_j \cdot \boldsymbol{B}_j^{\top} \cdot \boldsymbol{e}_t^{\boldsymbol{x}*} \otimes \frac{\overline{\boldsymbol{E}_t^{\boldsymbol{x}} \boldsymbol{E}_t^{\boldsymbol{x}\top}}}{\rho_t} \cdot \mathrm{Vec}(\boldsymbol{A})\right]\right)$$

$$= 2\mathrm{Re}\left(\mathbb{E}\left[\sum_{k=d+1}^{2d} a\left(\frac{\boldsymbol{E}_t^{\boldsymbol{x}} \boldsymbol{E}_t^{\boldsymbol{x}*}}{\rho_t} \boldsymbol{B}_j \boldsymbol{e}_t^{\boldsymbol{x}*}\right)_{k,k-d} \cdot \mathrm{Vec}(\frac{\overline{\boldsymbol{E}_t^{\boldsymbol{x}} \boldsymbol{E}_t^{\boldsymbol{x}\top}}}{\rho_t} \boldsymbol{B}_j \boldsymbol{e}_t^{\boldsymbol{x}\top})\right]\right).$$

For any $k \in [2d] - [d]$, we can calculate that

$$\left(\frac{\boldsymbol{E}_t^{\boldsymbol{x}} \boldsymbol{E}_t^{\boldsymbol{x}*}}{\rho_t} \boldsymbol{B}_j \boldsymbol{e}_t^{\boldsymbol{x}*}\right)_{k,k-d} = \frac{b}{\rho_t}(\sum_{i=1}^{t-1} \lambda_j^{-i} \lambda_{k-d}^{i-1})\lambda_{k-d}^{1-t} = \frac{b}{\rho_t} \sum_{i=1}^{t-1} \lambda_j^{-i} \lambda_{k-d}^{i-t},$$

and recall that for any $s, r \in [2d]$, we have

$$\left(\frac{\overline{\boldsymbol{E}_t^{\boldsymbol{x}} \boldsymbol{E}_t^{\boldsymbol{x}\top}}}{\rho_t} \boldsymbol{B}_j \boldsymbol{e}_t^{\boldsymbol{x}\top}\right)_{sr} = \begin{cases} \frac{b}{\rho_t}(\sum_{i=1}^{t} \lambda_j^{i-1} \lambda_s^{1-i})\lambda_r^{t-1}, & s \in [d], r \in [d], \\ \frac{b}{\rho_t}(\sum_{i=1}^{t-1} \lambda_j^{i} \lambda_{s-d}^{1-i})\lambda_r^{t-1}, & s \in [2d] - [d], r \in [d], \\ \frac{b}{\rho_t}(\sum_{i=1}^{t} \lambda_j^{i-1} \lambda_s^{1-i})\lambda_{r-d}^{t-2}, & s \in [d], r \in [2d] - [d], \\ \frac{b}{\rho_t}(\sum_{i=1}^{t-1} \lambda_j^{i} \lambda_{s-d}^{1-i})\lambda_{r-d}^{t-2}, & s \in [2d] - [d], r \in [2d] - [d]. \end{cases}$$

With careful computing, we have

$$\mathbb{E}\left[\sum_{k=d+1}^{2d} a\left(\frac{\boldsymbol{E}_t^{\boldsymbol{x}} \boldsymbol{E}_t^{\boldsymbol{x}*}}{\rho_t} \boldsymbol{B}_j \boldsymbol{e}_t^{\boldsymbol{x}*}\right)_{k,k-d} \cdot \left(\frac{\overline{\boldsymbol{E}_t^{\boldsymbol{x}} \boldsymbol{E}_t^{\boldsymbol{x}\top}}}{\rho_t} \boldsymbol{B}_j \boldsymbol{e}_t^{\boldsymbol{x}\top}\right)_{sr}\right]$$

$$= \begin{cases} \frac{ab^2}{\rho_{t_2}}(t-1)^2, & s = d+j, r = j, \\ \frac{ab^2}{\rho_{t_2}}(t-1), & s \neq d+j, r = j, \\ \frac{ab^2}{\rho_{t_2}}(t-1), & s = d+r, r \neq j, \\ \frac{ab^2}{\rho_{t_2}}(t-1), & s = d+j, r \neq j, \\ \frac{ab^2}{\rho_t^2}, & \text{remains in } \boldsymbol{W}_{32}^{KQ}, \\ 0, & \text{otherwise,} \end{cases}$$

which implies that the $l$-th element of $\frac{1}{2}\mathbb{E}\left[\nabla_{\boldsymbol{A}} \widehat{y}_{t,j}^{*} \widehat{y}_{t,j}\right]$ is

$$\frac{1}{2}\mathbb{E}\left[\nabla_{\boldsymbol{A}} \widehat{y}_{t,j}^{*} \widehat{y}_{t,j}\right] = \begin{cases} \frac{ab^2}{\rho_{t_2}}(t-1)^2, & s = d+j, r = j, \\ \frac{ab^2}{\rho_{t_2}}(t-1), & s \neq d+j, r = j, \\ \frac{ab^2}{\rho_{t_2}}(t-1), & s = d+r, r \neq j, \\ \frac{ab^2}{\rho_{t_2}}(t-1), & s = d+j, r \neq j, \\ \frac{ab^2}{\rho_t^2}, & \text{remains in } \boldsymbol{W}_{32}^{KQ}, \\ 0, & \text{otherwise.} \end{cases}$$

Based on these results, we can derive

$$
\frac{1}{2}\sum_{j=1}^{d}\mathbb{E}\Big[\nabla_{\boldsymbol{A}}\widehat{y}_{t,j}^{*}\,\widehat{y}_{t,j}\Big]=
\begin{cases}
\frac{ab^2}{\rho_{t}^2}\big[(t-1)^2+(d-1)(t-1)\big], & s-d=r, A_{sr}\in\boldsymbol{W}_{32}^{KQ},\\
\frac{ab^2}{\rho_{t}^2}\big[2(t-1)+d-2\big], & s-d\neq r, A_{sr}\in\boldsymbol{W}_{32}^{KQ},\\
0, & \text{otherwise.}
\end{cases}
$$

**Step six: calculate $\nabla_{\mathrm{Vec}(\boldsymbol{A})}L_t(\boldsymbol{\theta})$ and $\nabla_{\mathrm{Vec}(\boldsymbol{A})}L(\boldsymbol{\theta})$.** Based on steps four and five, the $(s,r)$-th element of $\nabla_{\boldsymbol{A}}L_t(\boldsymbol{\theta})$ can be derived as follows.

$$
\nabla_{\boldsymbol{A}}L_t(\boldsymbol{\theta})_{sr}=\frac{1}{2}\sum_{j=1}^{d}\mathbb{E}\Big[\nabla_{\boldsymbol{A}}\widehat{y}_{t,j}^{*}\,\widehat{y}_{t,j}\Big]-\sum_{j=1}^{d}\mathbb{E}\Big[\nabla_{\boldsymbol{A}}\mathrm{Re}\left(x_{t+1,j}^{*}\,\widehat{y}_{t,j}\right)\Big]
$$

$$
=
\begin{cases}
\frac{ab^2}{\rho_{t}^2}\big[(t-1)^2+(d-1)(t-1)\big]-\frac{b}{\rho_t}(t-1), & s-d=r, A_{sr}\in\boldsymbol{W}_{32}^{KQ},\\
\frac{ab^2}{\rho_{t}^2}\big[2(t-1)+d-2\big]-\frac{b}{\rho_t}, & s-d\neq r, A_{sr}\in\boldsymbol{W}_{32}^{KQ},\\
0, & \text{otherwise.}
\end{cases}
$$

Furthermore, the $(s,r)$-th element of $\nabla_{\boldsymbol{A}}L(\boldsymbol{\theta})$ is

$$
\nabla_{\boldsymbol{A}}L(\boldsymbol{\theta})_{sr}=\sum_{t=2}^{T-1}\nabla_{\boldsymbol{A}}L_t(\boldsymbol{\theta})_{sr}
$$

$$
=
\begin{cases}
\sum_{t=2}^{T-1}\left(\frac{ab^2}{\rho_t^2}\big[(t-1)^2+(d-1)(t-1)\big]-\frac{b}{\rho_t}(t-1)\right), & s-d=r, A_{sr}\in\boldsymbol{W}_{32}^{KQ},\\
\sum_{t=2}^{T-1}\left[\frac{ab^2}{\rho_t^2}\big[2(t-1)+d-2\big]-\frac{b}{\rho_t}\right], & s-d\neq r, A_{sr}\in\boldsymbol{W}_{32}^{KQ},\\
0, & \text{otherwise.}
\end{cases}
\tag{6}
$$

If we clip the non-diagonal gradient of $\boldsymbol{W}_{32}^{KQ}$, we have

$$
\nabla_{\boldsymbol{A}}L(\boldsymbol{\theta})_{sr}=\sum_{t=2}^{T-1}\nabla_{\boldsymbol{A}}L_t(\boldsymbol{\theta})_{sr}
$$

$$
=
\begin{cases}
\sum_{t=2}^{T-1}\left(\frac{ab^2}{\rho_t^2}\big[(t-1)^2+(d-1)(t-1)\big]-\frac{b}{\rho_t}(t-1)\right), & s-d=r, A_{sr}\in\boldsymbol{W}_{32}^{KQ},\\
0, & \text{otherwise,}
\end{cases}
$$

$$
=
\begin{cases}
ab^2\Big[(T-2)+\sum_{t=2}^{T-1}\frac{d-1}{t-1}\Big]-b(T-2), & s-d=r, A_{sr}\in\boldsymbol{W}_{32}^{KQ},\\
0, & \text{otherwise.}
\end{cases}
$$

**Step seven: summarize the result by induction.** From the gradient of $\nabla_{\boldsymbol{A}}L(\boldsymbol{\theta})$ and $\nabla_{\boldsymbol{B}_j}L(\boldsymbol{\theta})$, we observe that non-zero gradients only emerge in the diagonal of $\boldsymbol{W}_{32}^{KQ}$ and $\boldsymbol{W}_{12}^{PV}$, and they are same. Therefore, the parameter matrices keep the same structure as the initial time. Thus we can summarize the dynamic system as the following.

$$
\frac{\mathrm{d}}{\mathrm{d}\tau}a=-ab^2\left[(T-2)+\sum_{t=2}^{T-1}\frac{d-1}{t-1}\right]+b(T-2),
$$

$$
\frac{\mathrm{d}}{\mathrm{d}\tau}b=-a^2b\left[(T-2)+\sum_{t=2}^{T-1}\frac{d-1}{t-1}\right]+a(T-2),
$$

which completes the proof.

$\square$

**Lemma A.6.** *Suppose that the precondtions of Theorem 4.3 hold, and denote $(T-2)+\sum_{t=2}^{T-1}\frac{d-1}{t-1}$ and $T-2$ by $c_1$ and $c_2$, respectively. Then, the dynamics are the same as those of gradient flow on the following objective function:*

$$
\widetilde{\ell}(a,b)=\frac{1}{2c_1}(c_2-c_1ab)^2,
$$

*whose global minimums satisfy $ab=c_2/c_1$.*

Table 1: Step size in different simulations.

| $\boldsymbol{x}_1$ | $\sigma/c$ | step size |
|---|---|---|
| | 0.5 | 0.001 |
| Gaussian | 1 | 0.0001 |
| | 2 | 0.000002 |
| | 0.5 | 0.03 |
| Example 4.1 | 1 | 0.001 |
| | 2 | 0.0001 |
| $\mathbf{1}_d$ | - | 0.0005 |

*Proof.* The proof is the same as that of Lemma 5.3 in Appendix A.2.3. □

Using the results from the above lemmas and Lemma 5.4, we can conclude Theorem 4.3.

### A.8 Proof of Proposition 4.2

For the reader's convenience, we restate the proposition as the following.

**Proposition A.2.** *The limiting point found by the gradient does not share the same structure as that in Theorem 4.1, thus the trained transformer will not implement one step of gradient descent for minimizing $\frac{1}{2}\sum_{i=1}^{t-1}\|\boldsymbol{x}_{i+1} - W\boldsymbol{x}_i\|^2$.*

*Proof.* From Eq. 6 and Eq. 5, we know that when the parameters matrices share the same structure as the result in Theorem 4.1, the non-zero gradients of the non-diagonal elements of $\boldsymbol{W}_{32}^{KQ}$ and $\boldsymbol{W}_{12}^{PV}$ will occur, which implies the result. □

## Appendix B  Experimental details and additional results

### B.1  GPU and random seed

The random seed in the experiments is fixed as 1. All experiments are done on a single GeForce RTX 3090 GPU in one hour.

### B.2  Step size in simulations

The step size of the gradient descent in different simulations is summarized in Table 1.

### B.3  Additional results for Gaussian initial token

In practice, we estimate the ratio of two vectors by calculating the mean of the element-wise divide between two vectors. The results for $\sigma = 1, 2$ and $T = 100$ with diagonal initialization are presented in Figure 2. The results for $\sigma = 1$ and $T = 5$ (small-context scenarios) with diagonal initialization are presented in Figure 4. The results for $\sigma = 1$ and $T = 100$ with Gaussian initialization ($\sigma_w = 0.001, 0.01, 0.1$) are presented in Figure 5.

### B.4  Additional results for Sparse initial token (Example 4.1)

The results for $c = 1, 2$ and $T = 100$ with diagonal initialization are presented in Figure 3. The results for $c = 1, 2$ and $T = 100$ with Gaussian initialization ($\sigma_w = 0.001, 0.01, 0.1$) are presented in Figure 6.

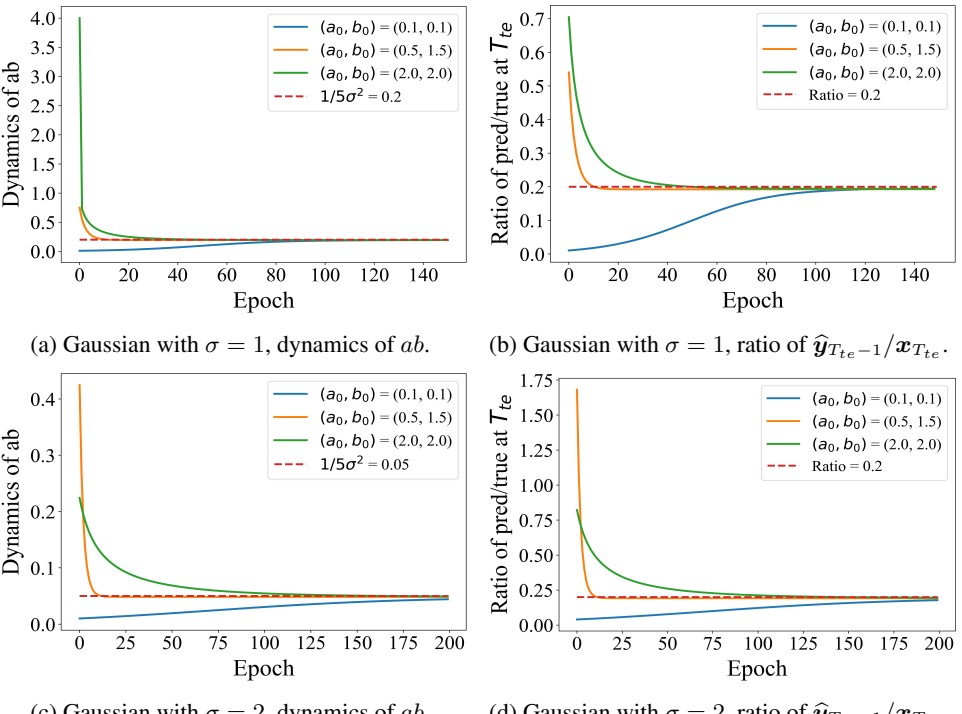

(a) Gaussian with $\sigma = 1$, dynamics of $ab$.     (b) Gaussian with $\sigma = 1$, ratio of $\widehat{\boldsymbol{y}}_{T_{te}-1}/\boldsymbol{x}_{T_{te}}$.

(c) Gaussian with $\sigma = 2$, dynamics of $ab$.     (d) Gaussian with $\sigma = 2$, ratio of $\widehat{\boldsymbol{y}}_{T_{te}-1}/\boldsymbol{x}_{T_{te}}$.

Figure 2: Simulations results in Gaussian initial token with $\sigma = 1, 2$. The results show that the convergence of $ab$ satisfies Theorem 4.1. In addition, the trained transformer can not recover the Gaussian initial token, which verifies Proposition 4.1.

## B.5   Additional results for full-one initial token

$\boldsymbol{W}^{KQ}$ and $\boldsymbol{W}^{PV}$ with different diagonal initializations are presented in Figure 7. From the results, we know that the gradient descent converges to

$$
\begin{pmatrix} \widetilde{\boldsymbol{W}_{22}^{KQ}} & \widetilde{\boldsymbol{W}_{23}^{KQ}} \\ \boldsymbol{W}_{32}^{KQ} & \boldsymbol{W}_{33}^{KQ} \end{pmatrix} = \begin{pmatrix} \boldsymbol{0}_{d\times d} & \boldsymbol{0}_{d\times d} \\ \boldsymbol{W}_1 & \boldsymbol{0}_{d\times d} \end{pmatrix}, \left( \widetilde{\boldsymbol{W}_{12}^{PV}} \quad \widetilde{\boldsymbol{W}_{13}^{PV}} \right) = \left( \boldsymbol{W}_2 \quad \boldsymbol{0}_{d\times d} \right),
$$

where $\boldsymbol{W}_1$ and $\boldsymbol{W}_2$ are some dense matrices. Similarly to the proof of Corollary 4.1 in Appendix 4.1, we have

$$
\begin{aligned}
\widehat{\boldsymbol{y}}_t &= \left( \boldsymbol{W}_{12}^{PV} \quad \boldsymbol{W}_{13}^{PV} \right) \frac{\boldsymbol{E}_t^{\boldsymbol{x}} \boldsymbol{E}_t^{\boldsymbol{x}*}}{\rho_t} \begin{pmatrix} \boldsymbol{W}_{22}^{KQ} & \boldsymbol{W}_{23}^{KQ} \\ \boldsymbol{W}_{32}^{KQ} & \boldsymbol{W}_{33}^{KQ} \end{pmatrix} \boldsymbol{e}_t^{\boldsymbol{x}} \\
&= \left( \boldsymbol{W}_2 \quad \boldsymbol{0}_{d\times d} \right) \frac{\boldsymbol{E}_t^{\boldsymbol{x}} \boldsymbol{E}_t^{\boldsymbol{x}*}}{\rho_t} \begin{pmatrix} \boldsymbol{0}_{d\times d} & \boldsymbol{0}_{d\times d} \\ \boldsymbol{W}_1 & \boldsymbol{0}_{d\times d} \end{pmatrix} \boldsymbol{e}_t^{\boldsymbol{x}} \\
&= \frac{1}{\rho_t} \left( \boldsymbol{W}_2 \quad \boldsymbol{0}_{d\times d} \right) \sum_{i=1}^{t} \boldsymbol{e}_i^{\boldsymbol{x}} \boldsymbol{e}_i^{\boldsymbol{x}*} \begin{pmatrix} \boldsymbol{0}_{d\times d} & \boldsymbol{0}_{d\times d} \\ \boldsymbol{W}_1 & \boldsymbol{0}_{d\times d} \end{pmatrix} \boldsymbol{e}_t^{\boldsymbol{x}} \\
&= \frac{1}{\rho_t} \sum_{i=1}^{t} \boldsymbol{W}_2 \boldsymbol{x}_i \left( \boldsymbol{x}_{i-1}^* \boldsymbol{W}_1 \quad \boldsymbol{0}_{d\times d} \right) \boldsymbol{e}_t^{\boldsymbol{x}} \\
&= \frac{1}{\rho_t} \sum_{i=1}^{t} \boldsymbol{W}_2 \boldsymbol{x}_i \boldsymbol{x}_{i-1}^* \boldsymbol{W}_1 \boldsymbol{x}_t.
\end{aligned}
$$

which can be seen as a somewhat preconditioned gradient descent on the OLS problem.

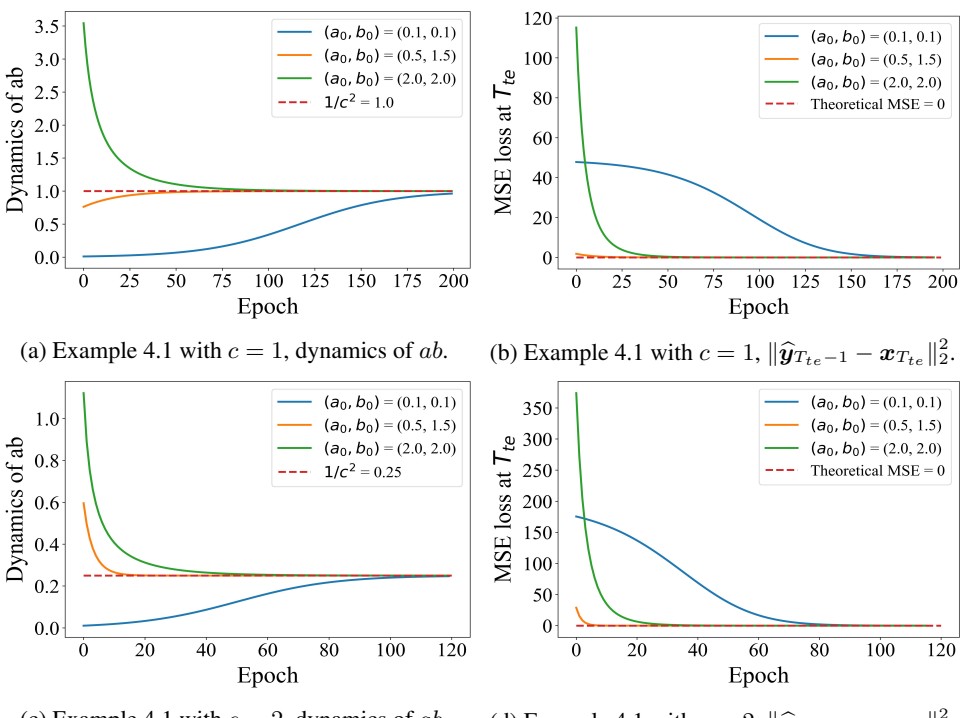

(a) Example 4.1 with $c = 1$, dynamics of $ab$.

(b) Example 4.1 with $c = 1$, $\|\widehat{\boldsymbol{y}}_{T_{te}-1} - \boldsymbol{x}_{T_{te}}\|_2^2$.

(c) Example 4.1 with $c = 2$, dynamics of $ab$.

(d) Example 4.1 with $c = 2$, $\|\widehat{\boldsymbol{y}}_{T_{te}-1} - \boldsymbol{x}_{T_{te}}\|_2^2$.

Figure 3: Simulations results on Example 4.1. Results show that the convergence of $ab$ satisfies Theorem 4.1. In addition, the trained transformer can recover the sequence with the initial token from Example 4.1, which verifies Theorem 4.2.

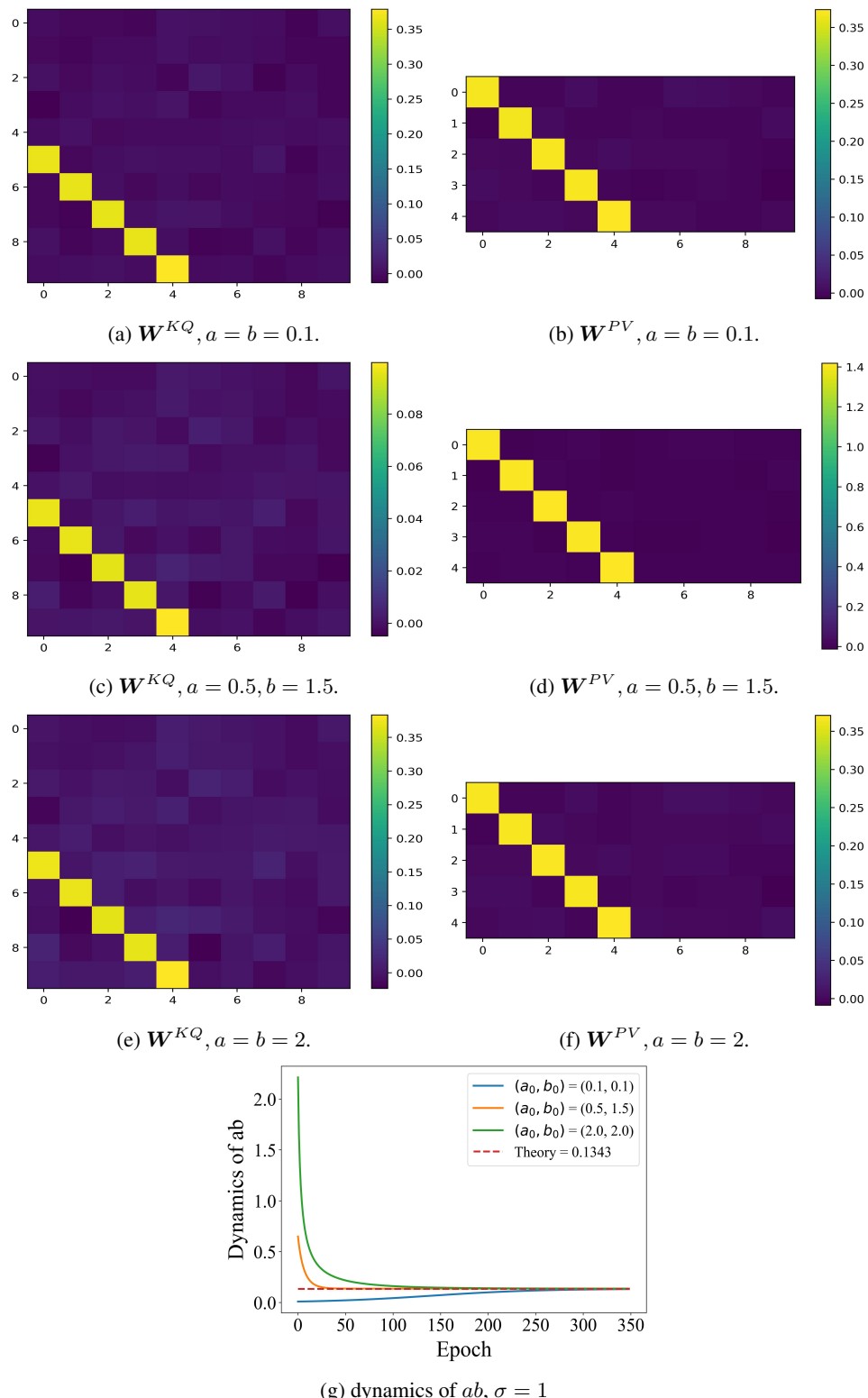

(a) $\boldsymbol{W}^{KQ}, a = b = 0.1$.

(b) $\boldsymbol{W}^{PV}, a = b = 0.1$.

(c) $\boldsymbol{W}^{KQ}, a = 0.5, b = 1.5$.

(d) $\boldsymbol{W}^{PV}, a = 0.5, b = 1.5$.

(e) $\boldsymbol{W}^{KQ}, a = b = 2$.

(f) $\boldsymbol{W}^{PV}, a = b = 2$.

(g) dynamics of $ab$, $\sigma = 1$

Figure 4: Results of small-context scenarios ($d = T = 5$) with Gaussian start point ($\sigma = 1$) and diagonal initialization, and the theoretical limit is $1/(5\sigma^2 + \frac{d-1}{T-2}\sigma^2 \sum_{t=2}^{T-1} \frac{1}{t-1})$ by Theorem 4.1. The read blocks in Assumption 3.1 are presented, which are related to the final prediction. The product $ab$ does converge to the results in Theorem 4.1.

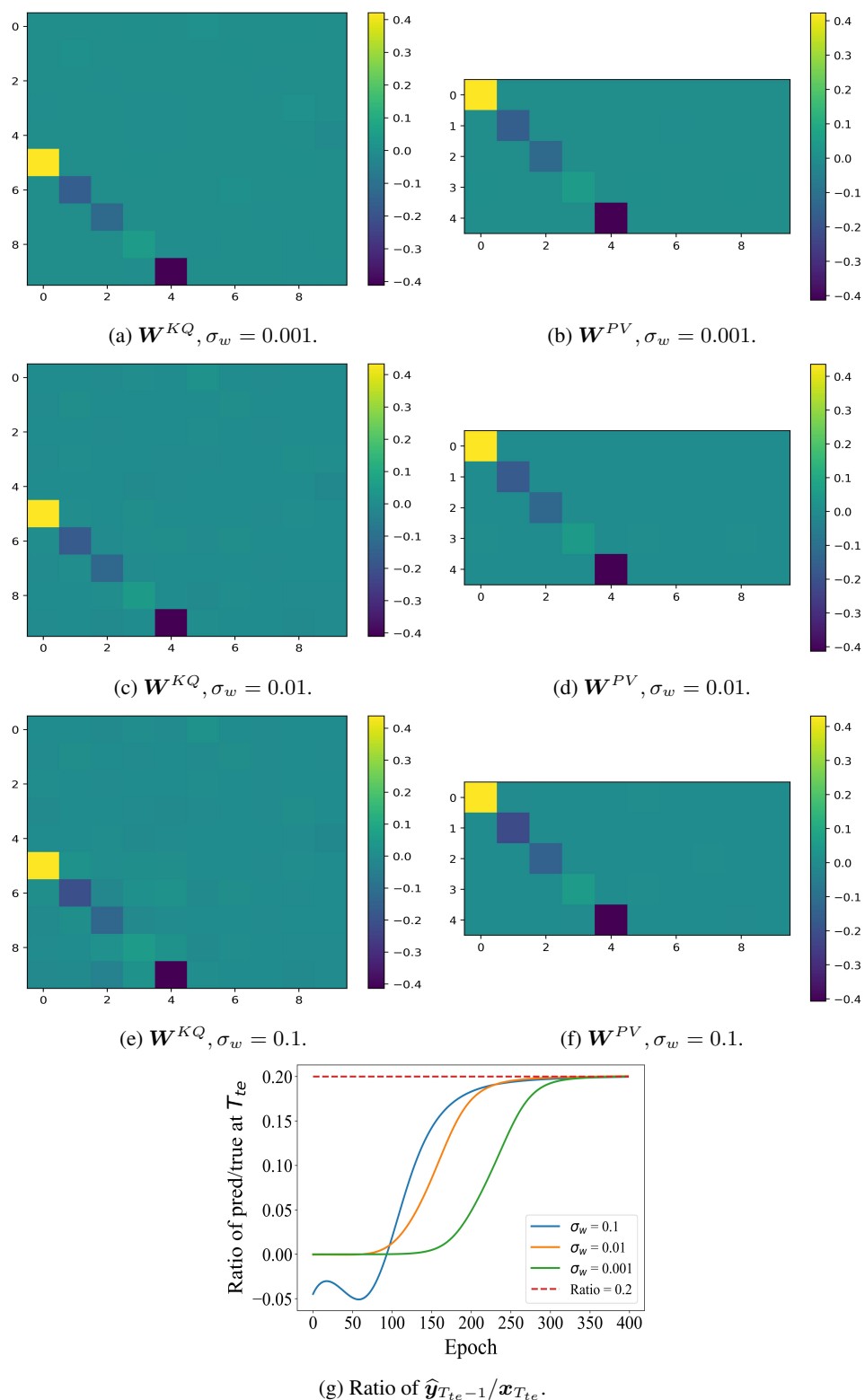

(a) $\boldsymbol{W}^{KQ}, \sigma_w = 0.001.$

(b) $\boldsymbol{W}^{PV}, \sigma_w = 0.001.$

(c) $\boldsymbol{W}^{KQ}, \sigma_w = 0.01.$

(d) $\boldsymbol{W}^{PV}, \sigma_w = 0.01.$

(e) $\boldsymbol{W}^{KQ}, \sigma_w = 0.1.$

(f) $\boldsymbol{W}^{PV}, \sigma_w = 0.1.$

(g) Ratio of $\widehat{\boldsymbol{y}}_{T_{te}-1}/\boldsymbol{x}_{T_{te}}.$

Figure 5: Results of Gaussian start point ($\sigma = 1$) and standard Gaussian initialization with different variance $\sigma_w$. The read blocks in Assumption 3.1 are presented, which are related to the final prediction. The parameter matrices retain the same strong diagonal structure and test performance as those of the diagonal initialization.

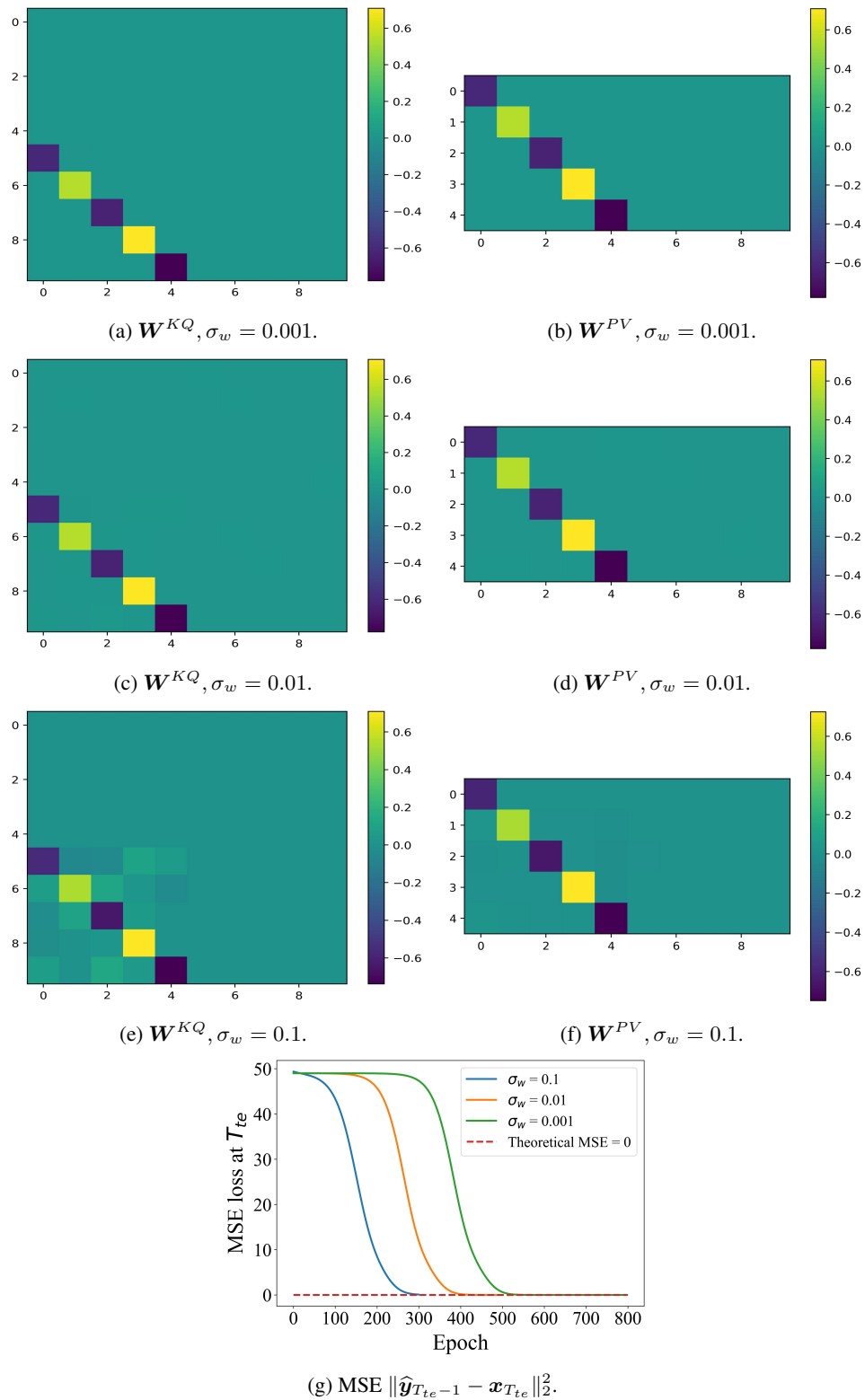

(a) $\boldsymbol{W}^{KQ}, \sigma_w = 0.001$.

(b) $\boldsymbol{W}^{PV}, \sigma_w = 0.001$.

(c) $\boldsymbol{W}^{KQ}, \sigma_w = 0.01$.

(d) $\boldsymbol{W}^{PV}, \sigma_w = 0.01$.

(e) $\boldsymbol{W}^{KQ}, \sigma_w = 0.1$.

(f) $\boldsymbol{W}^{PV}, \sigma_w = 0.1$.

(g) MSE $\|\widehat{\boldsymbol{y}}_{T_{te}-1} - \boldsymbol{x}_{T_{te}}\|_2^2$.

Figure 6: Results of Example 4.1 ($c = 1$) and standard Gaussian initialization with different variance $\sigma_w$. The read blocks in Assumption 3.1 are presented, which are related to the final prediction. The parameter matrices retain the same strong diagonal structure and test performance as those of the diagonal initialization.

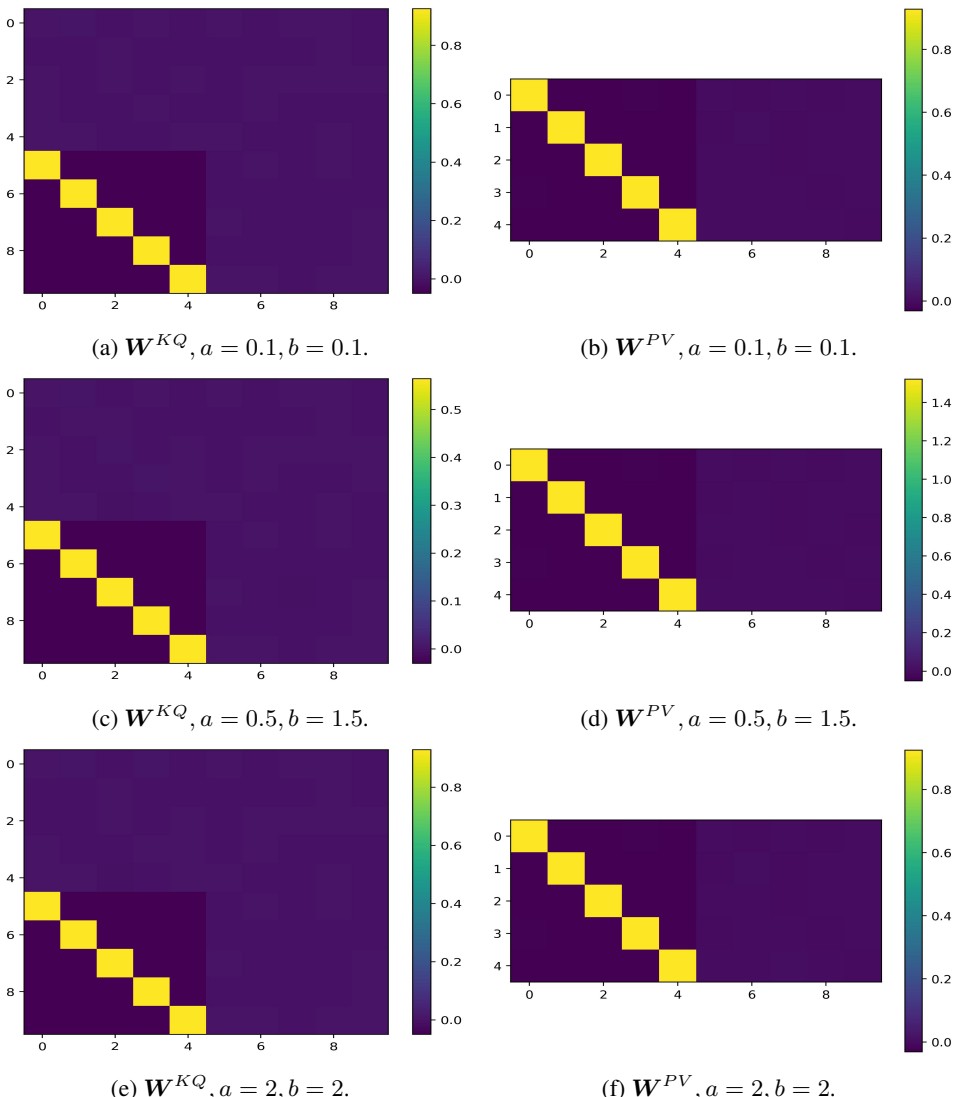

(a) $\boldsymbol{W}^{KQ}, a = 0.1, b = 0.1.$

(b) $\boldsymbol{W}^{PV}, a = 0.1, b = 0.1.$

(c) $\boldsymbol{W}^{KQ}, a = 0.5, b = 1.5.$

(d) $\boldsymbol{W}^{PV}, a = 0.5, b = 1.5.$

(e) $\boldsymbol{W}^{KQ}, a = 2, b = 2.$

(f) $\boldsymbol{W}^{PV}, a = 2, b = 2.$

Figure 7: $\boldsymbol{W}^{KQ}$ and $\boldsymbol{W}^{PV}$ of full-one start points with different diagonal initialization. The read blocks in Assumption 3.1 are presented, which are related to the final prediction.

