# OpenReview forum: "On Mesa-Optimization in Autoregressively Trained Transformers: Emergence and Capability"
_NeurIPS.cc/2024/Conference — NeurIPS 2024 poster_

### Official Review · Reviewer_c1Lc · 2024-06-19

**Soundness:** 3
**Presentation:** 2
**Contribution:** 2
**Rating:** 5
**Confidence:** 4

**Summary:**

This paper studies the behavior of a single unit of linear self-attention trained via gradient flow on an autoregressive (AR) loss on token sequences generated by a particular class of noiseless (deterministic) linear systems. The first theoretical results state that under some assumptions, the result of gradient flow from a particular class of initializations is a model whose in-context learning (ICL) predictions can be interpreted as those of another model that has been updated with one step of gradient descent (GD) using the context examples. The second part of the theoretical contribution is to say that a stronger assumption is necessary and sufficient for the result of this GD to give accurate predictions. These results are supported by numerical simulations of the studied setting.

**Strengths:**

- This paper aims to address an important issue in the ICL theory literature: most works study the emergence of ICL in transformers trained on an ICL objective (referred to as few-shot pertaining in the paper), whereas in practice transformers are trained on an autoregressive objective.
- The theoretical results are correct as far as I can tell from skimming the proofs. The notations are clearly defined and consistent.
- The proof sketch is well-written.
- The experiments nicely verify the theoretical results. Sufficient details are provided.

**Weaknesses:**

1. The results do not suggest that “ICL by AR pretraining is more difficult than ICL by few-shot pretraining”, as the authors claim. Instead, they suggest that ICL to predict the $T+1$-th token given the sequence $\mathbf{W}^0\mathbf{x}\_1, \dots, \mathbf{W}^{T-1}\mathbf{x}\_1$ is harder than ICL to predict $\mathbf{w}^\top \mathbf{x}_{T+1}$ given $\{(\mathbf{x}\_i, \mathbf{w}^\top \mathbf{x}\_i)\}\_{I=1}^T$ and $\mathbf{x}\_{T+1}$. These are two very different types of tasks and it is not fair to make conclusions about pretraining strategies based on their differing performance on these different tasks. Rather, one needs to compare the performance of a transformer trained in an autoregressive fashion and one trained on a few-shot learning objective under the same data/task model. In the data model in this, few-shot pertaining would use the same loss as the AR loss (equation 2) except that only the $t=T-1$ terms would remain in the loss. It is not clear whether this objective leads to better or worse solutions, in fact, this objective contains the same amount of information and therefore intuition suggests that it leads to solutions of the same quality. This is a key point as arguably the main message of the paper is that AR pretraining behaves differently than few-shot pertaining and thus the ICL community should focus on AR pretraining. I would suggest that the authors re-frame their discussion around the inadequacy of the regression data model considered by most of the existing ICL literature, as changing the data model is really the key contribution of this paper.

    2. The setting is very artificial such that the significance of the results seems dubious. The model consists only of one, linear attention unit, and the data is assumed to follow a deterministic linear system, i.e. $\mathbf{x}\_{i+1}=\mathbf{Wx}\_i$ where $\mathbf{W}$ is diagonal with diagonal entries having magnitude 1. The learning model and data model both being linear means that the ICL prediction of any learning model with parameters $\mathbf{W}^{KQ}\_{32}=a\mathbf{I}_d$ and $\mathbf{W}^V\_{12} = b\mathbf{I}\_d$ for $ab>0$ and all other parameters being zero can be interpreted as the prediction of a linear model $\hat{\mathbf{W}}$ trained directly to approximate $\mathbf{W}$ with one step of GD starting from $\mathbf{0}\_{d\times d}$. It is not clear at all how such a conclusion would follow if the softmax activation, shown to be crucial in practice, is reinstated, or if other components of the transformer (MLP layer, multiple attention heads, multiple attention layers, etc.) are added. Also, the token embedding is strange and redundant, with consecutive inputs $\mathbf{x}\_i, \mathbf{x}\_{i-1}$ concatenated in the same token. Another, though less significant, issue is that only the population loss is considered. The authors repeatedly claim that “We believe that our analysis can give insight to the establishment of theory in settings with more complex models” but never explain why.

    3. The model is already initialized as a model that implements gradient descent during ICL, and under the scalar-times-identity covariance assumption (4.1), or the enforced off-diagonal masking, it is not hard to show that the gradient flow dynamics reduce to the dynamics of two scalars ($a$ and $b$) which parameterize a set of models that always does GD during ICL. It is not clear whether the trained transformer can learn to perform GD during ICL if it is not initialized as a model that does this, and the experiments do not address this question.

    4. “Gradient clipping” is used to refer to the masking procedure that zeros out all off-diagonal elements of the gradient. This undersells the severity of this procedure — clipping is a non-zero-thresholding operation applied equally to all gradient elements and is commonly used in practice for optimization and privacy purposes, and does not use any prior knowledge of how the gradient should be structured. The masking procedure described here sends a particular subset of elements of the gradient to zero, to drive the dynamics towards a particular desired solution based on prior knowledge of the data-generating process, which is a much stronger, and impractical, procedure.

    5. Assumption 4.2 and Theorem 4.2 are tautological and not insightful. They should identify the properties of $\mathbf{x}\_1$ and $\mathbf{W}$ that lead to such behavior.

    6. $\kappa\_3$ has $\Omega(d)$ growth, which means the effective step size of the gradient descent step is $\tilde{O}(T/d)$, meaning $T$ (number of examples in the context length) must be $\tilde{\Omega}(d)$ for accurate predictions, which makes sense for this problem setting, but suggests the proposed setting is far from capturing ICL ability in practice, where the number of in-context examples is far fewer than the token embedding dimension. In the experiments, $T=20d$, which means the experiments do not evaluate whether the results can extend to smaller-context scenarios.

    7. Proposition 4.2 is not meaningful because it does not rule out whether a different initialization could lead to a model that performs GD.

    8. Even if we accept all of the simplifications and assumptions (including very long context) made to obtain a trained model that performs GD during ICL, this trained model does not learn the correct step size for GD, so it can’t perform ICL. To me, this suggests that GD is *not* the mechanism that transformers use to perform ICL — even in such an idealized setting, GD cannot explain the ability of transformers to perform ICL. However, the messaging around the paper is not consistent with this, as it continues to push the narrative that trained transformers are mesa-optimizers.

**Questions:**

Why is the y-axis in Figure 1-b defined as the ratio of two vectors?

**Limitations:**

The authors discuss limitations.

---

> ### Author Rebuttal · Authors · 2024-08-06
>
> # Response to Reviewer c1Lc
> We thank Reviewer c1Lc for the valuable comments.
> ## Weakness1: key contribution
> Thanks for the insightful suggestion. We agree that we do not have direct evidence that "ICL by AR pretraining is more difficult than ICL by few-shot pretraining" because the data model is changed. We will revise it to "ICL by AR pretraining is quite different from that by few-shot pretraining" following your suggestion.
>
> However, **we emphasize that our key contribution is understanding the ICL ability of AR pretraining, rather than changing the data model**. **First**, **adopting the new data model is not our target but a necessary approach** to investigate AR pretraining because the few-shot regression dataset in the previous ICL theory is not suitable since each input $x_{i+1}$ does not have a relation with the context. **Second**, we additionally need to scale the embedding strategy, the attention model, and the loss function to perform the full AR pretraining. All these parts are not well studied in the literature. **Third**, we are the first to investigate the training dynamics and the mesa-optimization hypothesis in this setting, which deepens people's understanding of autoregressive transformers.
>
> Besides, due to the non-trivial nature of the AR pretraining, we think our results will highlight the role of other components in practical transformers (more attention layers, MLP, etc), and boost researchers to study autoregressive transformers.
>
> ## Weakness2: artificial setting
> Thanks for the advice. **We note that all settings adopted in this paper are reasonable and have attracted attention from the recent ICL theory, which is also agreed by Reviewer WaqB.** We clarify key points here, and we will discuss these in more detail in the final version.
> 1. Single-layer one-head linear attention has been a popular setting in the recent ICL theory for few-shot regression [24,25,28,29,27]. As for the AR pretraining, the model in this paper is more complex and has also been adopted in recent works [16,17]. However, they do not investigate its training dynamics thus the convergence properties are still unclear.
> 2. Linear data model is proposed in [16] and simplified in [17] for tractable analysis. We use the same data model as [17], while they only analyze the loss landscape under the diagonal assumption for model parameters.
> 3. Token embeddings in this paper have been used in [16,17]. It is a natural extension of the embeddings in existing ICL theory because we need to predict each historical token here. In fact, practical transformers do learn similar token construction in the first softmax attention layer (see Fig. 4B in [16]).
> 4. Population loss is a common setting in ICL theory which considers loss landscape and training dynamics [17,24,28,49,51,52].
>
> **Implications on more complex models**.  We will revise the claim to "We believe that our analysis can give insight into the establishment of theory in settings with autoregressive transformers". Besides. analogous to the development of ICL theory for few-shot regression (MLP [52], softmax attention [30], etc), future works considering more complex transformers can adopt the same data model, token embeddings in this paper, and try to use the similar proof technique to derive the training dynamics.
> ## Weakness3: initialization & keeping diagonal structure
> First, the diagonal initialization is reasonable and we further conduct simulations with standard initialization. Second, we note that the findings of Assumption 4.1 and derivation are not trivial. Please see the details in the common concern 1 and 2, respectively.
> ## Weakness4: gradient clipping
> **We note that our target in Theorem 4.3 is to complement the theoretical result in [17]** from the perspective of optimization, rather than provide a theory for practical gradient clipping. We will change "gradient clipping" to "gradient masking" in the final version to avoid confusion.
> ## Weakness5: Assumption 4.2 and Theorem 4.2
> We think the sufficient and necessary condition Assumption 4.2 might not be further simplified and we have discussed some properties of $x_1$ in line 238-245. For Theorem 4.2, we will follow the reviewer's suggestion to move Theorem 4.2 to the Appendix.
> ## Weakness6: effect of $d$
> In the final version, **we will clarify that we fix $d$ as a finite number and focus on the asymptotic property w.r.t. $T$**. Besides, we conduct additional experiments in smaller-context scenarios $(d=5, T=3/5/10, x_1\sim N(0, I_d))$. As a result, the product $ab$ does converge to the results in Theorem 4.1. **The representative results can be found in the latest uploaded PDF**.
> ## Weakness7: Proposition 4.2
> **This is a misunderstanding**. In the final version, we will clarify that **Proposition 4.2 does not depend on diagonal initialization**. As we have mentioned in line 272-274, parameters that perform GD are not the critical points of the loss function. Therefore, the trained transformer will not perform vanilla gradient descent with any initialization.
> ## Weakness8: rationality of mesa-optimization
> **We disagree with this with our highest respect**. From the result of this paper, we can only conclude that **one-step GD is not enough to learn the data distribution**. However, even for more complex data ($W$ is not diagonal), **[16] has empirically verified that multi-layer linear attention can perform multi-step GD to learn the data distribution**. The theory for the multi-layer attention in AR pretraining setting is a meaningful future work.
> ## Question 1: ratio of two vectors
> As we have mentioned in line 326, we want to verify the correctness of Proposition 4.1, which claims that the prediction will converge to 1/5 of the true data. Therefore, we present the ratio and find it does converge to 1/5. In practice, we estimate this ratio by calculating the mean of the element-wise divide between two vectors.
>
> We will clarify this detail more carefully in the final version.

---

> > ### Comment · Reviewer_c1Lc · 2024-08-10
> >
> > I thank the reviewers for their thorough response, which has helped to alleviate my concerns. I especially appreciate the experimental results with non-diagonal initialization, pointers to related work and clarification on Proposition 4.2. I am still not overly impressed by the results, especially as they start with diagonal initialization (Theorem 4.1 and Theorem 4.3), and use gradient masking (Theorem 4.3). Nevertheless, as the authors point out, even in these simplified settings characterizing the limiting points of gradient flow is a novel contribution. I would recommend that in the revision, the authors put more emphasis on their results contributing a closed-form characterization of the convergence points of AR pre-training, rather than showing that AR pre-training results in a mesa-optimizer, since the latter point is weakened by the assumption that the initialization is already a mesa-optimizer. As a minor point, the empirical results with T=3 and T=10 appear to be missing from the attached pdf.

---

> ### Author Response · Authors · 2024-08-11
>
> Thanks for the helpful comments and positive feedback, as well as for increasing the score. We are glad that we have addressed the reviewer's main concerns.
>
> For the claim of theoretical contribution, we agree with the reviewer's suggestion. **We promise that we will put more emphasis on our results contributing a closed-form characterization of the limiting points of AR pre-training**. We will also **discuss the necessity and the limitation of the diagonal initialization more clearly** in the final version.
>
> For the complete experimental results in small-context ($T=3/10$) and standard-initialization ($\sigma_w=0.001/0.1$) scenarios, **the claims in the common response still hold, and we will add them to the final version**. We note that due to the space limitation, we only present the representative result in each scenario during the rebuttal period.
>
> We thank the reviewer once again for the valuable suggestions and insightful comments, which definitely improve the quality of this paper a lot.

---

### Official Review · Reviewer_WaqB · 2024-07-10

**Soundness:** 3
**Presentation:** 4
**Contribution:** 4
**Rating:** 7
**Confidence:** 4

**Summary:**

This paper studies the emergence of mesa-optimization/in-context learning capabilities of Transformers. In particular, it attempts to fill the gap of our understanding of training dynamics and specifically on non-convex dynamics where the sequences are autoregressively generated as $\vec{x}\_{t+1} = W \vec{x}\_{t}$. There are many contributions of this paper. First, it shows, under Assumption 4.1, transformer converges to the gradient-descent conjecture for ICL. Under the same assumption, the authors show moments of the data is necessary & sufficient for ICL. Finally, this papers study data distributions beyond Assumption 4.1, and shows in this case, transformers are *not* doing gradient-descent for ICL.

**Strengths:**

Presentation is very clear and easy to follow.

Contribution is significant. It bridges the gap between two main stream hypothesis of how Transformers perform mesa-optimization/ICL: (1) Transformer is doing gradient descent [1]; (2) Transformers is doing something beyond vanilla gradient descent, such as preconditioned GD  [2] or higher-order optimization methods [3,4]. The authors concluded the divergence between these two hypotheses are due to the data assumption (Assumption 4.1).

[1] Johannes von Oswald, Eyvind Niklasson, Ettore Randazzo, João Sacramento, Alexander Mordvintsev, Andrey Zhmoginov, and Max Vladymyrov. Transformers learn in-context by gradient descent. In ICML, volume 202, pages 35151–35174, 2023.

[2] Kwangjun Ahn, Xiang Cheng, Hadi Daneshmand, and Suvrit Sra. Transformers learn to implement preconditioned gradient descent for in-context learning. In NeurIPS, 2023.

[3] Deqing Fu, Tian-Qi Chen, Robin Jia, and Vatsal Sharan. Transformers learn higher-order optimization methods for in-context learning: A study with linear models, 2023.

[4] Giannou, Angeliki, Liu Yang, Tianhao Wang, Dimitris Papailiopoulos, and Jason D. Lee. How Well Can Transformers Emulate In-context Newton's Method?. arXiv preprint arXiv:2403.03183 (2024).

**Weaknesses:**

When contrasting with vanilla gradient descent conjecture, the authors only mentions the alternative *preconditioned GD* hypothesis. However, there are many other propositions, such as higher-order optimization proposition, such as a *Newton's method* [3,4,5]. I would recommend the authors have a deeper discussion there. For example, in [5], they argued the GD++ method in [1] is actually a higher-order optimization method with superlinear convergence rate as discussed in [3,4].

[5] Vladymyrov, Max, Johannes von Oswald, Mark Sandler and Rong Ge. Linear Transformers are Versatile In-Context Learners. ArXiv abs/2402.14180 (2024)

**Questions:**

Theories are built upon a one-layer linear causal self-attention model, what happens if we extend to (1) more layers or (2) non-linear activations?

**Limitations:**

Theories are only built upon a one-layer linear causal self-attention model, where there's is still a big gap compared to what's used in practice -- many-layer non-linear activation self-attention and feed-forward model with skip connections. But for theory, this is good enough.

---

> ### Author Rebuttal · Authors · 2024-08-06
>
> # Response to Reviewer WaqB
>
> We thank Reviewer WaqB for the positive support and valuable comments, which can definitely improve the quality of this paper.
>
> ## Weakness 1: additional related work
>
> Thanks for the helpful suggestion. We will cite and discuss the mentioned papers in the final version. We believe that the higher-order optimization hypothesis is very important for understanding practical ICL, especially in multi-layer cases.
>
> ## Question 1: multi-layer and non-linear activations
>
> Thanks for the nice suggestion. We discuss two parts respectively.
>
> ### Multiple layers
> We think we can explore this setting from expressive power and optimization/generalization perspectives, respectively.
>
> 1. Expressive power. We can try to construct the cases where multi-layer transformers perform multi-layer higher-order optimization (e.g., GD++, Newton's method) in the AR pretraining setting. We think [16] can provide many practical insights, and [a,b] can provide many theoretical insights.
> 2. Optimization/generalization. In this case, our work can be seen as a foundation. We can try to investigate the training dynamics and the generalization ability of the trained transformers. Here, we think [c] can also give some theoretical insights, which discuss the multi-layer attention in the regression setting.
>
> ### Non-linear activations
> We only discuss the softmax activation here because it is used most widely in practice.
>
> 1. [16] shows that there exists a strong connection between softmax attention and GD in the AR pretraining setting, but the mechanism is still unclear.
> 2. [16] also shows that in the AR pretraining, the first softmax layer can learn the copy behavior and obtain the embedding construction in this paper, but the mechanism is also still unclear. To this end, we think [51] can give some insights, which prove similar copy behavior of the softmax attention.
>
> We will discuss these in detail in the final version.
>
> [a] Transformers learn higher-order optimization methods for in-context learning: A study with linear models, 2023.
>
> [b] How Well Can Transformers Emulate In-context Newton's Method?, 2024
>
> [c] Can Looped Transformers Learn to Implement Multi-step Gradient Descent for In-context Learning?, ICML, 2024

---

> ### Author Response · Authors · 2024-08-13
>
> We thank the reviewer once again for the insightful comments and positive score, which give us much confidence to finish this rebuttal process.

---

### Official Review · Reviewer_dsNs · 2024-07-12

**Soundness:** 2
**Presentation:** 3
**Contribution:** 2
**Rating:** 5
**Confidence:** 4

**Summary:**

- The authors study the problem of in-context learning an autoregressive (AR) process (defined with a uniformly drawn diagonal unitary transformation) with a one-layer linear causal self-attention model, trained by gradient flow on square loss.
- Under a specific parameter initialization scheme and a distributional assumption on the initial token, they show that a trained model can implement one step of gradient descent (=mesa-optimizer) applied to a least-square problem estimating the transformation matrix of the AR process.
- Unfortunately, they show that the obtained mesa-optimizer can fail to learn the AR process with next-token prediction, even with infinitely many training samples and infinitely long test context length.
- They further propose a stronger assumption being a necessary and sufficient condition for the success of mesa-optimizer. They also provide an example of the distribution of the initial token satisfying the necessary and sufficient condition.
- They show that, without the distributional assumption, the trained model may not implement a gradient descent step.
- Lastly, they verify their theory with simulations.

**Strengths:**

- The paper is well-written overall.
- Most of the theoretical analyses seem correct and rigorous. The proof sketch in Section 5 provides a clear and plausible picture of the proof.
- The analyses do not require/assume the diagonal structure of submatrices of the parameters. Instead, the authors found a sufficient condition (a distributional assumption on the initial token) so that they can prove the diagonal structure.

**Weaknesses:**

- W1. The analyses and experiments cannot provide insight into the general cases of autoregressive training of Transformers.
    - This is mainly because the theoretical analyses greatly depend on the particular initialization scheme (Assumption 3.1). The initializations of weight matrices are too sparse: at initialization (and throughout training), they only focus on the relationship between very particular token embeddings. This seems very unnatural, thinking of the training transformers in practice. Quite ironically, the paper’s result corroborates that the initialization scheme is inappropriate for solving in-context learning with next-token prediction, under most of the plausible data distribution.
    - This assumption might be inevitable for a tractable analysis. However, the experiments are also confined to the framework under Assumption 3.1. I think the authors should have demonstrated many more simulations with various initialization schemes and even with diverse architectures (linear transformers with non-combined weight matrices, non-linear transformers…) to support their arguments (”the data distribution matters for the emergence of mesa-optimization”, “there is a capability limitation of the mesa-optimizer”…).
- W2. There are some weaknesses in theoretical analyses.
    - Even though the authors found a condition for the emergence of mesa-optimization in a very particular problem setting, the analysis does not go beyond the “diagonal structure” of weight matrices. Thus, for me, although the contribution is novel, it does not look significant.
    - For the comments and questions on Proposition 4.1, see Q3.
- W3. Minor comments on typos & mistakes.
    - Line 130: $W^{KQ} = {W^K}^\ast W^Q$ (rather than transpose)
    - Equation (2): I guess the sum should be applied over $t=1, …, T-1$.
    - Line 182: “$x_1 \in \mathbb{R}^d$”
    - Lines 207, 289: ‘$W$’ must be bold.
    - Figures 1b, 1d, and Line 330: “$T_{te} -1$”

**Questions:**

- Q1. In lines 186-187, you mentioned that any random vectors with i.i.d. sampled coordinates satisfy Assumption 4.1. Is it true? I think they only satisfy the first part of the assumption (before “In addition”). As a counterexample of the second part, if the distribution of each coordinate is heavy-tailed (e.g., infinite second moment), $\kappa_1$ and $\kappa_2$ are infinite.
- Q2. In Theorem 4.1, does different initialization $(a_0, b_0)$ “may” lead to different $(\tilde{a}, \tilde{b})$, or “always” lead to different point?
- Q3. Questions on Proposition 4.1
    - Isn’t $T_{te} \rightarrow \infty$ unnecessary? This is because we can show that for any $i\in [T_{te}-1]$, $\mathbb{E}\_{x_1} [x_i x_i^\ast] = W^{i-1} \mathbb{E}_{x_1} [x_1 x_1^\top] (W^\ast)^{i-1} = \sigma^2 W^{i-1}(W^\ast)^{i-1} = \sigma^2 I_d $ because $W$ is unitary and $x_1 \in \mathcal{N}(0_d, \sigma^2 I_d)$.
    - Isn’t $T_{tr} \rightarrow \infty$ also unnecessary? This is because it seems enough to show that $\tilde{a} \tilde{b} < \tfrac{1}{5}$.
    - Most importantly, is the last statement **correct**? Shouldn’t you prove that $\mathbb{E}_{x_1} [\hat{y}_T - Wx_T]$ converges to a nonzero vector? This is because $x_T = W^{T-1} x_1$ is also a random vector and depends on $T$.
    - From what I read, it seems like the data distribution $\mathcal{D}_{x_1}$ is the main cause of the failure of the mesa-optimizer. In my opinion, however, this is also because the “single-layer” linear self-attention model can only simulate a “single step” of GD for the OLS problem. Then isn’t the number of layers the problem? I think there are several reasons why the mesa-optimizer fails, and the authors should mention and analyze these factors as many as possible.
- Q4. I guess Theorem 4.2 requires Assumption 3.1. Or does it?

If the authors address all my concerns, I will definitely raise my score.

**Limitations:**

The limitations are provided in Section 7.

---

> ### Author Rebuttal · Authors · 2024-08-06
>
> # Response to Reviewer dsNs
> We thank Reviewer dsNs for the positive feedback and valuable comments, which can improve the quality of this paper.
> ## Weakness1: initialization & negative results
> Thanks for the nice suggestion. We discuss two concerns respectively.
> ### Initialization
> We argue the diagonal initialization is reasonable and further conduct simulations with standard initialization. Please see the details in common concern 1.
> ### Negative results
> The results show that one-step GD learned by the transformer can not recover the distribution, but this can be solved by more complex models. Even for more complex data ($W$ is not diagonal), [16] has empirically verified that multi-layer linear attention can perform multi-step gradient descent to learn the data distribution. Our work can be seen as a foundation for the theory of more practical transformers.
> ## Weakness2.1: diagonal structure
> Thanks for the suggestion. The findings of Assumption 4.1 and analysis are not trivial. Please see the details in the common concern 2.
> ## Weakness2.2 & Q3: questions on Proposition 4.1
> Thanks for the nice comment, which definitely helps us clarify Proposition 4.1!
> ### Refined Proposition 4.1
> First, we clarify the original intention of Proposition 4.1. We want to prove that in expectation w.r.t. data ($x_1$ and $W$), when the training sequence length $T_{tr}$ is large, the prediction $\hat{y}_ {T_ {te}}=W(\tilde{a}\tilde{b}\frac{\sum_ {i=1}^{T_ {te}-1}x_ ix_ i^*}{T_ {te}-1})x_ {T_ {te}}$ of the trained transformer converges to $1/5 * Wx_ {T_ {te}}$, rather than the true target $Wx_{T_{te}}$. In the submitted version, we prove this by claiming $E_{x_1}[\tilde{a}\tilde{b}\frac{\sum_{i=1}^{T_{te}-1}x_ix_i^*}{T_{te}-1}] \to \frac{1}{5}I_d$, which is not rigorous as the reviewer points. We modify the proposition rigorously as follows.
>
> **Refined Proposition 4.1**: Let $D_{x_1}$ be the normal distribution $N(0_d, \sigma^2I_d)$ with $\sigma^2 > 0$, then the AR process can not be recovered by the trained transformer even with long context. Formally, when the training sequence length $T_{tr}$ is large, for any fixed test context length $T_{te}$ and dimension $j\in [d]$, the prediction satisfies $E_ {x_ 1,W}[\frac{(\widehat{y}_ {T_ {te}})_ j}{(Wx_ {T_ {te}})_ j}] \to \frac{1}{5}$. Therefore, the prediction will not converge to the true next token.
>
> **Proof skeleton:** By directly calculating, we have
> $(Wx_ {T_ {te}})_ j = (W^{T_ {te}}x_ 1)_  j = \lambda_ j^{T_ {te}} x_ {1j}$, and $(\hat{y}_ {T_ {te}})_  j = \frac{\tilde{a}\tilde{b}}{T_ {te}-1} \sum_ {i=1}^{T_ {te}-1} \sum_ {k=1}^d \lambda_ j^i \lambda_ k^{T_ {te}-i} x_ {1j}x_ {1k}^2$.
> Therefore, we have
> $$
> E_ {x_ 1,W}[\frac{(\hat{y}_ {T_ {te}})_ j}{(Wx_ {T_ {te}})_ j}] = E_ {x_ 1,W}[\frac{\tilde{a}\tilde{b}}{T_ {te}-1} \sum_ {i=1}^{T_ {te}-1} \sum_ {k=1}^d \lambda_ j^{i-T_ {te}} \lambda_ k^{T_ {te}-i} x_ {1k}^2] = E_ {x_ 1}[\frac{\widetilde{a}\widetilde{b}}{T_ {te}-1} \sum_ {i=1}^{T_ {te}-1}  x_ {1j}^2]= \tilde{a}\tilde{b} \sigma^2.
> $$
> Since $\tilde{a}\tilde{b} < \frac{1}{5\sigma^2}$ and converges to $\frac{1}{5\sigma^2}$ when $T_{tr}$ is large, the proof is completed.
>
> ### Answers to Q3
> Q3.1: Yes, though for the new proof, the $T_{te}$ does not need to be large.
>
> Q3.2: Yes, $\widetilde{a}\widetilde{b} < \frac{1}{5\sigma^2}$ and thus we can prove the prediction will not converge to the true next token. However, we consider the long-context scenarios ($T_{tr}$ is large) to improve the readability and make the experimental verification more convenient. Furthermore, we conduct additional experiments in small-context scenarios and the representative results can be found in the latest uploaded PDF.
>
> Q3.3: This is a very helpful comment that inspires us to refine the proposition. We note that $E_ {x_ 1,W}[\widehat{y}_ {T_ {te}} - Wx_ {T_ {te}}] = \frac{1}{5}*0 - 0 = 0$ in this case, which fails to interpret the phenomenon. Thus, we choose to calculate the expectation of the ratio. The experiments in the main paper are also conducted to estimate the ratio with 10$k$ test examples (randomly sampled $x_1, W$), which verify the refined theoretical results here.
>
> Q3.4: Yes, the number of layers influences the effect of the mesa-optimizer largely. However, the main message from our paper is that given a **fixed** number of layers, the obtained mesa-optimizer has limited capability, and we investigate this rigorously. For multi-layer attention, [16] has empirically found it performs multi-step (accelerated) GD to solve the problem well. Theory for this case is a meaningful future work.
>
> ## Weakness3 & Q4: typos
> Thank you for the careful reading! We will modify these typos in the final version. Here, we only clarify that the summation in Eq. (2) is applied over 2 to $T-1$, which is also adopted in recent AR theory [17]. The reasons are as follows.
> 1. Given only $x_1$, we do not have any information to predict $x_2$.
> 2. Since we set $\rho_t = t-1$ according to the convention in ICL theory, $\rho_1 = 0$ will lead to the exploration of the transformer's output.
>
> We will discuss these in more detail in the final version.
>
> ## Q1: distributions that satisfy the Assumption 3.1
> Yes, thank you for the careful reading. We will modify the claim to "any random vectors $x_1$ whose coordinates are i.i.d. random variables with zero mean and finite moments satisfy this assumption". This class still includes many common distributions like the normal distribution, thus it is still meaningful.
>
> ## Q2: convergence of (a,b)
> Different initialization $(a_0, b_0)$ always lead to different limiting points $(\tilde{a},\tilde{b})$. For example, in the experiments with $x_1$ from $N(0_d, I_d)$, we approximately have
> 1. $(a_0,b_0) = (0.1,0.1) \to (\tilde{a},\tilde{b}) = (0.44,0.44)$
> 2. $(a_0,b_0) = (0.5,1.5) \to (\tilde{a},\tilde{b}) = (1.41, 0.14)$
> 3. $(a_0,b_0) = (1.5,0.5) \to (\tilde{a},\tilde{b}) = (0.14, 1.41)$
>
> We will definitely discuss these more clearly in the final version.

---

> > ### Comment · Reviewer_dsNs · 2024-08-10
> > **Thank you for the rebuttal**
> >
> > Thank you for your detailed response.
> >
> > - The general response and the additional experiments on standard initialization have addressed my concerns (W1 & W2). I also agree that finding Assumption 4.1 is non-trivial.
> > - Also, I am happy to see that the refined Proposition 4.1 and its proof sketch have satisfactorily addressed my concern on the theoretical contribution.
> > - Regarding the response to Q1, I think it is important to mention which orders of moments are finite.
> >
> > Overall, most of my concerns are addressed. Although I am still not absolutely certain, now I think the paper is above the acceptance bar. Hence I will raise my score to 5.

---

> ### Author Response · Authors · 2024-08-10
>
> We are glad that we have addressed most of the reviewer's concerns. For Q1, we will follow the reviewer's suggestion and further modify the claim to "any random vectors whose coordinates are i.i.d. random variables with zero mean and finite moments of order 2, 4, and 6 satisfy this assumption".
>
> We thank the reviewer once again for the valuable suggestions and insightful comments, which improve the quality of this paper.

---

### Official Review · Reviewer_ykMD · 2024-07-15

**Soundness:** 3
**Presentation:** 4
**Contribution:** 3
**Rating:** 5
**Confidence:** 3

**Summary:**

This paper studies the autoregressive training of a one-layer linear attention model for in-context learning of first-order AR processes. It is shown that under certain distribution of the initial point, the gradient flow on the population next-token prediction loss will converge to a model that makes the prediction based on the estimate given by one step of gradient descent over the in-context loss. Furthermore, a sufficient and necessary condition is provided to characterize learning capability of the trained model. Besides, it is shown that for more general data distribution, the trained model does not perform one step of gradient descent over the in-context OLS loss. Numerical simulations are conducted to support the theory.

**Strengths:**

The paper is well written and easy to follow. The explanation of the results is very clear, and logic flow of the organization of the paper is smooth. The current results present a solid contribution to the theoretical understanding of the in-context learning by transformers, especially for AR training.

**Weaknesses:**

It would be helpful to further highlight the technical contribution. In particular, how does the technique and analysis differ from those existing papers on training dynamics of transformers for in-context learning of regression problems?

**Questions:**

What's the value of the normalization factor $\rho_t$? This seems to be a non-standard design as $\rho_t$ depends on $t$. From my reading of line 634, it requires that $\rho_t=t-1$. Please clarify this and explain why this is reasonable.

**Limitations:**

The authors have adequately addressed the limitations.

---

> ### Author Rebuttal · Authors · 2024-08-06
>
> # Response to Reviewer ykMD
> We thank Reviewer ykMD for the positive score and valuable comments, which can definitely improve the quality of this paper.
> ## Weakness 1: further highlight the technical contribution
> Thanks for the nice suggestion. We consider the novel AR pertaining which makes our setting more complex. **First**, compared to ICL for regression, our data model breaks the independence between data at different times, which causes difficulty in decomposing and estimating the gradient terms. **Second**, we additionally modify the embedding strategy (more dimensions), scale the attention model (much more parameters), and change the loss function (more terms) to perform the full AR pertaining. All these parts are not well studied in the literature and make the gradients more complicated.
>
> We will discuss this in detail in the final version.
>
> ## Question 1: normalization factor $\rho_t$
> Yes, $\rho_t = t-1$ as we defined in line 134. **This setting follows the convention in ICL theory for regression problems**. We need this length-aware normalization factor to balance the outputs along different times because each element in $E_tE_t^*$ is an inner product of two vectors of size $t$, which scales as the $t$ increases. In ICL theory for regression problems, we only need to predict the last token, where the width of the embedding matrix $E$ is $T+1$ and the $\rho$ is defined as $T$. In this paper, the width of $E_t$ is $t$ thus we define $\rho_t = t-1$.
>
> In practice, this length-aware normalization factor is a non-standard design mainly due to the existence of the softmax operator, which naturally normalizes the outputs. The theory for softmax attention in the AR setting is a meaningful direction.
>
> We will definitely discuss this in detail in the final version.

---

> > ### Comment · Reviewer_ykMD · 2024-08-11
> > **Reply to rebuttal**
> >
> > I thank the authors for the explanation. I don't have further questions.

---

> > > ### Author Response · Authors · 2024-08-12
> > >
> > > Thanks for the helpful comments and positive score. We are glad that we have addressed the reviewer's concerns.

---

### Author Rebuttal · Authors · 2024-08-06

# Common concerns from reviewers
We thank all reviewers for their valuable and constructive comments. We address the common concerns here and post a point-to-point response to each reviewer as well. We believe the quality of the paper has been improved following the reviewers' suggestions.

## Common concern 1: diagonal initialization (from Reviewer dsNs and c1Lc)
1. We agree with Reviewer dsNs that this assumption might be inevitable for a tractable analysis here. However, we note that **diagonal initialization/structure has been used in recent ICL theory for casual transformers [17,30, 51]**. Especially, as we have mentioned repeatedly in the paper, the most related paper [17] considers a stronger diagonal structure than ours, but they only investigate the loss landscape. Our results definitely deepened the understanding of autoregressive transformers by considering practical training dynamics.
2. **We have added the new experimental results under standard initialization with different scales**. We initialize each parameter using normal distribution $N(0,\sigma_w^2)$, where $\sigma_w$ is chosen from $(10^{-3},10^{-2},10^{-1})$, respectively. The results are summarized as follows and **their representatives can be found in the latest uploaded PDF**.  We think it improves our experimental part a lot.
    1. Though the convergence results of parameters are not exactly the same as those under diagonal initialization, **they keep the same diagonal structure, which can be understood as GD with adaptive learning rate in different dimensions**.
    2. The test results (ratio, MSE loss) of the trained transformers under standard initialization **are the same as** those under diagonal initialization, which further verifies the capability of the trained transformers.

To sum up, the experimental results demonstrate that our theoretical results have a certain representativeness, which further supports the rationality of the diagonal initialization.

For the more complex architecture (concern from Reviewer dsNs), we kindly refer the reviewer to the existing work [16] that includes results of (multi-layer) linear attention, (multi-layer) softmax attention, and full transformer on similar AR process ($W$ is non-diagonal) with standard initialization. They have established a strong connection between practical transformers and multi-step GD, and the theory for more complex architectures is meaningful for future work.

We will discuss these in more detail in the final version.

## Common concern 2: keep diagonal structure (from Review dsNs and c1Lc)
1. Finding the condition for the emergence of mesa-optimization (Assumption 4.1) is non-trivial since we need to derive the training dynamics first. However, the derivation of the dynamics is more difficult than the existing ICL theory for regression [24] because our data, model, token construction, and loss function become more complex.
2. This paper does deepen the understanding of autoregressive transformers compared to previous studies, and the extended experiments show that the theory has a certain representativeness. For the general convergence results, as we have discussed in line 279-283, the computation of the gradient will be much more difficult and might be intractable, which is agreed by Review dsNs. Therefore, we leave the rigorous result of general convergence for future work.

We will discuss these in more detail in the final version.

---

### Author Response · Authors · 2024-08-13
**Summary of the revision**

# Summary of the revision

We sincerely thank all reviewers for their valuable comments and positive support, which help us further improve the quality of this paper. We also thank ACs for organizing this nice review process. We have thoroughly addressed the detailed comments and summarized the revision in the final version as follows.

## New experimental results
1. We add complete experimental results under standard initialization with different scales ($\sigma_w = 0.001/0.01/0.1$). Representatives of them have been presented in the latest uploaded PDF (Fig. a-f).
2. We add complete experimental results in small-context scenarios ($T = 3/5/10,d=5$). Representative results have been presented in the latest uploaded PDF (Fig. g-i).

## Refined theoretical result
We will refine the theory and proof of Proposition 4.1 as we presented in the answer to Reviewer dsNs.

## Writing
1. We fix the typos pointed out by Reviewer dsNs.
2. We clarify that Proposition 4.2 does not depend on diagonal initialization
3. We change "gradient clipping" to "gradient masking" in Section 4.3 to avoid confusion.
4. We clarify the ratio calculation between prediction and true vector in Fig. 1b.

## Discussion

### Contribution & Impact
1. We add more discussion about the key contribution of this paper in Section 4.
2. We add more discussion about our technical improvement in Sections 4 and 5, compared with existing studies on ICL.
3. We add more discussion about the contribution and the necessity of finding Assumption 4.1 to keep the diagonal structure, compared with existing studies. We also discuss its limitations.
4. We add more discussion about the negative results of one-layer linear attention, which can be solved when considering more complex transformers in the future.
5. We add more discussion about the impact of our work on more complex autoregressive transformers (e.g., multi-layer attention, non-linear activations).

### Setting & Assumptions
1. We add more discussion about the rationality of our setting (model, data distribution, token construction, and normalization factor).
2. We add more discussion about the necessity and limitation of the diagonal initialization near Assumption 3.1 and Theorem 4.1.
3. We discuss how the experimental results under standard initialization extend our theory and support the rationality of the diagonal initialization in Section 6.
4. We add more discussion about the properties of Assumption 4.2.

### Related work
1. We cite and discuss the additional related papers mentioned by Reviewer WaqB.

---

### Decision · Program_Chairs · 2024-09-25

**Decision:**

Accept (poster)

**Comment:**

There is consensus among all reviewers that the paper presents a novel significant contribution and is clearly written. Moreover, several reviewers checked the proofs.

We recommend acceptance as a poster.

To the authors: please make sure to take reviewer comments and to fix remaining typos in the final revision.